# Time or distance encoding by hippocampal neurons via heterogeneous ramping rates

Raphael Heldman[1,2,3], Dongyan Pang[1], Xiaoliang Zhao ⦿[1], Brett Mensh[4,5] & Yingxue Wang ⦿[1] ✉

To navigate their environments effectively, animals frequently track time elapsed or distance traveled while seeking food and avoiding threats. The hippocampus is implicated in this process, but the neural mechanisms remain unclear. Using virtual reality tasks that require mice to integrate time or distance to collect a reward, we identified two previously unknown functional subpopulations of CA1 pyramidal neurons. Both subpopulations encode time or distance via distinct ramping dynamics. The first subpopulation exhibits a rapid, synchronous rise in activity upon movement-initiated integration. Subsequently, individual neurons ramp down at heterogeneous rates, creating progressively diverging firing rates that encode elapsed time or distance. Closed-loop optogenetic inactivation of somatostatin-positive (SST) interneurons counterintuitively reduced the ramping activity, leading mice to prematurely attempt reward collection, suggesting impaired time/distance estimation. Conversely, the second CA1 subpopulation shows opposite dynamics — an initial rapid suppression followed by a gradual ramp-up. Inactivating parvalbumin-positive (PV) interneurons diminished this initial suppression, resulting in transient attempts to collect reward near integration onset. These findings reveal parallel hippocampal circuits that initiate and maintain time or distance encoding, controlled by PV and SST interneurons, respectively, and provide insights into the neural computations supporting goal-directed navigation.

When navigating a dimly lit path at night, we often rely on our sense of distance traveled to determine our location. This process, known as path integration, involves the hippocampus, which integrates self-motion information to estimate distance relative to recognizable landmarks, such as a distinctive tree or a notable building[1–4].

In well-lit environments rich with cues, some hippocampal pyramidal neurons, called place cells, preferentially activate at specific locations[5]. Yet, even in dimly lit environments or when environmental cues are sparse or absent, neurons can still activate at particular distances or time points relative to a landmark. These distance- or time-specific activity patterns are driven mainly by internal computation

rather than external cues, and are thus termed internally generated firing fields (IGFs)[6–12]. Collectively, neurons with IGFs activate in sequence to produce an internally generated sequence (IGS), which encodes moment-to-moment distance or time to aid path integration.

Recent research has advanced our understanding of the circuit-level mechanisms that govern place cell and IGF neuron activity, highlighting the critical role of local inhibitory interneurons. Different subtypes of interneurons are known to exhibit distinct temporal dynamics and target specific subcellular compartments of pyramidal neurons[13]. For example, somatostatin-positive (SST) interneurons primarily inhibit dendrites, whereas parvalbumin-positive (PV)

[1]Max Planck Florida Institute for Neuroscience, One Max Planck Way, Jupiter, FL, USA. [2]Florida Atlantic University, Boca Raton, FL, USA. [3]IMPRS for Synapses and Circuits, Jupiter, FL, USA. [4]Janelia Research Campus, 19700 Helix Way, Ashburn, VA, USA. [5]Optimize Science, 37 Joseph Ct., San Rafael, CA, USA. ✉e-mail: Yingxue.Wang@mpfi.org

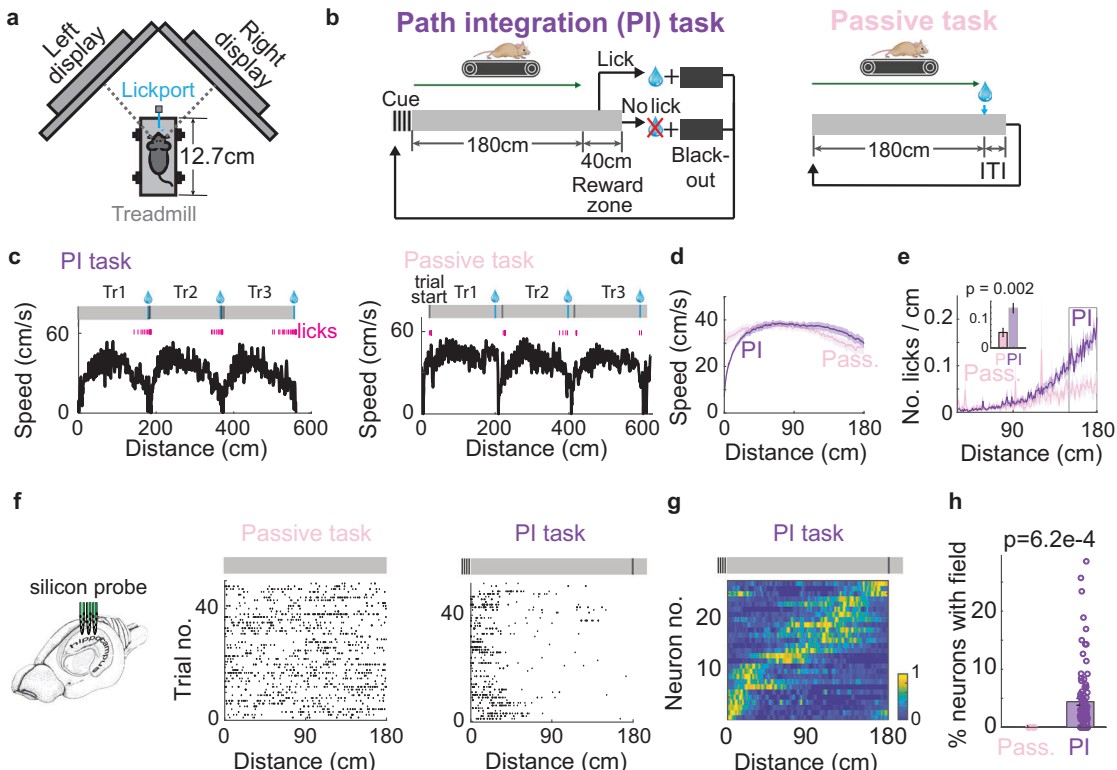

**Fig. 1 | A path integration task that expresses internally generated fields (IGFs).** **a** Schematic of the virtual reality setup: The mouse is head-fixed, with the eyes aligned with the two monitors. **b** Task schematics for PI (left) and passive (right) tasks (see Methods). **c** Example speed traces (black) and licking activity (magenta) plotted against distance during three consecutive trials for mice performing the PI task (left) or passive task (right). TrN: Trial N. Averaged speed traces (**d**) and lick histograms (**e**) across mice trained in either the passive (3 animals, 8 recordings, in pink) or PI task (4 animals, 10 recordings, in purple). The shaded area represents SEM (Same for other figures if not explicitly stated). The inset bar plot compares the number of licks/cm within 150–180 cm between the two tasks. PI recordings are selected to match the mean speed of the passive recordings. **f** Schematic of recordings with a silicon probe in the hippocampus (left). Spike rasters of example pyramidal neurons recorded in the passive (middle, no firing field) and PI task (right, with a firing field), plotted over distance within the cue-constant segment. (Adapted from Ref 7, Copyright © 2014, Springer Nature America, Inc. The same copyright notice applies to all other figures adapted from this image.) **g** An example IGS within one session, plotting over distance within the cue-constant segment. Each row shows one neuron's normalized firing rate change averaged over trials. **h** The percentage of neurons with fields per recording in the passive (3 animals, 8 recordings) and PI tasks (28 animals, 102 recordings). Each circle represents one recording.

interneurons preferentially inhibit the perisomatic region[13]. This compartment-specific inhibition regulates synaptic integration and spike output[14–17], which in turn shapes the spatial and temporal firing patterns of place cells[18,19]. Similar inhibitory mechanisms likely apply to neurons with IGFs.

However, given that only a subset of active pyramidal neurons exhibit IGF, a critical question arises: How do the broader population dynamics in CA1 support path integration, and how do inhibitory interneurons contribute to these broader dynamic patterns?

In this study, we uncovered a neural code in the hippocampal CA1 area, comprising two previously unknown functional subpopulations of neurons that track distance or time through their ramping activity. Investigating local inhibitory interneurons, we found that SST and PV interneurons[18–20] played distinct roles in shaping the activity of these subpopulations and influencing the accuracy of path integration. Together, our study reveals two parallel neural circuits within CA1 that complement each other to enable distance or time integration, offering insights into the circuit-level mechanisms underpinning navigation and memory encoding.

## Results

### A behavioral task that requires distance integration generates IGSs

To investigate neuronal dynamics during path integration, we developed a virtual reality (VR)-based behavioral task that required head-fixed mice to estimate distance by accumulating self-motion — a task we refer to as the path integration (PI) task (Fig. 1a). In each trial, following a brief visual cue, the animal had to run 180 cm through a constant gray VR background. This 180 cm run established a cue-constant segment. To receive a water reward, the animal was required to lick a designated port within a reward zone (Fig. 1b left; see Methods).

Trained animals displayed stereotypical running speed profiles and consistent licking patterns across trials (Fig. 1c left, d, e); notably, accurate performance depended on an intact hippocampus (Supplementary Fig. 1a–d). Animals typically initiated licking as they approached the end of the cue-constant segment (Fig. 1e; mean number of licks/cm from 100 to 180 cm: 0.087 ± 0.011. Mean ± SEM and Wilcoxon rank-sum test are used if not explicitly stated), implying that they tracked distance to anticipate the reward. However, the consistent running speed profiles across trials made it ambiguous whether animals were integrating over distance or time.

Extracellular recordings in CA1 using silicon probes revealed that a subset of pyramidal neurons fired at specific distances during the cue-constant segment, expressing internally generated firing fields (IGFs) (Fig. 1f right, h, and Supplementary Fig. 1e; 4.39 ± 0.58% neurons; see Methods). These neurons collectively formed an IGS that unfolded as the animal progressed through the cue-constant segment (Fig. 1g).

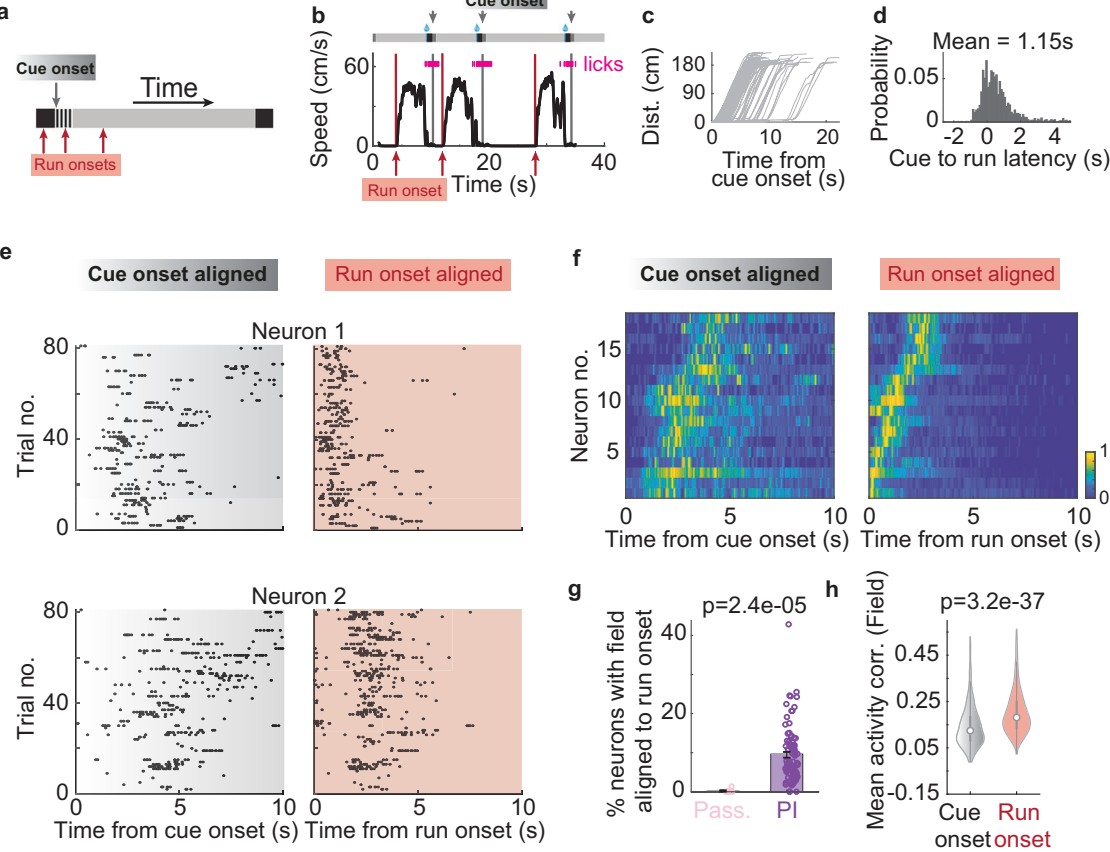

**Fig. 2 | Internally generated sequence (IGS) starts at run onset in the PI task.**
**a** The relationship between cue onset and run onset can be resolved when examining each trial in time instead of distance. Grey arrow: cue onset; red arrows: possible run onset times over multiple trials. **b** Example speed traces and licking behavior plotted against time unfolded across 3 consecutive trials, where run onset does not align with cue onset. Run onsets: red arrows. Cue onsets: grey arrows. **c** Distance traveled as a function of time from cue onset for all trials within one recording session. Each grey line corresponds to one trial. **d** Probability distribution of the time from cue onset to run onset (28 animals, 102 recordings). Spiking

activity of two example pyramidal neurons (**e**), and an example IGS (**f**) from a single session (the same session as in (**c**)), aligned with cue onset (left) vs. run onset (right). **g** The percentage of pyramidal neurons with IGFs per recording in the passive and PI tasks after aligning with run onset. **h** Mean activity correlation of neurons with IGFs when aligned with cue vs. run onset (650 neurons. The violin plot shows the median (white circle), with the box showing the interquartile range. Whiskers extend to the farthest data points that are not considered outliers. Mean ± SEM: Cue onset: 0.14 ± 0.0035, run onset: 0.20 ± 0.0037. Mean ± SEM and Wilcoxon rank-sum test are used if not explicitly stated).

To confirm that IGS expression requires distance or time integration, we compared the PI task with a control passive task. In the passive task, animals had to run for 180 cm while viewing the same gray screens, but a water reward was automatically delivered at the end without requiring licking (Fig. 1b right). To further discourage distance or time integration, we additionally introduced randomness in the start of the trial (see Methods). This passive task, which does not require animals to integrate over distance, replicates previously reported non-memory tasks[6,21].

Behaviorally, animals in the passive task ran at a mean speed similar to that in the PI task (Fig. 1c right and d; Passive: 34.41 ± 1.80 cm/s, PI: 34.73 ± 1.23 cm/s, $p = 0.63$; see Methods). However, there was a significant decrease in predictive licks before reward delivery (Fig. 1e; mean number of licks/cm from 100 to 180 cm: Passive: 0.042 ± 0.012, PI: 0.087 ± 0.011, $p = 0.012$).

Correspondingly, CA1 pyramidal neurons in the passive task exhibited significantly lower trial-by-trial activity correlation (Supplementary Fig. 1f; Passive: 0.006 ± 7.9e−04 (536 neurons), PI: 0.033 ± 5.2e−04 (7828 neurons), $p = 3.6e-94$). Neurons fired throughout the cue-constant segment (Fig. 1f middle), and the IGF was absent (Fig. 1h). These results collectively demonstrate that IGSs are preferentially expressed during tasks that require distance or time integration.

These findings thus validated the PI task as a suitable paradigm for investigating the internally generated hippocampal dynamics under conditions of distance or time integration.

## The initiation of an IGS aligns with the self-initiated onset of running

In the PI task, the animal's run onset, rather than the visual cue onset, likely served as the starting point of integration (Fig. 2a, b). There was a 1.15-second delay, on average, between the initial visual cue and run onset (Fig. 2a–d; see Methods), suggesting that animals likely self-initiated their running rather than merely responding to the cue.

Aligning neuronal activity with run onset in each trial revealed neurons that fired consistently at specific time points (IGFs). These IGFs were less evident when aligning the same activity with cue onset (Fig. 2e). Furthermore, run-onset alignment led to a greater proportion of neurons with IGFs (Fig. 2g; PI: 9.46 ± 0.72% neurons, compared to Fig. 1h). These IGF neurons also showed higher trial-by-trial activity correlation (Fig. 2h) and exhibited higher temporal information content compared to cue-onset alignment (cue onset: 0.29 ± 0.0077 bits/spike, run onset: 0.34 ± 0.0091 bits/spike, $p = 3.0e-05$). At the population level, IGSs became sharper with run-onset alignment (Fig. 2f; population correlation: cue onset: 0.20 ± 0.0086, run onset: 0.24 ± 0.0076, $p = 1.9e-04$). Finally, neurons with IGF exhibited more

pronounced phase precession[6,22,23] (Supplementary Fig. 2a, b), a hallmark of firing fields in CA1.

Beyond IGF neurons, we examined how run-onset alignment affects the broader activity patterns of all recorded CA1 pyramidal cells. Indeed, aligning activity to run onset enhanced trial-by-trial activity correlations throughout the CA1 population (Supplementary Fig. 2c, d). We then considered a potential confound, whether this effect was due to a switch in brain state (from non-theta to theta) as animals transitioned from stopping for reward to starting the next run. However, this is unlikely because the average pause time was brief, during which theta oscillations largely continued uninterrupted during this interval (pause mean time = 1.88 s, Supplementary Fig. 4a–c; see Methods).

Next, we investigated whether the lack of IGFs during the passive task was due to misalignment with the run onset or reflected a lack of integration. Even after run-onset alignment, IGFs remained nearly absent (Passive: $0.19 \pm 0.19\%$ of neurons; Fig. 2g), with only one IGF detected across all passive recordings (Supplementary Fig. 2g; see Methods). To rule out running speed as a confound, we further confirmed these results by analyzing PI task recordings with running speed profiles matched to those in the passive task (Supplementary Fig. 2h–j; see Methods). Consistent with the results above, pyramidal neurons in the passive task showed significantly lower trial-by-trial activity correlations compared to the PI task (Supplementary Fig. 2e, f). Together, these findings further support the requirement for distance or time integration for IGF generation.

To further address whether IGF neurons encode distance or time, we employed a Generalized Linear Model (GLM) approach[8] (see Methods). For each neuron, we modeled its firing activity using predictors including time and distance traveled since run onset, running speed, and recent spiking history. By estimating the unique contribution of each predictor, we found that a substantial proportion of neurons were significantly modulated by time (53.8%) and by distance (56.4%) (Supplementary Fig. 3a). Many individual neurons showed significant contributions from both factors (42.1% conjunctively tuned), while at the population level, there was no significant difference in the overall contribution of time compared to distance (Supplementary Fig. 3a).

In summary, run onset aligns the neuronal activity involved in path integration, likely marking the initiation of the neural computation for distance or time estimation.

## Accurate integration is correlated with the expression of IGSs

To investigate how IGS expression relates to task performance, we categorized each trial as either "good" or "bad" based on the animals' locomotion patterns and reward outcomes. Good trials met three criteria: animals came to a complete stop before trial run onset (thereby ensuring this onset marked the start of integration), maintained largely uninterrupted running, and successfully obtained the reward (good trials: $77.90 \pm 2.88\%$ of all trials, see Methods). In contrast, bad trials failed to meet at least one of these criteria. While bad trials maintained a mean running speed similar to that of good trials (Fig. 3a; good trials: $48.12 \pm 0.84$ cm/s, bad trials: $45.20 \pm 0.84$ cm/s), they resulted in a comparatively lower reward rate ($86.34 \pm 4.36\%$) than the 100% in good trials.

Examining anticipatory licking behavior revealed that, in bad trials, animals showed significantly more licking in the early part of running (Fig. 3b; mean licks/cm before 100 cm distance: good trials: $0.0056 \pm 0.0012$, bad trials: $0.013 \pm 0.0032$; mean licks/20 ms between 0 to 1 s time: good trials: $0.0081 \pm 0.0015$, bad trials: $0.042 \pm 0.005$, $p = 1.31e{-}10$). This behavior implied an inaccurate estimation of distance or time.

Correspondingly, neuron activity in bad trials showed clear impairments: IGF neurons exhibited reduced trial-by-trial activity correlation (Fig. 3d), and the IGSs were disrupted (Fig. 3c and

Supplementary Fig. 5a). We confirmed this impairment through several control analyses: It persisted after random subsampling of good trials to match the number of bad trials (Supplementary Fig. 5b; see Methods) and when IGSs were plotted against distance rather than time (Supplementary Fig. 5c, d). Furthermore, to exclude the possibility that the IGS impairment in bad trials was due to a lack of proper run onset, we reanalyzed good and bad trials, selecting only those where animals reached a complete stop before trial run onset (10 animals, 13 recordings, good trials: $68.17 \pm 4.19\%$ of all trials, rewarded bad trials: $79.54 \pm 7.24\%$). The new criteria led to similar IGS impairment (Supplementary Fig. 6a–c). Together, these findings indicate that the IGS expression is closely linked to accurate distance or time estimation, as reflected in both neuronal and behavioral measures.

## Subpopulations of pyramidal neurons display distinct responses around run onset

Given that only a small subset of CA1 pyramidal neurons exhibited IGF in the PI task, we aimed to investigate the dynamics of all pyramidal neurons as a function of time or distance from run onset. We sorted all recorded neurons (7114 neurons, 28 animals, including 650 IGF neurons) based on their change in firing rate around run onset. These changes were quantified as the firing rate ratio R (Fig. 3e and Supplementary Fig. 5e; $R = FR_{aft}/FR_{bef}$, where $FR_{bef}$ was calculated from $-1.5$ to $-0.5$ s and $FR_{aft}$ from 0.5 to 1.5 s relative to run onset).

This approach revealed two distinct functional subpopulations of pyramidal neurons. "PyrUp" neurons (37.2% of the neurons, $R > 3/2$) rapidly increased their activity at run onset and then gradually ramped down as the animal approached the reward zone (Fig. 3f left, g, h), whereas "PyrDown" neurons (19.4%, $R < 2/3$) initially reduced their activity but then ramped up toward the reward zone (Fig. 3f right, i, j). Neurons exhibiting intermediate ratios ($2/3 \leq R \leq 3/2$) were classified as "PyrOther" (Supplementary Fig. 7a).

To validate these functional subpopulations, we compared observed R values against a null distribution generated by circularly shifting individual trial firing rate profiles (1000 iterations). Neurons with R values beyond the 95th percentile or below the 5th percentile of this distribution were classified as PyrUp or PyrDown neurons, respectively. This approach confirmed the robustness of these subpopulations (Supplementary Fig. 7b–d; see Methods).

PyrUp and PyrDown neurons largely overlap in their laminar location within the CA1 pyramidal layer[24,25] (Supplementary Fig. 7e, f; see Methods), implying that each subpopulation likely comprises both genetically and functionally distinct deep and superficial CA1 neurons[24,26–29].

## PyrUp/PyrDown responses reflect internal integration, not solely locomotion

We next investigated whether PyrUp/PyrDown activity encoded internal computations related to time or distance integration, or if it merely reflected external motor variables such as running speed or licking.

First, we quantified the contribution of various factors to PyrUp/PyrDown activity using GLMs (see Methods)[8]. These analyses revealed that approximately half of both PyrUp (49.2%) and PyrDown (49.7%) neurons were significantly influenced by time, and a comparable proportion by distance (PyrUp: 50.5%; PyrDown: 49.5%, Fig. 4a, b). In contrast, the majority of PyrUp and PyrDown neurons were not significantly modulated by running speed (not modulated: PyrUp: 63.9%; PyrDown: 64.6%), licking (not modulated: PyrUp: 69.2%; PyrDown: 71.2%; Supplementary Fig. 8a, b), or acceleration (modulated: PyrUp: 4.2%; PyrDown: 3.8%; see Methods). Collectively, these GLM results suggest that PyrUp/PyrDown activity preferentially reflects internal integration processes rather than being simple correlates of motor outputs.

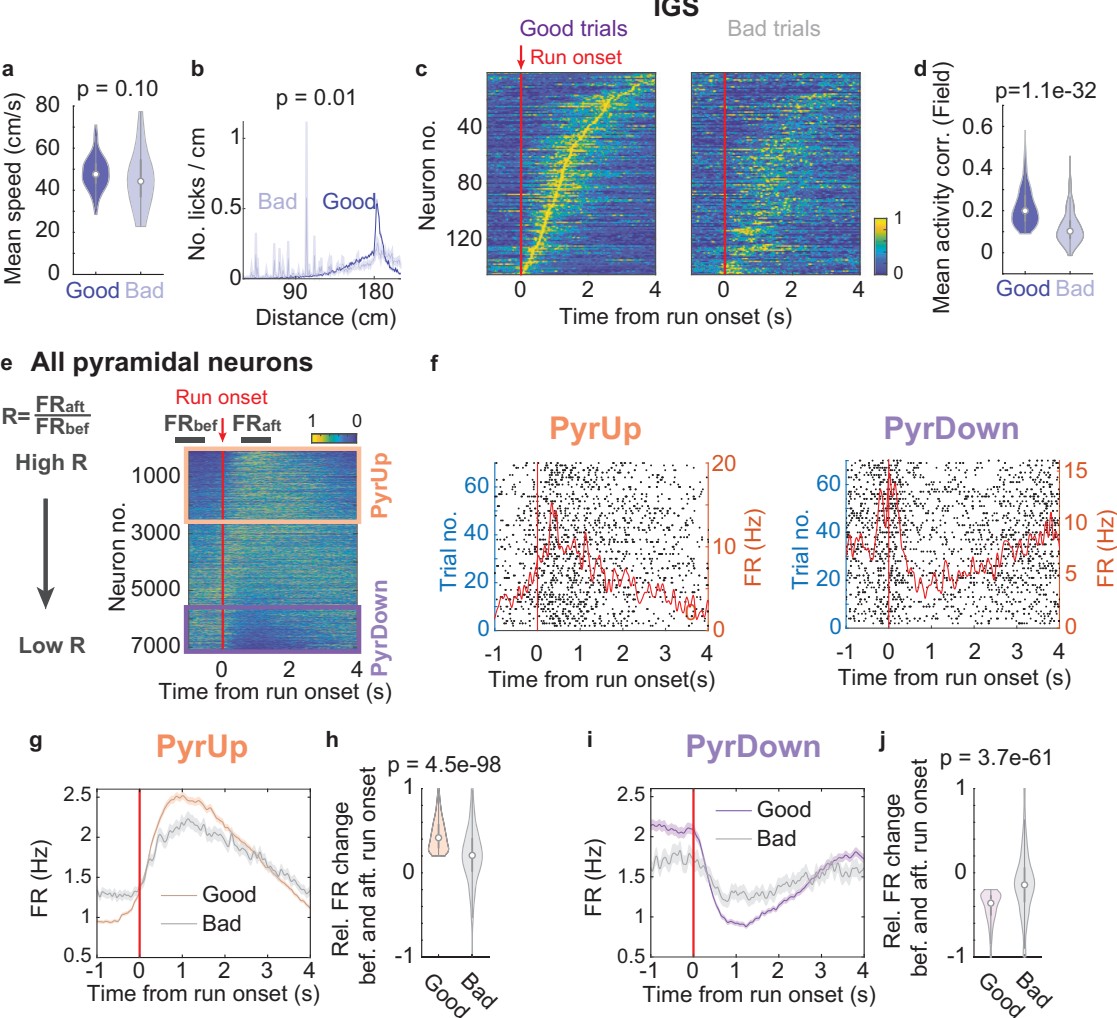

**Fig. 3 | CA1 pyramidal neurons display distinct responses around the start of integration.** Mean running speed (**a**), and lick histograms (reported *p*-value is for mean number of licks/cm between 30–100 cm) (**b**), broken down by good (blue) and bad trials (light blue). **c** IGS formed by neurons with IGFs from all the recordings with at least 15 bad trials (15 animals, 22 recordings), averaged across good (left) and bad (right) trials. **d** Mean trial-by-trial activity correlation of neurons with IGFs, broken down by good (blue) and bad trials (light blue). Good trials: 0.22 ± 0.0035, bad trials: 0.12 ± 0.0066. **e** After aligning with run onset, normalized firing rate heatmaps of all pyramidal neurons ordered by their firing rate ratio R = FR$_{aft}$/FR$_{bef}$. The black bars at the top indicate the time window for FR$_{bef}$ and FR$_{aft}$

calculations, with 0 s corresponding to run onset. Recordings with at least 15 good trials are included. PyrUp (2645 neurons) and PyrDown (1377 neurons) neurons are denoted by orange and purple boxes, respectively. **f** Spike rasters for example PyrUp (left) and PyrDown (right) neurons. The averaged firing rate over time is superimposed (red line). **g** The firing rate profile averaged across all PyrUp neurons, broken down into good and bad trials. The shaded area shows the SEM. **h** Relative firing rate change before and after run onset (FR$_{aft}$ − FR$_{bef}$)/(FR$_{bef}$ + FR$_{aft}$) for PyrUp neurons. Good trials: 0.45 ± 0.0036, bad trials: 0.20 ± 0.012. **i, j** Same as in (**g, h**), for all recorded PyrDown neurons. Good trials: −0.41 ± 0.0048, bad trials: −0.13 ± 0.020.

Second, we examined the temporal relationship between PyrUp/PyrDown firing rates and speed profiles. Both PyrUp and PyrDown neurons reached peak activity significantly earlier than the peak in running speed (Supplementary Fig. 9c–h; median±SEM: PyrUp: 1.24 ± 0.02 s; PyrDown: 1.21 ± 0.03 s; speed: 1.69 ± 0.08 s; p = 1.1e−7 for PyrUp vs. speed; p = 2.9e−8 for PyrDown vs. speed, Wilcoxon rank-sum test). This anticipatory firing before the speed peak suggests that PyrUp and PyrDown responses are not merely a direct readout of locomotion patterns. Moreover, the acceleration profile showed no clear relationship to PyrUp/PyrDown responses (Supplementary Fig. 9a, b).

Third, we investigated whether PyrUp/PyrDown activity directly tracks variations in locomotion speed. We categorized trials by high (>45 cm/s) and low (35–45 cm/s) average speeds (2199 PyrUp neurons, 1246 PyrDown neurons, 84 recordings, 41 animals; see Methods). Despite apparent differences in running speed, the firing rates of both PyrUp and PyrDown neurons remained largely stable (Fig. 4c–f; with PyrUp activity stability evident within the first three seconds).

To further reveal potential speed modulation at distinct time points, we characterized the time-resolved correlation between running speed and the instantaneous firing rate in each neuron. We observed only weak correlations, with a transient increase primarily during the first second after run onset (Supplementary Fig. 13a–d; see Methods). Yet, even this transient increase was significantly weaker than that observed in simultaneously recorded interneurons (Supplementary Fig. 13e), a population known for strong speed modulation[30,31]. These results support that PyrUp/PyrDown activity is only weakly and transiently modulated by locomotion speed.

Fourth, we evaluated whether PyrUp/PyrDown responses are context-dependent, specifically tied to the trial-start run onset (TRO) that marks the beginning of integration, rather than being generic responses to any run onset. We analyzed neuronal activity around spontaneous run onset (SRO), which occasionally occurred in the middle of a trial (17 animals, 32 sessions, 2902 neurons; see Methods).

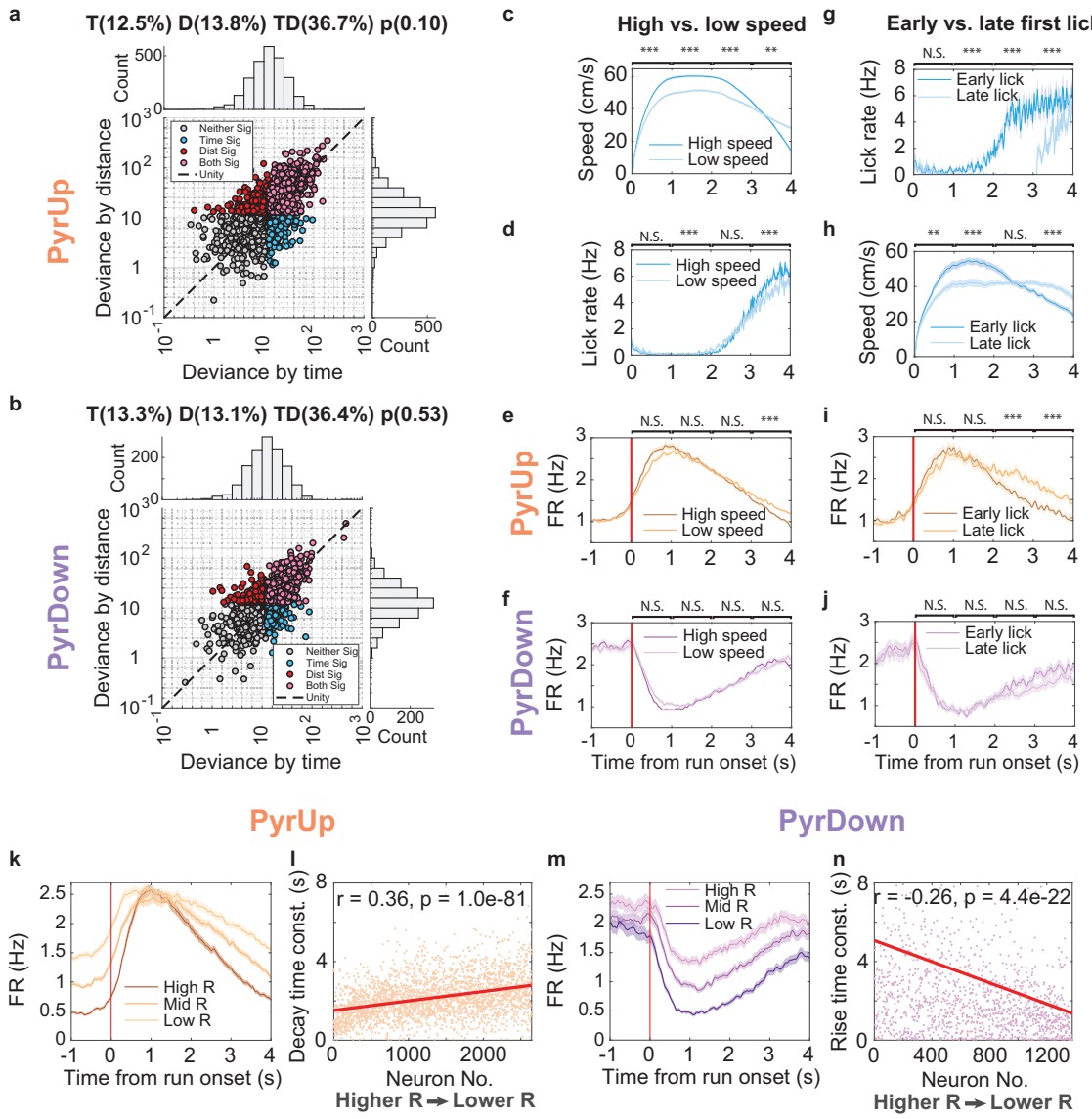

**Fig. 4 | PyrUp/PyrDown neurons encode time/distance through a two-phase mechanism. a** Time versus distance contribution to PyrUp activity using GLM analyses. X-values represent the deviance explained by time (see Methods). A larger x value indicates a more significant contribution from time. Similarly, y-values represent the deviance explained by distance. Each point represents a single neuron. Points colored in pink are significantly influenced by both time and distance. Points colored in blue are significantly influenced by just time. Points colored in red are significantly influenced by just distance. Points colored in grey are not significantly influenced by either time or distance. The top panel shows the histogram of deviance by time, and the right panel shows the histogram of deviance by distance. The title shows the percentage of neurons that are significantly influenced by just time (T), just distance (D), and both time and distance (TD). *p*-value is calculated using Wilcoxon rank-sum test between the deviance by time and by distance.

**b** Time versus distance contribution to PyrDown activity. **c**–**f** Top to bottom: **c** speed profiles, **d** licking profiles, **e** PyrUp firing rate profiles, and **f** PyrDown firing rate profiles, comparing trials with high and low speed. Only considering good trials. Statistics are calculated between two groups on the mean value within each second from 0 to 4 s, * *p* < 0.05, ** *p* < 0.01, *** *p* < 0.005, Wilcoxon rank-sum test. **g**–**j** Same as **c**–**f**, comparing trials with early and late first lick. Only considering good trials. Top to bottom: **g** licking profiles, **h** speed profiles, **i** PyrUp firing rate profiles, and **j** PyrDown firing rate profiles. **k** Average firing rate profiles among three equal-sized PyrUp neuron groups based on their R values: high R, mid R, and low R (dark to light lines). Average peak activity time for high R vs low R groups: *p* = 0.47. **l** R versus decay time constant for PyrUp neurons (orange dots; see Methods). The linear regression line (red) illustrates the change of decay time constants across neurons. **m**, **n** Same as **k**, **l**, for all the PyrDown neurons.

Behaviorally, animals did not exhibit a clear reward zone overshooting after a SRO (percent of rewarded trials: trials with SR: 92.98 ± 1.93%, trials without SR (TR): 96.17 ± 1.63%, *p* = 0.07), and lick patterns in trials with spontaneous runs remained unchanged (mean lick rates: 30–100 cm: SR: 0.011 ± 0.0026 licks/cm, TR: 0.011 ± 0.0027 licks/cm, *p* = 0.83; 100–140 cm: SR: 0.044 ± 0.0070 licks/cm, TR: 0.054 ± 0.0085 licks/cm, *p* = 0.48; 140–180 cm: SR: 0.13 ± 0.010 licks/cm, TR: 0.14 ± 0.013 licks/cm, *p* = 0.54). These results suggested that mid-trial stops did not reset the ongoing time or distance integration, and SRO was unlikely to mark a new integration start.

Correspondingly, both PyrUp/PyrDown responses were significantly attenuated at SRO compared to TRO, despite similar mean firing rates (Supplementary Fig. 10a, b; PyrUp: SRO: 1.86 ± 0.054 Hz, TRO: 1.85 ± 0.050 Hz, PyrDown: SRO: 1.77 ± 0.081 Hz, TRO: 1.60 ± 0.065 Hz). This attenuation persisted even after matching running speeds between SRO and TRO runs (Supplementary Fig. 10c–e; 13 animals, 21 sessions, 1731 neurons; see Methods) and after sub-sampling TRO runs to match the number of SRO (Supplementary Fig. 10d). Notably, these speed-matched SRO runs were not just brief movements; their duration was 2.81 ± 0.05 s (mean ± SEM), compared

to TRO runs of $3.83 \pm 0.04$ s. Thus, PyrUp/PyrDown responses are context-dependent and specifically elicit at the run onset that initiates time or distance integration.

Fifth, to demonstrate that PyrUp and PyrDown responses do not obligatorily depend on locomotion, we designed an immobile task. In this task, animals remained stationary[32] but were required to estimate a 4-second time period while viewing gray screens (Supplementary Fig. 11a, b; see Methods). Even in this immobile state, similar proportions of PyrUp and PyrDown neurons emerged (Supplementary Fig. 11d–k; 5 animals, 15 sessions, 748 neurons, PyrUp-Imm: $36.56 \pm 3.74\%$, PyrDown-Imm: $18.75 \pm 2.80\%$). Consistent with self-initiated time estimation, neuronal activity in this task aligned well with the last lick of the previous trial, rather than the visual cue (Supplementary Fig. 11d–j), and IGF neurons were also present (Supplementary Fig. 11c; $6.72 \pm 1.56\%$ neurons). Thus, PyrUp/PyrDown responses occur during tasks requiring time or distance integration across distinct behavioral states, including both locomotion and immobility.

Finally, to provide further evidence that PyrUp responses contain time information, we applied Bayesian decoding[33]. This analysis, performed on the single-trial population activity of PyrUp neurons without IGFs, effectively decoded time passage (Supplementary Fig. 12a–d; see Methods).

Collectively, these data supported that PyrUp/PyrDown activity encodes time or distance, rather than simply reflecting locomotion.

### PyrUp/PyrDown responses potentially predict reward-anticipatory behavior

We next probed whether PyrUp/PyrDown dynamics are correlated with task performance.

First, PyrUp/PyrDown responses around run onset were more pronounced in good trials compared to bad trials (Fig. 3g–j), despite similar overall firing rates (measured from −1 to 4 seconds, PyrUp: good trials: $1.76 \pm 0.032$ Hz, bad trials: $1.72 \pm 0.061$ Hz, $p = 0.19$, PyrDown: good trials: $1.49 \pm 0.042$ Hz, bad trials: $1.50 \pm 0.087$ Hz, $p = 0.27$). This result implies that the strength of their run onset responses correlates with the accuracy of time or distance estimation.

Second, we probed the correlation between PyrUp/PyrDown dynamics and the animal's internal estimate of reward timing. In the PI task, anticipatory licking serves as a behavioral proxy for the animal's expectation of reward, which is contingent on time or distance integration. We grouped trials based on the timing of the animal's first anticipatory lick (early lick: <2.5 s vs. late lick: >3.1 s, 492 PyrUp and 327 PyrDown neurons, 18 recordings, 15 animals; see Methods). Although speed profiles differed between these groups, PyrUp activity remained stable in the first 2 seconds, and PyrDown neurons showed no significant changes overall (Fig. 4g–j). However, the PyrUp decay time constant increased significantly in late lick trials (PyrUp: early lick: $1.89 \pm 0.05$ s vs. late lick: $2.00 \pm 0.05$ s, $p = 0.036$; Fig. 4i). The PyrDown rise time constant showed a similar, but not statistically significant, trend (PyrDown: early lick: $2.34 \pm 0.22$ s; late lick: $3.02 \pm 0.26$ s; $p = 0.27$; Fig. 4j). These changes in the decay/rise rate indicated that the PyrUp/PyrDown dynamics adapt to the animal's internal estimate of when sufficient time or distance has elapsed. This consistency between neuronal dynamics and behavior suggests that the decay/rise rates of PyrUp/PyrDown responses may predict reward-anticipatory behavior.

### PyrUp and PyrDown neurons employ a two-phase coding mechanism

Having established that PyrUp and PyrDown neurons encode time or distance and are not merely locomotor correlates, we next characterized their coding strategy. Unlike the sequential activation of IGF neurons tuned to specific time points, PyrUp neurons exhibited a collective peak at similar times shortly after run onset (Fig. 4k). Their

activity then decayed at heterogeneous, neuron-specific rates. This observation led us to propose a two-phase integration mechanism.

In this two-phase integration mechanism, the first phase is characterized by a rapid, synchronized increase in activity across the PyrUp population. The stability of PyrUp activity during this period across early and late lick trials, its specificity to the trial start run onset, and its increased speed modulation together suggest that this phase primarily functions as a synchronized signal that marks the initiation of the trial's integration process.

Following this initial phase, the second phase is characterized by a slower, heterogeneous decay of activity within the PyrUp population. We found that the decay rates of PyrUp neurons were not random; neurons with faster decay rates tended to exhibit higher activity changes around run onset (higher R values, Fig. 4k, i) and had lower baseline firing before run onset (correlation between $FR_{bef}$ and decay time constant: $r = 0.26$, $p = 1.18e\text{-}41$). A similar coding mechanism was observed in PyrDown neurons: those with faster rise rates leading to the reward tended to have lower R values (Fig. 4m, n).

To test the functional relevance of these two-phase dynamics, we analyzed neuronal activity during the passive task that does not require time/distance integration. The results provided several lines of evidence supporting our proposed model. First, there was a differential change in the proportion of PyrUp and PyrDown neurons (Supplementary Fig. 14a, b). While the percentage of PyrUp neurons remained similar between tasks (160/423 neurons, 37.8% in passive vs. 37.2% in PI), the percentage of PyrDown neurons was significantly reduced in the passive task (47/423, 11.1% in passive vs. 19.4% in PI; Supplementary Fig. 14c, d). This change aligns with the significant reduction in predictive licking observed in the passive task.

In contrast, although the proportion of PyrUp neurons was similar, their dynamics were significantly altered in the passive task, particularly in the second phase. PyrUp neurons exhibited significantly longer decay time constants (Supplementary Fig. 14g), and their decay rates were no longer significantly correlated with changes in firing responses at run onset (Supplementary Fig. 14e, f), thereby losing the heterogeneity observed in the PI task. This difference in decay time constant persisted when we analyzed PI task recordings with matching speed to the passive task (Supplementary Fig. 2h, i for speed and licking profiles after speed matching; R vs. decay time constant: correlation $r = 0.45$, $p = 3.4e\text{-}24$ (t-statistics); Tau: PI: $2.94 \pm 0.06$ s, Passive: $3.32 \pm 0.11$ s, $p = 1.9e\text{-}03$). This result further supports that the observed PyrUp dynamics in the PI task do not merely reflect locomotion. Conversely, the rise time constants of the small number of PyrDown neurons remained largely intact and were significantly correlated with their response changes around run onset, similar to the PI task (R vs. decay time constant: correlation $r = -0.62$, $p = 3.0e\text{-}06$; Tau: PI: $2.30 \pm 0.07$ s, Passive: $3.40 \pm 0.59$ s, $p = 0.26$). In addition, PyrUp neurons showed a significant delay in peak time in the passive task (Supplementary Fig. 14h), while PyrDown neurons' trough time remained similar (Peak time: PI: $1.49 \pm 0.05$ s, Passive: $1.67 \pm 0.10$ s, $p = 0.14$). The loss of heterogeneous ramping and the delayed peak time in the absence of an integration demand provide evidence that PyrUp dynamics are a feature of the time/distance integration process. The differential changes in PyrUp/PyrDown dynamics during the passive task suggest that these subpopulations may play distinct functional roles during time/distance integration.

In summary, PyrUp and PyrDown neurons exhibit a synchronized start response followed by activity increases or decreases with heterogeneous, neuron-specific rates. This synchronized "start" may establish a reference point for subsequent time/distance encoding. This coding mechanism is distinct from the sequential, point-by-point encoding characteristic of neurons with IGFs.

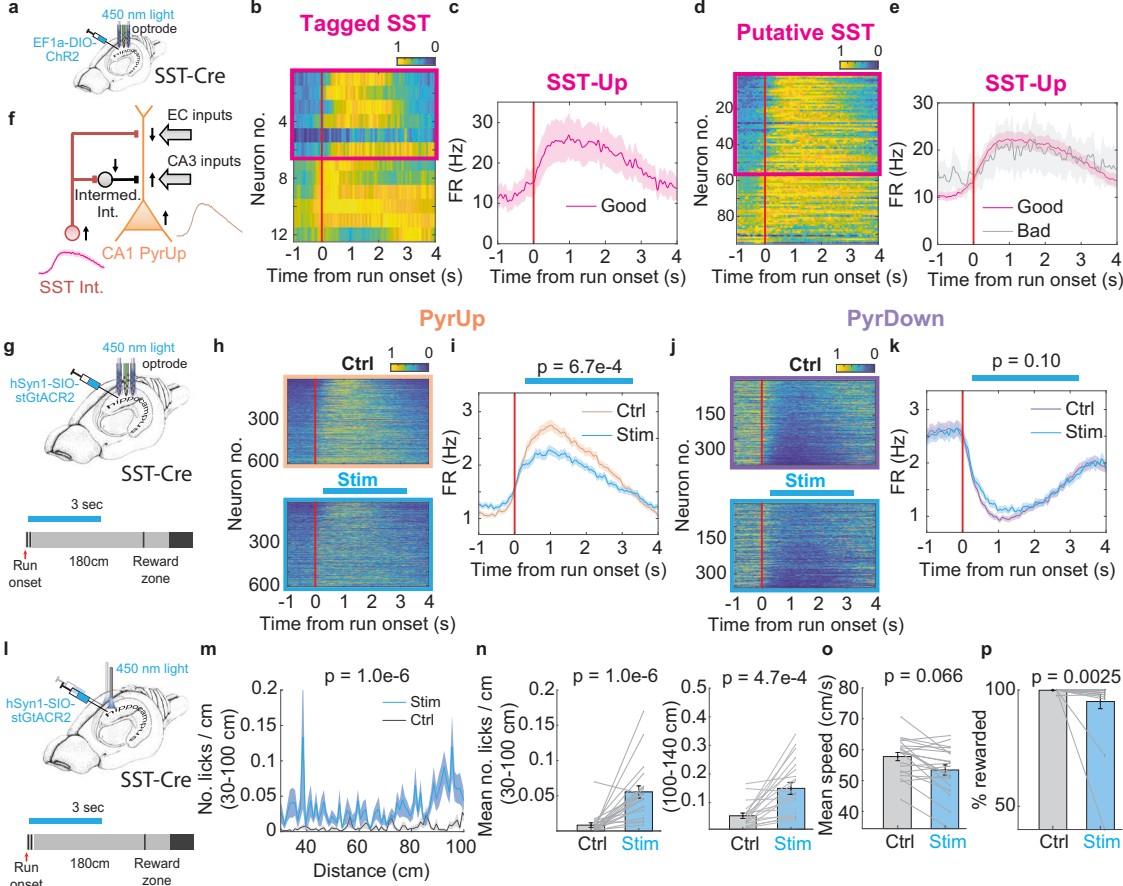

**Fig. 5 | CA1 somatostatin-positive interneurons regulate PyrUp response and time/distance integration. a** Schematic for the opto-tagging experiment[7]. **b** Normalized firing rate heatmaps of interneurons identified as SST-positive from the tagging experiments and at the same time belonging to the putative SST interneuron cluster. Neurons are ordered by their firing rate ratio R around run onset. The magenta box denotes SST-Up neurons with R > 3/2 (6/12 neurons). **c** Firing rate profile averaged across tagged SST-Up neurons highlighted within the magenta box in (b). Bad trials are not shown because the number of bad trials is less than 15 trials in most of these recordings. **d** Same as in (**b**), but for all the interneurons belonging to the putative SST interneuron cluster. The magenta box denotes SST-Up neurons with R > 3/2. **e** Firing rate profile averaged across all SST-Up neurons, broken down into good and bad trials. Relative FR change: good trials: 0.35 ± 0.015, bad trials: 0.076 ± 0.054, p = 6.2e−06. **f** A circuit diagram illustrating the proposed mechanism by which SST interneurons regulate PyrUp response. SST interneurons disinhibit the CA3 inputs to PyrUp neurons by inhibiting intermediate interneurons. **g** Experimental setup for unilaterally inactivating SST-interneurons[7]. Light stimulation is applied with an optrode for 3 s starting after run onset in a

subset of trials (see Methods). **h** Normalized firing rate heatmaps of all PyrUp neurons during control (top) and stimulation trials (bottom). Neurons are ordered by their firing rate ratio around run onset. The blue horizontal bar illustrates the estimated stimulation window. **i** Firing rate profiles averaged across PyrUp neurons for control (orange) and stimulation trials (blue). Mean FR between 0.5–2 s: control: 2.55 ± 0.086 Hz, stimulation: 2.15 ± 0.077 Hz, p = 6.7e−04, two-sample t-test. **j** Same as in (**h**), for PyrDown neurons. **k** Same as in **i**, for PyrDown neurons. Mean FR between 0.5–2 s: control: 1.05 ± 0.056 Hz, stimulation: 1.19 ± 0.065 Hz, p = 0.10, two-sample t-test. **l** Experimental setup to assess the behavioral effects of SST inactivation[7]. The same stimulation protocol as in (**g**) was applied to a separate cohort of mice with bilaterally implanted optic fibers. **m** Lick histograms within the first 100 cm of running for control (grey) and stimulation (blue) trials (6 animals, 21 recordings). **n** Mean number of licks/cm calculated based on (**m**) (**n** left) (control: 0.0082 ± 0.0033, stimulation: 0.055 ± 0.0086), mean number of licks/cm between 100–140 cm (n right) (control: 0.054 ± 0.0094, stimulation: 0.15 ± 0.021). **o** Mean running speed (control: 57.77 ± 1.32 cm/s, stimulation: 53.42 ± 1.65 cm/s). **p** Percentage of rewarded trials (control: 99.88 ± 0.12%, stimulation: 95.04 ± 3.05%).

## Somatostatin-positive interneurons display slow ramping activity around run onset

To reveal the circuit-level mechanisms underlying PyrUp/PyrDown responses, we first examined local somatostatin-positive (SST) inhibitory interneurons. These interneurons coordinate interactions between CA3 and the entorhinal cortex inputs to CA1 pyramidal neurons, primarily by targeting their dendrites[13,18,19].

To identify SST interneurons, we used optogenetic tagging by transfecting the CA1 SST interneurons with channelrhodopsin-2 (ChR2) in SST-IRES-Cre animals (Fig. 5a and Supplementary Fig. 15b; 3 animals). Meanwhile, we classified putative inhibitory neurons[15] by their spike properties and relative locations to the pyramidal layer center[13] (Supplementary Fig. 15a for clustering overview and Supplementary Fig. 15c for spike properties of an example putative SST interneuron; see Methods). We then identified a cluster that best

overlapped with the tagged neurons (12/24 tagged neurons, see Methods), exhibiting characteristics consistent with SST-expressing oriens-lacunosum moleculare (OLM) cells[13] (Supplementary Fig. 15d, e).

Within this identified cluster, a majority of neurons (55/95 neurons, Fig. 5d, e), including half of the tagged cells (6/12 neurons, Fig. 5b, c), showed an increase in activity at run onset that subsequently decayed as the reward approached in the PI task. We termed these neurons "SST-Up neurons". In bad trials, the SST-Up response at run onset was weakened, despite no change in the mean firing rate (Fig. 5e; mean FR: good trials: 17.52 ± 1.42 Hz, bad trials: 17.84 ± 4.19 Hz, p = 0.95), suggesting a correlation with the accurate estimation of time or distance.

Previous evidence suggests that OLM interneurons directly inhibit entorhinal cortex inputs to CA1 pyramidal neurons, while

simultaneously providing indirect disinhibition to their CA3 inputs through other inhibitory interneurons[14,16,17] (Fig. 5f). We thus hypothesize that the period during which SST-Up neurons displayed elevated firing after run onset likely favors the influence of CA3 inputs.

### Inactivating SST interneurons after run onset impairs PyrUp response, IGSs, and task performance

To determine how SST activity influences PyrUp/PyrDown responses, we utilized soma-targeted anion-conducting channelrhodopsins (stGtACR2) to hyperpolarize SST interneurons[34]. We recorded the neuronal activity unilaterally from dorsal CA1 using an optrode (Fig. 5g top) while performing closed-loop inactivation of SST interneurons for 3 seconds immediately after run onset (Fig. 5g bottom; see Methods). This manipulation effectively silenced some putative SST cells (Supplementary Fig. 15f, g).

Silencing SST interneurons significantly reduced the PyrUp response after run onset (Fig. 5h, i; 6 animals, 24 recordings, 608/1721 neurons), but did not significantly alter the PyrDown response (Fig. 5j, k; 359/1721 neurons).

Furthermore, IGF neurons showed a significant decrease in their activity correlation with control trials without a change in overall firing rate (Supplementary Fig. 15h, i; 160/1721 neurons; mean activity corr.: control: $0.22 \pm 0.0073$, stimulation: $0.19 \pm 0.0071$, p = 0.0029; mean FR: control: $1.99 \pm 0.11$ Hz, stimulation: $1.97 \pm 0.12$ Hz, $p = 0.44$). These inactivation-induced impairments on the PyrUp response and IGSs resembled the results in bad trials (Fig. 3c, g).

To assess behavioral consequences, we conducted a separate experiment in which we silenced SST interneurons bilaterally for 3 seconds starting at run onset, in a different cohort of animals (Fig. 5l; see Methods). In the stimulation sessions, animals exhibited significant impairment in task performance. Specifically, they showed a significant increase in early licks before 100 cm (Fig. 5m, n left; 6 animals), an increase in predictive licks between 100-140 cm (Fig. 5n right), and a decrease in rewarded trials (Fig. 5p). However, the mean running speed did not change significantly (Fig. 5o), and the speed profile showed only a slight, brief reduction between 2-3 seconds (Supplementary Fig. 15j–l), indicating largely intact locomotion. Control experiments confirmed that these effects were not due to laser light alone (Supplementary Fig. 18f–j, p–t; see Methods).

Overall, these findings demonstrate that elevated SST activity after run onset is necessary for producing the PyrUp response and IGSs (Fig. 5f). Reducing SST activity impairs the PyrUp response and IGSs, thereby disrupting time or distance integration.

### Inactivating SST interneurons close to reward does not affect PyrUp response or task performance

As SST activity declined near the reward, it is plausible that CA3 inputs to PyrUp neurons are gradually blocked (Fig. 5f). We thus hypothesize that SST interneurons may establish a critical time window, spanning from run onset to near the reward, that facilitates the PyrUp response. Supporting this hypothesis, silencing SST interneurons for 3 seconds starting at 120 cm (Fig. 6a) had no significant effect on PyrUp neurons or PyrDown neurons (Fig. 6b–e; 6 animals, 18 recordings), and only a minor impact on IGSs (Supplementary Fig. 16a–c; 161/1516 neurons; mean activity corr.: control: $0.23 \pm 0.0074$, stimulation: $0.21 \pm 0.0074$, $p = 0.021$; mean FR: control: $1.71 \pm 0.088$ Hz, stimulation: $1.71 \pm 0.099$ Hz, $p = 0.52$).

Similarly, when we bilaterally inactivated SST interneurons after 120 cm, using the same cohort of animals as in Fig. 5l, the animals' behavior did not alter (Fig. 6f–j). Therefore, SST interneurons are necessary for accurate time or distance integration, primarily within the time window from run onset to when animals approach the reward zone.

To reconcile our finding that SST inactivation after run onset impaired IGSs with previous studies that reported no clear disruption

in place field sequence in cue-rich environments[19], we introduced visual cues during the cue-constant segment of the PI task (Fig. 6k). In this cue-rich context, animals maintained similar running speed and licking profiles (Supplementary Fig. 16g, h), and PyrUp/PyrDown responses persisted (Fig. 6l–o), suggesting that time or distance integration remains a default strategy, supplemented by external cue guidance. We then repeated SST inactivation after run onset. Despite a shutdown of some putative SST cells, we observed no significant changes in neuronal sequences or the PyrUp response (Fig. 6l–o and Supplementary Fig. 16d–f; 3 animals, 9 recordings; mean activity corr.: control: $0.24 \pm 0.015$, stimulation: $0.20 \pm 0.014$, $p = 0.062$). Our results indicate that SST activity is required to sustain PyrUp responses driven by internal time or distance estimation, but not those evoked by external cues.

### Parvalbumin-positive cells display ramping activity around run onset

Because SST inactivation did not significantly affect the PyrDown neurons, we asked whether parvalbumin-positive (PV) interneurons, which primarily target the perisomatic region and regulate the spike timing and synchrony of pyramidal neurons[35–37], may play a role in controlling the shutdown of PyrDown neurons at run onset.

To identify PV cells, we used optogenetic tagging by transfecting CA1 PV interneurons with ChR2 in PV-Cre animals (Fig. 7a and Supplementary 17a; 7 animals). A large proportion of these tagged neurons overlapped with one putative inhibitory neuron cluster located primarily in the pyramidal layer (13/30 tagged neurons; Supplementary Fig. 15a and 17b (spike properties of an example putative PV interneuron)). Neurons within this cluster exhibited characteristics consistent with PV basket cells (Supplementary Fig. 17c, d)[13].

Within this identified cluster, a majority of neurons (75/114 neurons, Fig. 7d, e), including most of the tagged cells (10/13 neurons, Fig. 7b, c), exhibited a rapid increase in activity around run onset. We termed these "PV-Up neurons". In bad trials, the PV-Up response was weakened without a change in mean firing rate (Fig. 7e; mean FR: good trials: $20.16 \pm 1.29$ Hz, bad trials: $18.27 \pm 2.15$ Hz, $p = 0.47$).

We further investigated the timing of the PV-Up response relative to PyrUp/PyrDown responses. Both PV-Up and PyrUp neurons showed concurrent increases in normalized firing rates around run onset, with PV-Up neurons initiating their activity earlier than PyrUp neurons (Supplementary Fig. 17e left; time to reach 90% of maximum firing rate within $[-1\ 1.5]$s window ($t_{90\%}$): PyrUp: $0.77 \pm 0.01$ s, PV-Up: $0.50 \pm 0.04$ s, $p = 7.53$e-09). In contrast, the PyrDown response started to decrease only after the initial rise of PV-Up response and reached its minimum after the peak of PV-Up response (Supplementary Fig. 17e right; $t_{90\%}$: PyrDown: $0.60 \pm 0.01$ s, $p = 6.62$e-04 compared with PV-Up). These results imply that PV-Up neurons may preferentially inhibit PyrDown neurons.

### Parvalbumin-positive interneurons are necessary to start time/distance counting at run onset

To determine if PV interneurons predominantly evoke inhibition on PyrDown neurons, we used stGtACR2 to silence CA1 PV interneurons immediately after run onset for 3 seconds (Fig. 7g; see Methods)[34]. This manipulation effectively silenced some putative PV cells (Supplementary Fig. 17f, g).

Silencing PV interneurons caused a significant elevation in the PyrDown response after run onset (Fig. 7j, k; 3 animals, 6 recordings, 108/528 neurons), with an effect comparable to what was observed in bad trials (Fig. 3i). Yet, neither the PyrUp response (Fig. 7h, i; 205/528 neurons) nor IGSs (Supplementary Fig. 17h, i; 39/528 neurons; mean activity corr.: control: $0.27 \pm 0.019$, stimulation: $0.28 \pm 0.019$, $p = 0.79$; mean FR: control: $1.96 \pm 0.18$ Hz, stimulation: $2.23 \pm 0.23$ Hz, $p = 0.33$) showed significant changes.

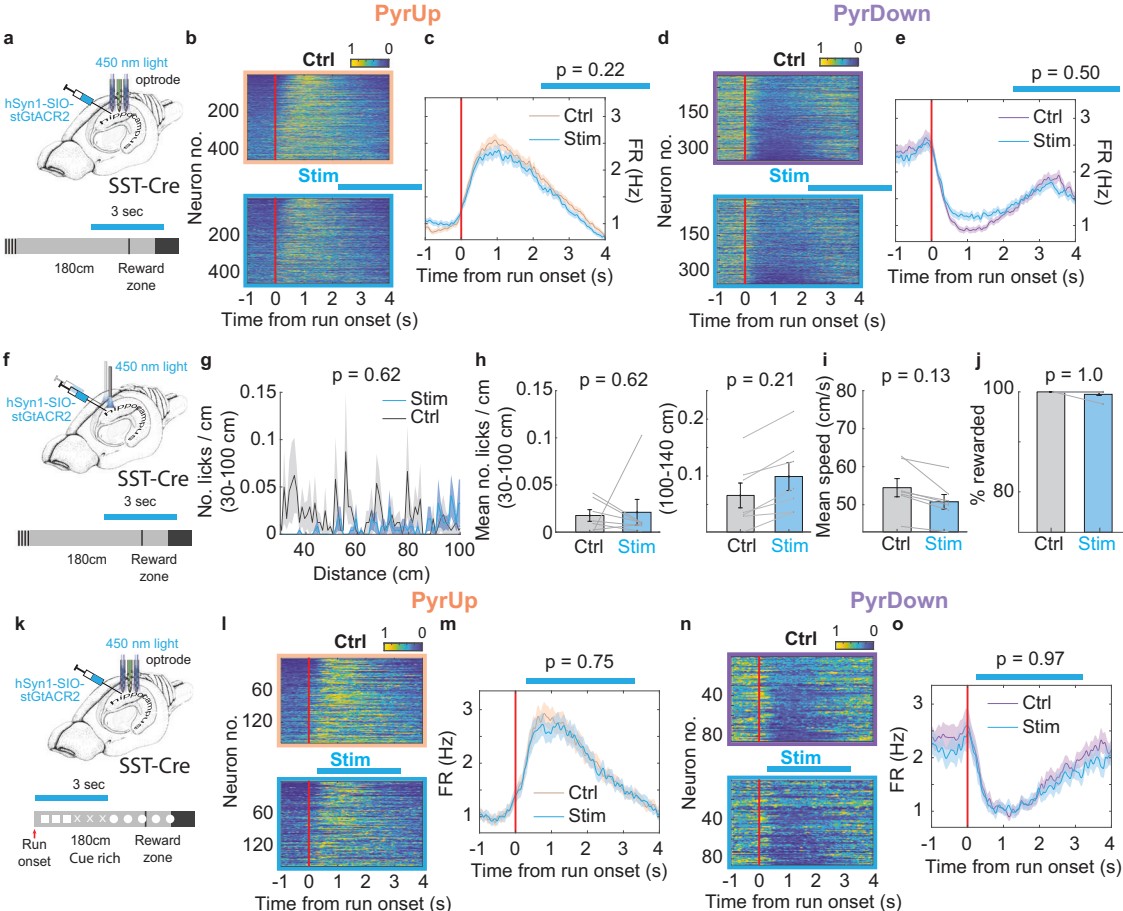

**Fig. 6 | Inactivating CA1 somatostatin-positive interneurons close to the reward zone and in a cue-rich environment has no effect. a** Experimental setup for unilaterally inactivating SST interneurons[7]. Light stimulation is applied with an optrode for 3 s, starting at 120 cm in a subset of trials. **b** Normalized firing rate heatmaps of all PyrUp neurons during control (top) and stimulation trials (bottom). PyrUp: 454/1516 neurons. **c** Firing rate profiles averaged across all PyrUp neurons for control (orange) and stimulation trials (blue). Mean FR between 2.5-4 s: control: $1.22 \pm 0.058$ Hz, stimulation: $1.13 \pm 0.055$ Hz, $p = 0.23$, two-sample t-test. The blue horizontal bar illustrates the estimated stimulation window. **d**, **e** Same as **b**, **c**, for PyrDown: 348/1516 neurons; mean FR between 2.5-4 s: control: $1.72 \pm 0.088$ Hz, stimulation: $1.63 \pm 0.082$ Hz, $p = 0.50$, two-sample t-test. **f** Experimental setup to assess the behavioral effects of SST inactivation at 120 cm[7].

**g** Lick histograms within the first 100 cm of running, including 4 animals, 7 recordings. **h** Mean number of licks/cm calculated based on **g** (**h** left), and mean number of licks/cm between 100–140 cm (**h** right). **i** Mean running speed. **j** Percentage of rewarded trials. **k** Experimental setup for unilaterally inactivating SST interneurons in a cue-rich environment[7]. Visual cues are displayed on the screens (square, X, and circle, each occupying 60 cm of the 180 cm track). Light stimulation is applied with an optrode for 3 s, starting after run onset in a subset of trials. **l**–**o** Same as **b**–**e**, for inactivating SST interneurons in a cue-rich environment, as shown in (**k**). PyrUp: 158/422 neurons; mean FR between 0.5–2 s: control: $2.63 \pm 0.18$ Hz, stimulation: $2.55 \pm 0.18$ Hz, $p = 0.75$, two-sample t-test; PyrDown: 86/422 neurons; mean FR between 0.5-2 s: control: $1.10 \pm 0.093$ Hz, stimulation: $1.09 \pm 0.11$ Hz, $p = 0.97$, two-sample t-test.

To assess behavioral consequences, we silenced PV interneurons bilaterally at run onset in a separate cohort of animals (Fig. 7l). In the stimulation sessions, animals exhibited an impairment mostly confined to the period shortly after run onset. Specifically, there was a significant increase in early licks within 100 cm (Fig. 7m, n left), but no changes beyond 100 cm (Fig. 7n right). The percentage of rewarded trials (Fig. 7p) and mean running speed (Fig. 7o) did not change. Control experiments confirmed that the effects were not due to laser light alone (Supplementary Fig. 18a–e, k–o; see Methods).

The timing of the behavioral impairment, which occurs shortly after run onset, implies that the animals cannot accurately initiate integration. Thus, we posit that PV interneurons are necessary for shutting down PyrDown neurons at run onset (Fig. 7f) and initiating integration over time or distance at the beginning of each trial.

## Discussion

In this study, we identified and characterized two distinct functional subpopulations of CA1 pyramidal neurons, PyrUp and PyrDown neurons, and proposed a two-phase mechanism for encoding elapsed time or traveled distance in both subpopulations. In this mechanism, PyrUp neurons first exhibit a synchronized increase in activity at the onset of integration (Phase I: initiation), which is followed by individual neurons ramping down at varying, neuron-specific rates that encode the passage of time or distance (Phase II: encoding). Conversely, PyrDown neurons initially decrease their activity before gradually ramping up toward the reward, following a similar two-phase strategy. Both PyrUp and PyrDown neurons may use their Phase I response as a reference point for subsequent time or distance encoding.

We further elucidated the circuit-level mechanisms, identifying two parallel CA1 inhibitory circuits that distinctly regulate individual subpopulations and influence integration accuracy. Somatostatin-positive interneurons preferentially modulated the PyrUp response and IGSs and were necessary for accurate time or distance integration. In contrast, parvalbumin-positive interneurons primarily regulated the activity of PyrDown neurons and were necessary for correctly initiating the integration process.

These findings support the hypothesis that SST interneurons define an "integration window", facilitating the PyrUp response and

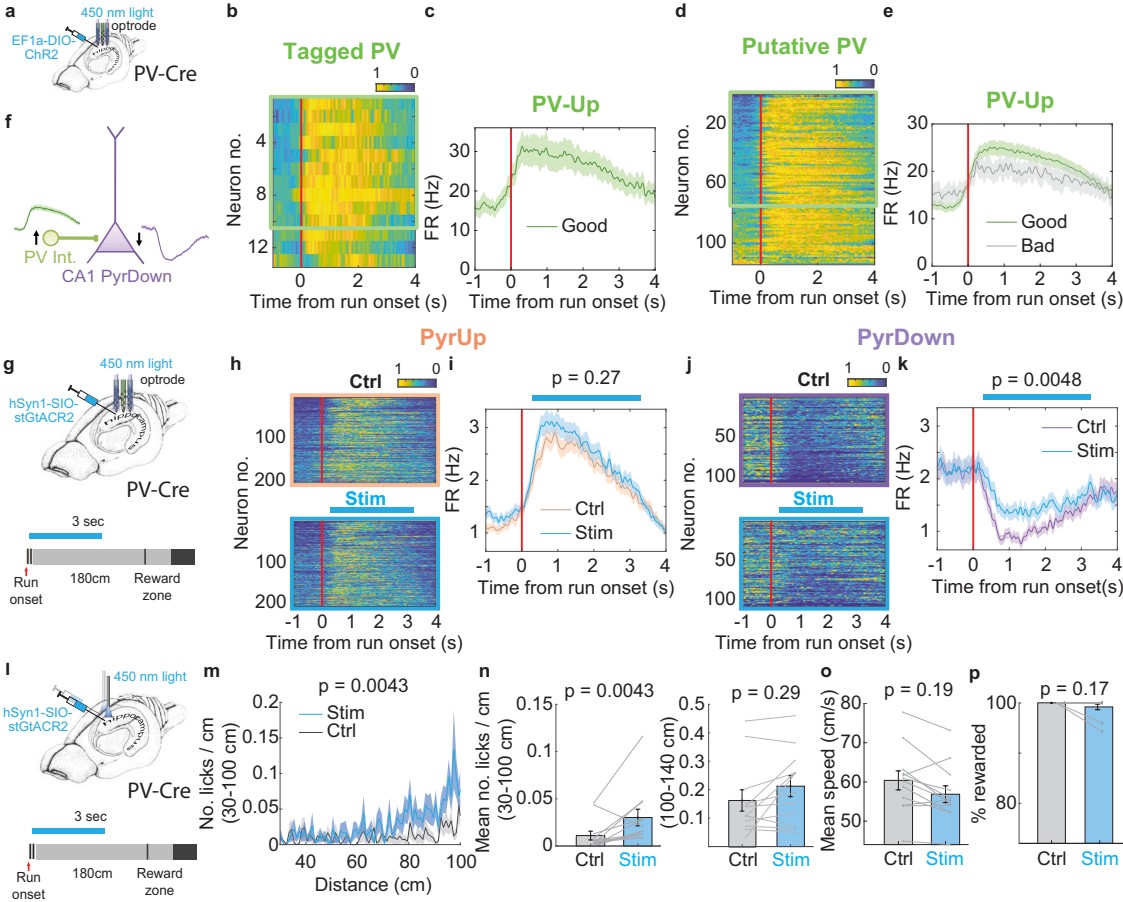

**Fig. 7 | CA1 parvalbumin-positive interneurons regulate PyrDown response and the start of integration. a** Schematic for the opto-tagging experiment[7]. **b** Normalized firing rate heatmaps of interneurons identified as PV-positive in the tagging experiments and at the same time belonging to the putative PV interneuron cluster. Neurons are ordered by their firing rate ratio R around run onset. The green box denotes PV-Up neurons with R > 3/2 (10/13 neurons). **c** Firing rate profile averaged across tagged PV-Up neurons highlighted within the green box in (**b**). Bad trials are not shown because the number of bad trials is less than 15 trials in most of these recordings. **d** Same as in **b**, but for all the interneurons belonging to the putative PV interneuron cluster. The green box denotes PV-Up neurons whose R > 3/2. **e** Firing rate profile averaged across all PV-Up neurons, broken down into good and bad trials. Relative FR change: good trials: 0.33 ± 0.012, bad trials: 0.16 ± 0.021, $p = 1.02e-8$. **f** A circuit diagram illustrating the proposed mechanism by which PV inactivation impacts PyrDown response. **g** Experimental setup for unilaterally inactivating PV interneurons[7]. Light stimulation is applied with an optrode for 3 s, starting after run onset in a subset of trials. **h** Normalized firing rate heatmaps of all PyrUp neurons during control (top) and stimulation trials (bottom). **i** Firing rate profiles averaged across PyrUp neurons. Mean FR between 0.5-2 s: control: 2.64 ± 0.16 Hz, stimulation: 2.90 ± 0.17 Hz, $p = 0.27$, two-sample t-test. **j, k** Same as in (**h, i**), for PyrDown neurons. Mean FR between 0.5-2 s: control: 0.97 ± 0.091 Hz, stimulation: 1.43 ± 0.13 Hz, $p = 0.0048$, two-sample t-test. **l** Experimental setup to assess the behavioral effects of PV inactivation at run onset[7]. **m** Lick histograms within the first 100 cm of running, including 4 animals, 12 recordings. **n** Mean number of licks/cm calculated based on (**m**) (**n** left) (control: 0.011 ± 0.0046, stimulation: 0.030 ± 0.0087), and mean number of licks/cm between 100−140 cm (n right) (control: 0.16 ± 0.037, stimulation: 0.21 ± 0.037). **o** Mean running speed (control: 60.40 ± 2.42 cm/s, stimulation: 56.87 ± 2.14 cm/s). **p** Percentage of rewarded trials (control: 100.00 ± 0.00%, stimulation: 99.18 ± 0.56%).

IGSs during ongoing integration; in contrast, PV interneurons generate a "reset" signal, shutting down PyrDown neurons and reinitiating integration. The coordinated interaction between these interneuron types and their corresponding pyramidal subpopulations orchestrates different aspects of the time or distance integration.

## PyrUp/PyrDown responses provide a coding mechanism for time or distance

Ramping activity, characterized by a gradual increase or decrease in firing rate over seconds to minutes, has been reported in various brain regions involved in spatial and temporal processing, including the entorhinal cortex[38–40], pre- and para-subiculum[39], retrosplenial cortex[41], prefrontal cortex[42,43], and striatum[44]. While a recent study provided descriptive examples of ramping activity in the rat hippocampus[45], a systematic characterization of this phenomenon has been lacking. Our study expanded on these observations by identifying and quantitatively characterizing two functionally distinct populations

of CA1 pyramidal neurons with opposing ramping profiles—PyrUp and PyrDown—during a virtual navigation task requiring path integration.

Several lines of evidence suggest that PyrUp and PyrDown neurons represent distinct functional subpopulations. First, they were differentially modulated by local inhibitory circuits. Optogenetic manipulation of SST and PV interneurons preferentially affected the PyrUp and PyrDown populations, respectively, and resulted in distinct behavioral impairments (Figs. 5 and 7). Second, the two populations were independently modulated by task demands. In a task that does not require path integration, the proportion of PyrDown neurons was significantly lower while the PyrUp population persisted, albeit with altered dynamics (Supplementary Fig. 14).

Furthermore, our data are consistent with a two-phase model that explains how these populations may contribute to path integration. The initiation phase (Phase I) is tightly linked to the start of time/distance integration. During this phase, we observed a transient increase in neurons' correlation with running speed (Supplementary

Fig. 13a–d). Yet, despite differences in the animal's internal estimate of time/distance (reflected in early vs. late anticipatory licking), the neurons' firing rates remained stable (Fig. 4g–j). This result suggests that Phase I establishes a consistent starting signal, irrespective of variations in the animal's internal estimates of time or distance.

In contrast, the encoding phase (Phase II) involves subsequent ramping dynamics that are correlated with the animal's behavioral report of its estimated distance. For example, the decay time constant of PyrUp neurons was longer on trials where the animal initiated anticipatory licking later (Fig. 4g–j). The heterogeneity of PyrUp neurons' ramping rates was notably reduced in the passive task (Supplementary Fig. 14e–g), suggesting that Phase II ramping is a specific feature of the time/distance integration process.

In summary, the opposing, two-phase dynamics of PyrUp and PyrDown neurons represent a distinct coding mechanism within the hippocampus, operating in parallel with the well-established sequential firing of place cells and IGSs. This framework offers a potential mechanism for encoding elapsed time or distance. Further investigation, including manipulation of task parameters like travel distance, will be necessary to fully elucidate the specific contribution of this ramping dynamic to path integration and memory.

### Factors that influence the expression of IGS

Firing fields in hippocampal neurons are well-established. Their expression is influenced by the availability of sensory and vestibular inputs, and by the demands of behavioral tasks.

Regarding the impact of sensory and vestibular inputs, as animals navigate environments with minimal sensory cues, hippocampal neurons often show reduced spatial tuning, and a relatively small proportion of neurons exhibit firing fields[46–49]. However, robust place fields can still emerge in freely moving animals navigating in complete darkness without explicit path integration demands, such as during random foraging or running on linear tracks with rewards at ends. In these scenarios, non-visual cues, including vestibular and proprioceptive inputs, olfactory traces, and tactile information from the environment, may support stable spatial representations. Head-fixation, however, presents a distinct challenge by impoverishing vestibular input, which is known to be critical for spatial tuning of place cells and grid cells[50]. Multiple studies have attributed degraded spatial selectivity in head-fixed virtual reality (VR) environments to the lack of vestibular input[48,49,51]. Nevertheless, when sensory cues were provided, a high percentage of neurons still exhibited robust firing fields under head-fixed settings[46,47]. Thus, impoverished vestibular input alone might not fully account for the low prevalence of IGS observed in some head-fixed behavioral tasks.

Beyond sensory constraints, cognitive and memory demands significantly influence IGS generation. Even in cue-poor or cue-absent environments, the prevalence of IGSs typically increases under conditions requiring higher demands, such as tasks that involve temporal integration or working memory[6,8–10,12,21,52].

Consistent with these prior findings, our PI and immobile tasks represent environments with minimal sensory cues and impoverished vestibular input. These tasks, while less demanding than hippocampus-dependent working memory tasks[6,8–10,52], still require animals to track time or distance. Correspondingly, in these tasks, a small percentage of neurons exhibited IGFs (Fig. 2g and Supplementary Fig. 11c).

In contrast, IGFs were mostly absent in the passive task, which was designed to disrupt a robust integration strategy. First, this task lacked a start cue, and the screens remained gray throughout the recording session. Second, the distance counting within a trial was initiated by the animal's own movement, contingent upon it meeting a specific speed criterion and a minimum delay having passed since the previous reward. These contingencies introduce temporal uncertainty, thereby reducing the reliability of events such as the "last lick" or "run onset" as anchors for initiating distance or time

integration. The animal's significantly diminished predictive licking in this task confirms the effectiveness of our design. The lack of fields in the passive task aligns with previous studies reporting no clear firing fields during spontaneous wheel running in freely moving rats when no task demands were present[6,21]. Notably, the absence of fields in these freely moving rats suggests that impoverished vestibular input alone might not explain the lack of firing fields in the passive task. Our findings are also consistent with earlier evidence showing IGS impairment when animals fail to accurately estimate time or distance[6,7,53]. Collectively, these studies suggest that behavioral engagement and task demands contribute to the recruitment of internally generated sequences.

### Potential advantages of head-fixed behavioral settings

While head-fixation constrains vestibular inputs, it offers advantages for experimental control. Our VR setup was designed to minimize confounding sensory cues and ensure consistent visual input across trials. The compact treadmill design also limited tactile and olfactory cues. These features allowed us to isolate the integration of time and distance under well-controlled sensory and behavioral conditions—a level of experimental precision that remains challenging in freely moving animals, where dynamic multimodal cues are abundant and difficult to account for.

### Behaviorally aligned IGSs from run onset

While IGSs initiated from run onset were previously reported by Villette and colleagues[12], the IGSs observed in our current study differ in several aspects, including task design, behavioral demands, and sequence characteristics.

First, the task demands and behavioral relevance differ significantly. Our path integration (PI) task explicitly required animals to integrate distance to receive a reward, thereby creating a demand for integration. In contrast, the task used by Villette and colleagues did not impose explicit integration requirements and involved no reward[12], implying distinct motivational states and task demands.

Second, the relationship between the sequences and ongoing behavior differs. In our study, IGSs consistently aligned with specific behavioral events – the animal's self-initiated running toward the reward within individual trials. While Villette and colleagues also reported sequences that often began at run onset, these sequences tended to recur multiple times within a single running bout and were not locked to a structured trial-by-trial behavioral epoch. The authors characterized their sequences as "internally recurring hippocampal sequences" that encode traveled distance but without a consistent relationship to ongoing behavior[12].

Third, the sequence lengths exhibit notable differences. Villette and colleagues reported a broad distribution of sequence lengths, ranging from very short (<1 cm) to over 100 cm, further underscoring their potential dissociation from discrete, structured behavioral epochs[12]. In our task, however, the unfolding of sequences aligned with structured locomotion as animals traversed the ~180 cm cue-constant segment.

Therefore, while both studies report IGS initiated at run onset, the sequences presented in our study are uniquely defined by their consistent behavioral alignment with the task's trial-by-trial structure and explicit integration demands.

### PyrUp/PyrDown neurons are not simply "wide" firing fields

In this study, we define PyrUp and PyrDown neurons as pyramidal neurons that exhibit a clear change in firing rate around run onset. This criterion does not exclude neurons with IGFs. Indeed, some pyramidal neurons classified as PyrUp or PyrDown neurons exhibited IGFs around the run onset or near the reward zone. However, in the PI task, only ~10% of pyramidal neurons were identified as having IGFs, whereas PyrUp and PyrDown neurons collectively accounted

for ~56% of all recorded pyramidal neurons. Excluding the subset of PyrUp and PyrDown neurons that exhibited IGFs did not alter our core findings.

We emphasize that PyrUp and PyrDown neurons without IGFs are unlikely to be simply neurons with wide firing fields. Neurons with IGFs exhibited significantly stronger phase precession, a key characteristic of firing fields[22,23], than those without IGFs (Supplementary Fig. 2b). Moreover, classifying all PyrUp and PyrDown neurons as firing-field cells—regardless of their tuning specificity—would mean that approximately 56% of the entire pyramidal population exhibits IGFs. This proportion is notably higher than typically reported for neurons with firing fields in similar cue-poor contexts[46,47]. In summary, while PyrUp/PyrDown neurons and neurons with IGFs are not mutually exclusive, most PyrUp and PyrDown neurons do not meet commonly accepted criteria for firing fields in hippocampal research, including tuning specificity and robust phase precession.

Even if all PyrUp and PyrDown neurons were classified as neurons with IGFs, our central conclusion remains valid: PyrUp/PyrDown neurons can encode time and distance through a mechanism distinct from the sequential patterns formed by neurons with classical firing fields. While neurons with classical firing fields encode time or distance through specific tuning around particular time points or distances, PyrUp and PyrDown neurons use a population-level two-phase mechanism that does not rely on such tuning specificity. This coding mechanism provides insights into alternative and complementary coding mechanisms within the hippocampal network.

### The relationship between PyrUp/PyrDown responses, time/distance integration, and behavioral performance

This study demonstrates that PyrUp/PyrDown neuronal responses reflect internal time or distance integration processes, rather than simply being correlates of locomotion.

Multiple lines of evidence supported their link to integration. First, nearly half of both PyrUp and PyrDown neurons were modulated by time or distance[8]. Second, PyrUp/PyrDown neurons exhibited context-dependent responses, responding significantly stronger at the onset of trial-start runs than spontaneous runs occurring during a trial (Supplementary Fig. 10a–e). Third, PyrUp and PyrDown responses were also observed in an immobile time-estimation task, demonstrating that these responses can arise independently of locomotion (Supplementary Fig. 11d–f and k). Finally, Bayesian decoding further confirmed that the population activity of PyrUp neurons effectively encodes the passage of time (Supplementary Fig. 12a–d).

We further provided evidence dissociating PyrUp/PyrDown responses from locomotion. First, GLM analyses suggested that most PyrUp/PyrDown neurons were not significantly influenced by running speed (Supplementary Fig. 8a, b) or acceleration. Second, the temporal dynamics of PyrUp/PyrDown neurons differed from speed profiles; for example, their responses peaked earlier than the running speed (Supplementary Fig. 9c–h). Furthermore, the PyrUp/PyrDown responses showed only minor differences when comparing trials with high and low running speed (Fig. 4c–f).

Beyond encoding time or distance, PyrUp/PyrDown responses also correlated with behavioral performance. First, the strength of responses around run onset correlated with behavioral accuracy, being more pronounced in trials where animals correctly estimated time or distance (Fig. 3g–j). Second, PyrUp neurons exhibited slower decay rates in trials with a later first anticipatory lick, a behavioral proxy for the animal's internal estimate of elapsed time or distance. This result suggests that the dynamics of PyrUp neurons can predict the timing of reward-anticipatory behavior (Fig. 4g–j).

In summary, our data are consistent with the hypothesis that PyrUp/PyrDown responses represent internal integration processes that predict behavioral performance.

### Differential role of CA1 SST and PV interneurons in internally generated neuronal dynamics

Our results underscore the distinct and complementary roles of CA1 SST and PV interneurons in shaping internally generated dynamic patterns, including PyrUp/PyrDown responses and IGSs.

First, our data support that SST interneurons define an "integration window". They achieve this by generating a window of inhibition that begins with a gradual increase in firing around run onset and then tapers off as the reward zone approaches. Within this window, these interneurons exert stronger control over PyrUp neurons and IGSs (Fig. 5). Counter-intuitively, silencing SST interneurons after run onset inhibited the PyrUp response, implying that SST interneurons can disinhibit PyrUp neurons. This disinhibition likely occurs by suppressing other inhibitory interneurons, thereby favoring CA3 inputs. Behaviorally, inactivating SST interneurons impaired time or distance estimation when performed after run onset but not when performed before the reward zone (Fig. 6a–j). These results support the role of SST interneurons in sustaining continuous integration.

Furthermore, our data suggest that SST interneurons primarily affect internally generated neuronal dynamics, rather than sensory-driven responses. When visual cues were introduced into the cue-constant segment of the PI task (cue-rich task), inactivating SST interneurons after run onset no longer yielded significant changes in neuronal sequences or PyrUp response (Fig. 6k–o).

Second, PV interneurons appear to generate a reset signal at the start of integration. They achieve this by rapidly ramping up firing around run onset, leading to the synchronized shutdown of PyrDown neurons (Fig. 7). Behaviorally, inactivating PV interneurons after run onset affected integration accuracy only at the start of the trial. These results align with the hypothesis that PV interneurons reset the time or distance counter, thereby starting a new period of integration.

While PV interneurons may be necessary for proper initiation of integration, they are unlikely to be the only circuit elements involved. The initiation of integration is likely encoded by diverse populations of neurons across multiple regions that provide inputs to CA1, including other hippocampal areas and the entorhinal cortex. This idea is supported by the mild behavioral deficit observed during PV interneuron inactivation after run onset.

### Overlap of circuits that initiate integration and delineate ongoing experience into discrete units

Our data suggest that, in the PI task, the run onset at trial start may serve as the trigger for neural computation involved in time or distance integration. First, IGSs consistently started at run onset in the PI task, aligning with previous reports that IGFs can begin following the onset of locomotion[12]. Second, PyrUp and PyrDown neurons exhibited strong firing rate changes around run onset at the trial start, but not around spontaneous run onset within a trial. However, we acknowledge that the run onset is not the only cue signaling the onset of integration. For instance, in our immobile task, the last lick appears to serve a role similar to that of an initiating cue.

More generally, these findings highlight how animals can use salient sensory[11,54] or motor cues[12] with a trial's structure to initiate integration and delineate continuous experience into individual units. These results resonate with studies on human memory encoding, which suggest that the continuous stream of experience can be segmented into discrete units by detecting event boundaries[55,56]. The salient sensory or motor cues described in both past and current studies may serve as such boundaries, with the integration process within each trial comparable to integrating ongoing experience within a discrete unit.

Overall, the CA1 neural circuit motifs we identified, involving PyrUp/PyrDown neurons and their interneuron modulators, may contribute to computations underlying the detection of event

boundaries and the encoding of continuous experiences into individual units.

## Methods

### Experimental model and subject details

**Mice**. This study was based on both male and female mice (age > 8 weeks). Male mice were preferentially used in the running tasks because they were found to exhibit more consistent running behavior. We used four mouse lines: C57Bl/6J (JAX #000664), PV-IRES-Cre (JAX #017320), SST-IRES-Cre (JAX #013044)[57], and Ai14 (JAX # 007914)[58]. All procedures were in accordance with protocols approved by the Institutional Animal Care and Use Committee at Max Planck Florida Institute for Neuroscience. Mice were housed in a 12:12 reverse light: dark cycle and behaviorally tested during the dark phase.

### Method details

**Virtual reality setups**. For the virtual reality (VR) setups, a small treadmill (Janelia Design) was positioned in front of two visual displays, such that after head-fixation, the eyes aligned with the two visual displays. Customized software was written using Psychtoolbox to display the virtual environment. An Arduino control board was used to control when to display the visual stimuli. For the small treadmill, the animal's speed was measured using an encoder attached to the back wheel axis. A microprocessor-based (Arduino) behavioral control system (the miniBCS board, designed at Janelia) interfaced with a MATLAB graphical user interface controlled the trial structure, the water valve, and the encoder. A separate lick port detector (designed at Janelia) was used to convert a touch on a metal lick port into a digital pulse and to send the information to the miniBCS board. In another version of the VR setup, instead of using two monitors, a curved screen was used where the virtual reality environment was projected onto the curved screen using a projector (AKASO Mini Projector, a design kindly shared by Christopher Harvey's lab at Harvard University). In this case, customized software written with Unity was used to display the virtual environment.

In addition, a separate microprocessor (Arduino) interfaced with a customized MATLAB graphical user interface was used to operate the laser or laser diode on-off for the optogenetic perturbation experiments and to control the closed-loop manipulation of PV and SST interneurons. Behavioral data were monitored and recorded as well using MATLAB functions.

### Behavioral training

Before any surgery was performed, running wheels were added to the home cages. At 3–5 days after the headbar or fiber implantation, mice were placed on water restriction (typically 0.8-1 ml per day). After each training or recording session, mice were supplemented with additional water to ensure the same amount of water intake per day. After habituating the animals to the treadmill for at least 1-2 days, animals were trained to run head-fixed on the treadmill.

In the passive task, the screens were kept gray throughout each recording session. After the animal ran for 180 cm while this visual stimulus stayed constant, a water reward was automatically delivered. After that, an inter-trial-interval (ITI) started. The next trial started at least after 0.5 s and only when the animal's speed had exceeded 30 cm/s for more than 0.3 s and the time since the last lick had exceeded 0.3 s. Therefore, there was a random ITI of at least 0.5 s until the next trial started. This random ITI was designed to introduce variability in trial initiation. This approach aimed to discourage animals from relying on a fixed distance or temporal structure to anticipate reward delivery. It took about 1-2 weeks for the animals to run smoothly on a treadmill and to perform this task.

In the PI task, a trial started with a visual grating stimulus that lasted either 0.5 s or 1 s, and stayed the same within each session. After that, a gray-colored stimulus turned on. The animal was required to run for a fixed distance (180 cm) while this visual stimulus stayed constant, and was then required to lick within a 40 cm (in some sessions, 80 cm) unmarked reward zone to receive a water reward. The first lick in the reward zone triggered a reward, and simultaneously, the screens turned black for either 0.5 s or 1 s (kept constant within each session). If no lick occurred in the reward zone, the screens turned black to signify the error for 0.5-2 s (kept constant within each session). It took about 2 weeks for the animals to learn this task.

In the immobile task, the animal sat within a plastic tube[32]. Each trial started with a 1 s visual grating stimulus, which was then followed by a gray-colored stimulus that lasted 4 s. After that, there was a reward lasting 2 s. If the animal licked within the reward zone, a drop of water reward was triggered, and the screens turned black for a random time between 0.5-2.5 s. If the animal failed to lick within the reward zone, the screens turned black for a random time between 0.5-2.5 s. To train the animal to perform this task, the animal was first habituated to stay in the tube, and then head fixation was gradually applied. Initially, the animal was exposed to the complete trial structure with a 2 s delay period. As mice learned to concentrate licking towards the reward zone, the delay period was gradually increased in 0.5 s increments until the final 4 s delay was reached. This task took about 2 weeks to train.

### Virus injection, headbar, cannula and fiber implantation

Adult mice (2-4 months of age) underwent aseptic stereotaxic surgery to implant a custom lightweight 3D printed headbar under isoflurane anesthesia (2–3% for induction, 1–1.5% for maintenance). Buprenorphine SR LAB (0.5 mg/kg, SC) or buprenorphine (0.1 mg/kg, SC), and Meloxicam SR (5 mg/kg, SC) were administered immediately after the surgery. For acute extracellular recordings, after training to perform behavior tasks, craniotomies were performed, which were centered around anteroposterior 2.1 mm from bregma, mediolateral +/−1.7 mm from the midline to target dorsal CA1 region. Meanwhile, a ground wire was attached to the headbar, and a thin layer of silver paint (GC ELECTRONICS 22-0023-0000) was applied to the surface of the headbar for noise reduction during the recording. The skull was covered using KwikSil (World Precision Instruments) and was only removed during recordings.

For extracellular recordings with optogenetic manipulation of interneurons, AAV1_hSyn1-SIO-stGtACR2-FusionRed (Addgene 105677, 2.1e + 12 after 10 fold dilution) was used for optogenetic inactivation, and AAV5-EF1a-double-floxed-hChR2(H134R)-mCherry-WPRE-hGHpA (Addgene 20297, titer 1.4e + 13) was used for optogenetic activation. 50 nl virus per hemisphere was injected bilaterally in the CA1 region of PV-IRES-Cre and SST-IRES-Cre mice. The injection was done at least 3 days before the headbar implantation, and at least 3 weeks before the recording. The following coordinates were used for viral injections: 2.1 mm from bregma, mediolateral +/−1.7 mm from the midline, and 1.24 mm dorsoventral from the brain surface. The injection system comprised a pulled glass pipette (broken and beveled to 15-20 µm inside diameter; Drummond, 3-000-203-G/X), backfilled with mineral oil (Sigma). A fitted plunger was inserted into the pipette and advanced to displace the contents using a manipulator (Drummond, Nanoject II or Nanoject III). Retraction of the plunger was used to load the pipette with the virus. The injection pipette was positioned onto a Kopf manipulator.

For optogenetic experiments without extracellular recordings, the viral injection procedure was the same as described in the last paragraph. At least 3 days after the viral injection, optical fibers (core diameter of 200 um) were chronically implanted bilaterally to target the CA1 region using the following coordinates: 2.1 mm from bregma, mediolateral +/−1.7 mm from the midline, and 0.9 mm dorsoventral from the brain surface.

For muscimol infusion experiments, guide cannulas (26 Gauge) were implanted bilaterally above the CA1 region (coordinates: 2.1 mm from bregma, mediolateral ±1.7 mm from the midline and 1.1 mm

dorsoventral from the brain surface) during the head bar surgery. Dummy cannulas (33 Gauge) of the same length as the guide cannulas were inserted into the guide cannulas.

## Optogenetics without extracellular recordings

For each optogenetic session, a control session (~40 trials) was performed. It was then followed by a stimulation session (~100 trials in total), where the photostimulation was deployed every third trial. After that, there was a post-stimulation control session (~40 trials). To prevent mice from distinguishing photostimulation trials from control trials through stimulation light, a masking blue light (470 nm LEDs (Thorlab)) was on throughout the sessions. For both stGtACR2 and ChR2, we used a 473 nm laser (Ningbo Lasever Inc.). The laser power used was <= 5 mW. During the stimulation, 100 Hz light pulses were used to reduce the heating effect of a constant laser light. The duty cycle was varied to adjust the light power.

For interneuron inactivation using stGtACR2, the stimulation lasted 3 seconds. The stimulation can be triggered at (1) the running onset or (2) 120 cm location during the cue-constant segment. The run onset stimulation was triggered at the first time point where the animal's speed exceeded 10 cm/s for at least 200 msec after the trial start. The stimulation trial types were randomly selected for each stimulation session. The behavior analysis was performed using all the trials during the stimulation session. Only considering the stimulated trials led to similar results.

As a control, in some recording sessions, the optogenetics patch cables were only loosely mated with the implanted fiber optic cannulas, preventing light from propagating into CA1. In addition to this within animal control, we also performed control experiments where interneurons were infected with CAG-FLEX-tdTomato(AAV9) (Addgene, titer 1.90E + 12 after dilution to approximate concentration for the inactivation experiments).

## Acute extracellular electrophysiology

Two days before the recording, the animal was acclimated to the recording condition: including turning on the microscope light, removing Kwiksil that covered the skull, and adding saline to keep the craniotomy wet. On the day before the recording, a craniotomy was opened. During electrophysiological recordings, a 64-channel silicon probe (neuronexus, buzsaki64sp) was slowly lowered into the hippocampal CA1 region. Data from all channels were filtered (0.3 Hz to 10 kHz), amplified (gain = 400), and continuously sampled at 20 kHz using the Amplipex system (16-bit resolution)[59]. Time stamps of behavioral events and electrophysiological recording data were synchronized, recorded, and stored on a computer.

For acute extracellular recordings, craniotomies were centered around the dorsal CA1 region of the hippocampus using stereotaxic coordinates (AP: −2.1 mm from Bregma; ML: ±1.7 mm from midline).

Before starting a recording, probes were lowered very slowly into the brain under electrophysiological monitoring. The typical target depth from the brain surface to reach the CA1 pyramidal layer is approximately 0.95 to 1.40 mm. However, this can vary slightly between animals. During probe advancement, we continuously monitored online electrophysiological signals, including local field potentials (LFPs) and multi-unit activity (MUA), across the different recording channels of the probe.

The CA1 pyramidal layer (Stratum Pyramidale, SP) was identified based on a convergence of established electrophysiological landmarks, consistent with criteria described in previous literature (e.g., Mizuseki et al. (Nature Neuroscience, 2011)[24]): (1) Prominent theta oscillations: Strong and coherent theta-band activity (typically 6-10 Hz) in the LFP, particularly evident during locomotion. (2) Sharp-wave ripples (SWRs): The presence of high-frequency ripple oscillations (100-250 Hz) superimposed on larger sharp waves in the LFP, which are most prominent in CA1 SP and are a hallmark of this layer,

especially during quiet wakefulness or immobility. (3) Increased MUA and spike bursts: A noticeable increase in the density of MUA and the occurrence of complex spike bursts, characteristic of pyramidal neuron firing, as the probe contacts traversed SP. (4) Polarity inversion of sharp-waves/ripples: A key indicator is the phase or polarity inversion of sharp-waves (and sometimes theta waves) across the radial axis of CA1. We looked for this inversion pattern across the shanks/sites of our high-density probes. These features were used collectively to determine that the probe was positioned in the CA1 pyramidal cell layer.

For extracellular recordings with optogenetics, a 64-channel silicon probe with fibers mounted on the shanks (neuronexus, buzsaki64sp-OA64LP) was used. Photostimulation followed the same protocol as described in the optogenetics section above. The light source was a self-constructed laser diode array (6 diodes, 450 nm, osram-os, PL450B). Each diode was coupled to one shank on the probe. 2 to 5 diodes were coactivated during the stimulation. The stimulation laser power was <1.2 mW at the tip of the probe.

For optogenetic tagging experiments, interneurons were tagged using 1 ms pulses of blue light. Spike times around each pulse were binned with a resolution of 1 ms and the significance of light responses in a 5 ms window after the light pulse was assessed using the stimulus-associated spike latency test (SALT)[60]. Neurons with a p < 0.01 that responded to at least 40% of light pulses were identified as PV- or SST-positive. 30–40 tagging pulses were applied per recording.

## Muscimol infusion

Muscimol hydrobromide (Tocris-0289) was dissolved in 0.9% saline. The control solution was saline. The muscimol injections were carried out as follows. First, an injection cannula was connected to a 10-µl Hamilton syringe through Tygon tubing (Tygon 720993) and filled with muscimol (1 mg ml$^{-1}$) or saline (0.9%). Then the syringe was mounted into a microinjection pump (UMP3 with SYS-Micro4 controller). At the beginning of the injection procedure, the dummy cannula was removed from the guide cannula and replaced by the injection cannula (33 Gauge), which extended 0.5 mm deeper into the brain than the guide and dummy cannulae. Then 300–500 nl of muscimol or saline was slowly injected (100 nl/min) into CA1 and the injection cannula was left in place for another 3 min after the infusion was completed. Next, the dummy cannula that was cleaned with alcohol and dipped in sterile mineral oil was inserted back into the guide cannula. During the infusion, animals performed the PI task with water automatically delivered at the end of each trial. Animals did not show signs of stress or discomfort during the procedure. The animals were released from the setup after the infusion, and ~30 min after the completion of the infusion, animals were placed back onto the VR setup. A typical muscimol infusion recording was composed of one control session before the infusion, and multiple recording sessions at 30 min, 1 h, 2 h after the infusion. Sessions after infusion were combined as "Musc" sessions in the analysis.

## Spike sorting

To identify spikes, we performed off-line spike sorting on the recorded files following published methods using Klusters[61] and Kilosort[62]. Units were further selected based on the percentage of spikes that violated refractory period and the mahalanobis distance from other units.

## Histology

Mice were perfused transcardially with PBS followed by 4% PFA. Brains were post-fixed overnight and transferred to PBS before sectioning using a vibratome. Coronal 50 µm free-floating sections were processed using standard fluorescent immunohistochemical techniques. PV immunoreactivity (Swant, GP72; 1:2000, guinea pig). The primary antibody was visualized by Alexa Fluor 488 AffiniPure Donkey Anti-Guinea Pig IgG (H + L) (1:1,000, Jackson ImmunoResearch Laboratories, 706-545-148). SST immunoreactivity (Millipore, MAB354; 1:250,

rat). The primary antibody was visualized by Alexa Fluor 488 AffiniPure Donkey Anti-Rat IgG (H + L) (1:1,000, Jackson ImmunoResearch Laboratories, 712-545-153).

### Quantification and statistical analysis

**Behavior analysis.** To calculate how the speed changed over distance during the cue-constant segment in the PI and passive tasks, the histogram of speed within each trial was calculated using 1 mm distance bins. 15 trials were randomly selected from each session, and the average and SEM of speed were calculated across all the selected trials from all the sessions in each task. Similarly, to calculate how the number of licks changed over distance, a lick histogram with a bin size of 1 cm was first calculated for each trial. Then the same calculation as the speed was applied to get the average and SEM. To calculate the overall mean speed or lick within a certain distance range, the mean speed or lick number was first calculated for each session, then averaged across sessions from the same task. To calculate total stop time, we first smoothed running speed using a moving average with an 80 ms window, and then summed the time when the speed was lower than 1 cm/s from the running onset to the end of the trial.

We found recordings from the PI task with matching running speed to the passive task, and identified 4 animals (10 recordings in total) where the mean speed did not differ from the passive task recordings. These recordings were used to compare neurons with IGF in both tasks.

### Extracellular recording analysis

**Alignment with sensory or motor cues.** For both the passive and PI tasks, both the neuronal activity and the behavior of each trial were aligned with either the start of the trial or the running onset. In both tasks, the running onset for each trial was first defined as the onset of the first running bout that lasted more than 0.3 sec with a speed > 10 cm/s, starting from the previous trial after the animal had traveled the 180 cm cue-constant segment. The onset was then traced back to when the running speed reached 1 cm/s before the running bout and after 180 cm in the cue-constant segment of the previous trial. If the speed never reached 1 cm/s, the time point when the speed was the lowest was considered as the onset, and this trial was classified as "no clear run onset".

**Firing rate profile of each neuron.** Pyramidal neurons were identified based on their spike waveforms, and if their firing rates fell between [0.15 7] Hz. The spike train of each pyramidal neuron was first smoothed with a Gaussian function (s.d. = 30 ms) and then averaged across trials to get the average firing rate profile of that neuron. When examining how the firing rate changes over distance, the spike train was first binned using 1 mm spatial bins, and then smoothed with a Gaussian function (s.d.=20 mm).

**Firing field identification.** The averaged firing rate profile of each neuron was used to identify firing fields (internally generated fields).

To identify firing fields over distance from the trial start, as shown in Fig. 1, the following criteria were applied: (1) a minimum mean firing rate of 0.09 Hz; (2) minimum mean trial-by-trial Spearman correlation of the firing rate profile > 0.15 and minimum spatial information[63] > 0.25 bits/spike; or minimum mean trial by trial correlation > 0.09 and minimum spatial information > 0.7 bits/spike. (3) total number of trials > 15, and the neuron was active in at least 40% of the trials. (4) the maximum field width was 150 cm, measured by the width when the firing rate fell below 10% on both sides of the peak firing rate. (5) There was no other peak within 30 cm outside the field. The parameters were chosen based on visual inspection.

To identify firing fields over time from run onset, as shown in Fig. 2, we used similar criteria as mentioned above. Some of the parameters were readjusted based on visual inspection: (1) minimum

mean trial-by-trial correlation > 0.12 and minimum temporal information > 1 bits/spike; or minimum mean trial-by-trial correlation > 0.09 and minimum temporal information > 2.5 bits/spike. (2) The maximum field width was set to 4.5 sec. (3) There was no other peak within 1.7 sec outside the field. IGFs were identified based on neuronal activity profiles when aligned with run onset. For the selected population of IGF neurons, we then showed how their trial-by-trial correlation changes if their activity was instead aligned with cue onset.

For the immobile task, the same method was used to identify firing fields over time with slightly different parameters. Neuronal activity was aligned with the last lick of each trial in the immobile task, and the following parameters were used: minimum mean trial-by-trial correlation > 0.1 and minimum temporal information > 0.65 bits/spike.

**Auto-correlogram.** The spike time auto-correlogram of each neuron was calculated after binning its spike train using 1 ms time bins.

**Phase precession.** Phase precession was calculated as the difference between the theta modulation frequency of individual neurons and the theta frequency of the local field potential (LFP). The theta modulation frequency of individual neurons was calculated using the peak time of its auto-correlogram within 100–200 ms and −200- −100 ms windows[64]. One LFP trace recorded from the CA1 pyramidal layer was filtered with a third-order Chebyshev Type II filter (4-16 Hz). LFP instantaneous frequency in the theta band was calculated using the Hilbert transform. The instantaneous frequency was then averaged over each trial and across trials to compute the LFP theta frequency.

**Criteria for good and bad trials.** The criteria for identifying good trials in the PI task are: (1) The animal came to a complete stop at the reward location of the previous trial just before the run onset (speed <1 cm/s). (2) The subsequent running was mostly continuous, with a total stop time shorter than 2 s after run onset. (3) The animal obtained a reward. Otherwise, the trial was considered a bad trial. In trials classified as "bad" due to the animal not coming to a complete stop before initiating a run, we observed that the animal typically exhibited a significant deceleration. For these trials, the "run onset" for analysis was defined as the moment of lowest instantaneous speed between the animal reaching the reward zone in the previous trial and the start of the current trial. This approach aimed to identify a consistent behavioral marker for the initiation of a new potential integration period, thereby allowing for a more direct comparison with "good" trials where a clear stop preceded the run.

In addition, we also redefined bad trials by removing the "no stop" criterion, and trials were included if they met criteria (2) and (3).

For the immobile task, trials where mice missed the reward or began licking before the halfway point of the delay period were labeled as bad trials.

Recordings with at least 15 good trials and 15 bad trials were included in the analysis.

**Comparing firing fields between good and bad trials.** To directly control for the effect of the trial count difference between good and bad trials, we performed a random subsampling procedure. For each recording session that has a minimum of 15 bad trials, we randomly selected a subset of "good" trials equal in number to the "bad" trials from that same session.

Using these trial-count-matched subsets, we recalculated the trial-by-trial activity Spearman correlation for IGF neurons, which served as a measure of field reliability.

This entire subsampling and correlation recalculation process was repeated five times to ensure the robustness of the findings and account for variability in the random sampling. Due to space limitations, representative results from 3 iterations were included.

**Identification of PyrUp and PyrDown neurons.** The change in firing rate around the run onset for each neuron was defined as the ratio between the averaged firing rate in a window 0.5 to 1.5 s ($FR_{aft}$) and −1.5 to −0.5 s ($FR_{bef}$) about the running onset ($R=FR_{aft}/FR_{bef}$). The PyrUp neurons were those with $R > 3/2$, while the PyrDown neurons were those with $R < 2/3$. PyrOther neurons were those with R between 3/2 and 2/3. The presence of PyrUp/PyrDown neurons is not sensitive to the threshold value for R. The relative firing rate change of the average firing rate profile around run bouts and trial run onsets was calculated for each neuron in a window 0.5 to 1.5 and −1.5 to −0.5 s about the running onset with the formula $(FR_{aft} − FR_{bef})/(FR_{bef} + FR_{aft})$.

The same criteria were used to identify PV-Up and SST-Up neurons.

To validate the PyrUp and PyrDown subpopulations, a shuffling method was used. For each neuron, we randomly circularly shifted the firing rate profile of individual trials and recalculated $FR_{aft}/FR_{bef}$. We repeated this operation for 1000 iterations to generate a null distribution of $FR_{aft}/FR_{bef}$. Neurons with original $FR_{aft}/FR_{bef}$ values exceeding the 95th percentile or falling below the 5th percentile of this shuffled distribution were classified as PyrUp or PyrDown, respectively.

The same method was used to identify PyrUp and PyrDown neurons with respect to the run onset in the passive task, and to the last lick in the immobile task.

**PyrUp and PyrDown neurons with respect to running bouts in the PI task.** Run bouts were identified as periods where locomotion exceeded 10 cm/s for a duration of at least 0.5 s. The onset of each run bout was traced back to the point where speed first exceeded 2 cm/s. If running speed in the window 0.5 s before a run bout exceeded 10 cm/s at any time point that run bout was excluded from analysis. Run bouts that occurred within the first 50 cm of a trial were also excluded from the analysis. Spike times for each neuron were aligned with the run bouts and firing rates were calculated in 2.5 ms bins and smoothed with a Gaussian function(s.d. = 30 ms).

Speed matching was performed for each recording by first calculating the mean speed and standard deviation of spontaneous run bouts in a period from −1.5 to 0 s and 0 to 1.5 s around bout onsets. Trial start runs in the same session were considered speed-matched if they fell within one standard deviation of the mean speed for both periods. Recordings with less than 15 total run bouts and less than 15 speed-matched trial run onsets were excluded from the analysis.

**Comparing PyrUp/PyrDown responses across high and low speed trials.** We categorized good trials based on the mean running speed during the initial 4 s post-run onset into "high speed" ( > 45 cm/s) and "low speed" (35–45 cm/s) groups. The speed thresholds were selected based on the mean running speed in the PI task. Recordings with at least 15 trials in each category are included.

**Comparing PyrUp/PyrDown responses across early and late first lick trials.** We categorized good trials based on the animal's first lick time into "early lick" (first lick <2.5 s) and "late lick" (first lick > 3.1 s) groups. Recordings with at least 15 trials in each category were included. Selecting other lick time thresholds did not significantly alter the results.

**Decay/rise time constant of PyrUp/PyrDown response.** For each PyrUp neuron, the first peak in its mean firing rate profile was identified. The decay time constant was extracted by fitting the post-peak mean firing rate profile with the function a*exp(bx), where the time constant is defined as −1/b.

Similarly, for each PyrDown neuron, the significant peak 1 sec after run onset was identified as for PyrUp neurons. The time constant was

calculated by fitting the function a*exp(bx) from the peak backward to 1 sec after run onset.

**PyrUp/PyrDown neuron laminar location within CA1 pyramidal layer.** We used two methods to estimate the locations of PyrUp/PyrDown neurons within CA1 pyramidal layer. First, we used sharp wave ripple (SWR) power for depth estimation, following the methodology described in Mizuseki et al., Nature Neuroscience, 2011[24]. This method has potential limitations in the precision of layer center estimation using SWR.

Second, we implemented the relative anatomical depth estimation method proposed by Geiller et al., Nature Communication, 2017[25]. This approach utilizes pairwise comparisons of spike amplitudes from neurons recorded on the same shank, allowing for a more sensitive assessment of relative depth.

**PyrUp/PyrDown neuron speed modulation.** Animal speed was computed from the wheel encoder in 30 ms bins and smoothed using a 1D-Gaussian with a 250-ms standard deviation[31]. For each neuron, firing rates were calculated in 30 ms bins and smoothed using the same Gaussian. The correlation between the resulting time series of speed and firing rate was calculated using Pearson's r. For visualization, scatterplots of speed versus firing rate were downsampled to 10% of all data points. To estimate confidence intervals for the relationship between speed and firing rate, data points were first grouped into equally spaced, non-overlapping speed bins. For each bin, the mean firing rate was calculated, and the 95% confidence interval of the mean was obtained using the critical values from the t-distribution. Comparisons between the speed correlations of different groups of neurons were made using the Wilcoxon rank sum test.

**Theta phase.** The peaks, valleys, and zero crossing points of LFP were detected and assigned as phases 0, 180, and 90 or 270, respectively. Phases between these points were linearly interpolated[65]. Peaks in the LFP corresponded to the minimum firing of CA1 pyramidal cells. Please refer to Wang et.al. 2015[7] for more details. Theta phase histogram of each neuron was calculated using 5-degree bins between 0 and 360 degrees.

**Interneuron clustering.** All recorded neurons were first broadly classified as either putative pyramidal cells or putative interneurons based on established electrophysiological criteria, including average firing rates (pyramidal neurons: > 0.15 Hz and <7 Hz, interneurons: > 3 Hz) and spike waveform and auto-correlogram characteristics (e.g., Csicsvari et al. Neuron, 1998[66]).

Opto-tagged SST+ and PV+ interneurons, identified by their direct, short-latency light-evoked responses, served as ground truth references. We then extracted electrophysiological features (theta phase histogram, auto-correlogram and laminar location) from both tagged and untagged putative interneurons. The estimation of depths of interneurons and pyramidal neurons relative to the center of the pyramidal layer was calculated following the method in Mizuseki et.al., 2011[24].

Principle component analysis (PCA) was performed on the matrix formed by the normalized theta phase histograms of all the interneurons. The number of PCA components was selected such that the accumulated explained variance reached 95% unless the number of components was larger than 30. The same treatment was done on the matrix formed by the auto-correlograms of all the interneurons. The PCA components of the theta phase histograms, the auto-correlograms, and the estimation of the depth of interneurons relative to the center of the pyramidal layer were features used to perform K-means clustering of interneurons. The number of clusters was determined by the evalclusters function in MATLAB, and then went through visual inspection. The cluster exhibiting the highest degree of overlap with

the tagged SST or PV interneurons and similar firing phenotypes established for the SST or PV cell group[13,19] was classified as putative SST or PV interneurons.

**Decoding analysis.** For each recording session, we trained a naïve Bayesian classifier built around the Matlab function fitcnb[33] to decode time from the population activity of PyrUp neurons without firing fields. Only recordings with at least 30 good trials and 20 neurons were included in the analysis. A uniform prior probability distribution was used for decoding. PyrUp neurons were identified by thresholding their R ratio. Bayesian decoding posterior probability and decoding error were determined using tenfold cross-validation. Chance was determined by calculating the decoding error after performing randomized circular shifts of the firing rate profiles to each trial for N = 50 times. The time bin for decoding was 0.2 s.

We attempted the same decoding analysis using PyrDown neurons. However, due to the significantly lower number of PyrDown neurons per recording compared to PyrUp neurons, few recordings met the necessary selection criteria for reliable analysis. Therefore, these results were not presented here.

**Spectrograms and theta state changes.** Multitaper spectrograms were constructed from the LFP channels in the layer center of CA1 using the mtspecgramc function in the Chronux library (http://chronux.org/). A sliding window of T = 1 s and a frequency resolution of $\Delta$f=2 Hz was used. The time-half-bandwidth product and the number of tapers used were calculated as in Prerau et.al.[67] Theta state was quantified by a theta/delta ratio, calculated as the quotient of the average power in the 6-8 Hz and 2–4 Hz bands. A theta/delta ratio of less than 4 during locomotion was used as a threshold to exclude recordings with movement artifacts from the analysis.

**Long pauses and inter-trial intervals.** Long pauses were defined by immobile periods when running speeds were below 2 cm/s for 5 s or longer. For the purposes of brain state change analysis, running bouts were defined by periods when running speeds were 10 cm/s or greater for a minimum of 3 s. Inter-trial intervals were defined as the periods in between the delivery of a reward in one trial and the running onset of the following trial. Slow runs were defined as periods where running speeds fell between 5 and 20 cm/s.

**Generalized Linear Model (GLM).** Our GLM analysis followed the method in Kraus et al., Neuron 2013[8]. Please find the details in their paper.

Briefly, the spiking activity was modeled as an inhomogeneous Poisson process with the firing rate a function of various covariates that modulate spiking activity[9,68].

*For neurons with IGF*, the spiking activity was modeled as

$$\lambda_{T+D}(t) = \lambda_{time}(t) \times \lambda_{distance}(t) \times \lambda_{speed}(t) \times \lambda_{history}(t) \tag{1}$$

Here $\lambda_{T+D}(t)$ is the instantaneous firing rate. For 1 ms time bin, it can be interpreted as the probability of a spike per bin given the small bin size ("T" and "D" stand for "time" and "distance" respectively). $\ln(\lambda_{time}(t))$ is a sixth polynomial of time relative to the start of each run onset (Eq. 2), $\ln(\lambda_{distance}(t))$ is a six-order polynomial of the distance the belt moved since the start of each run onset (Eq. 3), $\ln(\lambda_{speed}(t))$ is a first-order polynomial of the running speed (Eq. 4), and $\lambda_{history}(t)$ contains the spiking history of the neuron (5). Using a six-order polynomial for $\ln(\lambda_{time}(t))$ and $\ln(\lambda_{distance}(t))$ yields similar results.

$$\lambda_{time}(t) = e^{\sum_{i=1}^{6} \alpha_i \tau(t)^i} \tag{2}$$

$$\lambda_{distance}(t) = e^{\sum_{i=1}^{6} \beta_i d(t)^i} \tag{3}$$

$$\lambda_{speed}(t) = e^{\delta_1 + \delta_2 s(t)} \tag{4}$$

$$\lambda_{history}(t) = e^{\sum_{i=1}^{5} \theta_i n(t-(i)ms, t-(i-1)ms) + \sum_{i=6}^{11} \theta_i n(t-(25i-120)ms, t-(25i-145)ms)} \tag{5}$$

In Eq. 2, $\tau(t)$ refers to the time since the run onset last started, and the six $\alpha$'s are parameters that control the degree to which the spike rate is modulated by time. In Eq. 3, $d(t)$ refers to the distance traveled since the start of each run onset, and the six $\beta$'s are parameters that specify the influence of this distance on spike rate. In Eq. 4, $\delta_1$ is a constant influencing the mean firing rate, $s(t)$ refers to the running speed at time t, and $\delta_2$ specifies the influence of speed on spike rate. In Eq. 5, $n(t_1, t_2)$ is the number of spikes that occurred between times $t_1$ and $t_2$. The eleven history terms represent five 1 ms bins going back 5 ms (0–1 ms, 1–2 ms, 2–3 ms, 3–4 ms, 4–5 ms) and six 25 ms bins going back an additional 150 ms (5–30 ms, 30–55 ms, 55–80 ms, 80–105 ms, 105–130 ms, 130–155 ms). Each history term is modulated by one $\theta$ parameter.

In the reduced model, either $\lambda_{time}(t)$ or $\lambda_{distance}(t)$ was removed from the full model $\lambda_{T+D}(t)$, and the deviance of the reduced model from the full model was calculated.

*For PyrUp and PyrDown neurons*, the spiking activity was modeled as

$$\lambda_{T+D}(t) = \lambda_{time}(t) \times \lambda_{distance}(t) \times \lambda_{speed}(t) \times \lambda_{lick}(t) \times_{history}(t) \tag{6}$$

$$\lambda_{lick}(t) = e^{\gamma_1 + \gamma_2 l(t)} \tag{7}$$

The full model in Eq. 6 added the contribution of licking into Eq. 2, where $\ln(\lambda_{lick}(t))$ is a first-order polynomial of the lick rate over time (Eq. 7). In Eq. 7, $\gamma_1$ is a constant that influences the mean firing rate, $l(t)$ refers to the lick rate at time t, and $\gamma_2$ specifies the influence of lick rate on spike rate.

In the reduced model, either $\lambda_{time}(t)$ or $\lambda_{distance}(t)$ or $\lambda_{speed}(t)$ or $\lambda_{lick}(t)$ was removed from the full model $\lambda_{T+D}(t)$, and the deviance of the reduced model from the full model was calculated.

Similarly, to investigate the contribution of acceleration, the full model was modified to

$$\lambda_{T+D}(t) = \lambda_{time}(t) \times \lambda_{distance}(t) \times \lambda_{speed}(t) \times \lambda_{lick}(t) \times \lambda_{acceleration}(t) \times_{history}(t) \tag{8}$$

$$\lambda_{acceleration}(t) = e^{\omega_1 + \omega_2 a(t)} \tag{9}$$

where $\ln(\lambda_{acceleration}(t))$ is a first-order polynomial of the acceleration over time (Eq. 9). In Eq. 9, $\omega_1$ is a constant influencing the mean lick rate, $a(t)$ refers to the acceleration at time t, and $\omega_2$ specifies the influence of acceleration on spike rate.

In the reduced model, $\lambda_{acceleration}(t)$ was removed from the full model $\lambda_{T+D}(t)$, and the deviance of the reduced model from the full model was calculated.

## Statistics

No statistical methods were used to predetermine sample sizes, but our sample sizes are similar to those reported in previous publications. Data collection was not performed blindly to the conditions of the experiments, however, the data collection and data analysis were performed by two experimentalists to ensure the reproducibility of the results. For optogenetic experiments, stimulation types were randomly determined for each session. We used statistical tests to estimate significant differences between groups: non-parametric Wilcoxon rank-sum test, non-parametric Kruskal-Wallis test, and parametric two-sample t-test. The Wilcoxon rank-sum test and two-sample t-test are two-sided. All shuffling was done over 1000 iterations.

**Reporting summary**

Further information on research design is available in the Nature Portfolio Reporting Summary linked to this article.

## Data availability

Data corresponding to all main figures are available at the following open source repository: https://doi.org/10.17617/3.I2QI6Q. Source data are provided with this paper.

## Code availability

Custom scripts were written in MATLAB 2023b. Code related to this paper is available at GitHub (https://github.com/the-wang-lab/Code-Heldman-NC-2025).

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

## Acknowledgements

We thank T. Harris and E. Moser for comments on the manuscript; L. Abbott, E. Schuman, and D. Fitzpatrick for discussions; S. Sawtelle, B. Wisnicki, and X. Zhao for help with the virtual reality setups. M. Klement, N. Daniel, and the machine shop at MPFI for making mechanical parts for the experimental setups; J. Wells and ARC at MPFI for taking care of animals. This work was funded by the Max Planck Society and the Max Planck Foundation, NIH R01 NS119503.

## Author contributions

Y.W. conceived the project. Y.W. and R.H. designed experiments (with input from D.P.). R.H. performed electrophysiological experiments. D.P., Y.W., and X.Z. performed behavioral experiments with optogenetics. Y.W. and R.H. analyzed data. Y.W., R.H., and B.M. discussed the results and wrote the manuscript, with contributions from all the authors.

## Funding

## Competing interests

The authors declare no competing interests.
