## [Transparent Peer Review file · Nature Communications]

Time or distance encoding by hippocampal neurons via heterogeneous ramping rates

Corresponding Author: Dr Yingxue Wang

Version 0:

Reviewer comments:

Reviewer #1

(Remarks to the Author)

In this manuscript, Heldman and colleagues present a compelling study that uses dense electrophysiological recordings in mice to investigate CA1 neural activity in a novel path-integration task adapted for a head-fixed preparation. In this task, mice are trained to run on a treadmill surrounded by screens, waiting for a cue to appear before running approximately 180 cm and licking to receive an otherwise hidden reward. The authors uncover a previously undescribed activity pattern in CA1 pyramidal neurons, where one subset increases firing following cue presentation at the onset of the run toward the goal (PyrUp), while another decreases (PyrDown). Importantly, they demonstrate that this activity pattern is not merely a reflection of task structure but instead correlates with cognitive demand, as evidenced by differences in firing rates between good and bad trials. They propose a mechanistic model in which PyrUp neurons initiate an integrative process (tracking distance or time), setting the stage for PyrDown neurons to guide accurate estimation of the goal location. Through elegant optogenetic manipulations targeting distinct interneuron subtypes (PV and SST), the authors establish causal relationships between these activity patterns and behavioral performance.

Overall, this study is a tour de force. It integrates large-scale neural recordings, a cognitive task, and specific causal manipulations to reveal a previously uncharacterized circuit mechanism underlying distance estimation in path integration. The dataset is extensive and is combined with rigorous analyses and well-controlled experiments. Every potential logical gap is thoughtfully addressed, either experimentally or analytically.

It is evident that this manuscript has undergone substantial revisions and incorporates an impressive amount of additional data since its initial conception. I have no further data requests and only a minor analytical suggestion that could help clarify a small aspect of one result. The depth of the study (its strength) also means that the manuscript is dense, and I have a few minor suggestions to enhance clarity and readability. But these are primarily stylistic and should not impact the core findings. I have no major concerns regarding how data was collected processed and analyzed.

Comments:

1. The authors have examined the anatomical distribution of PyrUp and PyrDown neurons (deep vs. superficial) as reported in Figure S2F, but I could not find a methods description of how depth was estimated. Was this based on CSD, SWR power, or another approach? A “blunt” depth estimation using these metrics may not always yield reliable results and I’d like to suggest using a relative anatomical estimation among neurons recorded from the same shanks, relying on a spike amplitude metric (we describe this approach in Geiller et al., 2017, Nature Communications). Maybe the authors will find it useful. This pairwise comparison method could reveal more subtle differences, given the small size of the CA1 cell layer in mice (~50 μm).

2. This is subjective, but I suggest reducing the amount of information in the main figures, as the key message of each figure feels somewhat diluted. I found it difficult to parse the primary takeaways from individual panels. For example, the behavioral impact of interneuron manipulations is a critical finding but feels somewhat “buried” within the figures. Of course, the authors should feel free to disregard this suggestion if they strongly prefer their current structure.

3. This is more a point of curiosity than a comment: in Figure 5M, the authors introduce cues on the track to shift the task from path integration to a goal-oriented behavior, yet they still observe PyrUp/PyrDown responses. Could the authors provide some insight into why this occurs? One possibility is that distance integration remains a default strategy because that is what the animals were initially trained to do?

4. Regarding Figure 5N-S, have the authors checked whether the place cells detected in this condition are also IGS cells

when cues are removed? I am not sure someone has ever looked at that?

I am confident that this paper will be well-received by the community, and I strongly recommend its publication in Nature Communications.

Tristan Geiller

(Remarks on code availability)

Reviewer #2

(Remarks to the Author)

The authors describe time-cell-like spiking sequences in the mouse CA1 as well as two populations of pyramidal neurons - one that is excited and one that is inhibited - during a cued path integration task in mice. Given that these groups exhibit their peak excitation/inhibition at the same time (of peak speed?) and that they have variable decay/rebound rates, time can be decoded during path integration. By optogenetically inactivating SST or PV interneurons in transgenic mice during this behaviour, they then find that SST-silencing suppressed the first subpopulation and PV-silencing suppressed the second one.

The manuscript contains an impressive amount of work (electrophysiology, behavioural tests, analysis, closed-loop optogenetics) with extended analyses and clear, polished figures. Unfortunately, the results and conclusions are not particularly novel or convincing. The main issue stems from the task design where locomotion motifs make it very difficult to dissociate time from distance or exclude the strong possibility that neuronal activation is simply triggered by locomotion and does not necessarily time- or distance-integrate, blurring most conclusions.

The manuscript also suffers from lack of cohesion, making it feel like 2-3 small papers combined in one. E.g. Fig 1 is about distance dependence leading to a major conclusion that firing fields encode distance. Yet practically the rest of the paper is about time encoding and interneuronal circuits. Moreover, the analysis on firing fields and spiking sequences is mostly abandoned from Fig 3 onwards, and it is not even mentioned in the abstract.

Major points:

- Internally generated spiking sequences, triggered by run onset and integrating distance is not a novel observation. They have been described before by Villette, Malvache et al Neuron 2015. The authors should discuss their Fig1-2 analyses in relation to those findings.
- The biggest issue with the manuscript is that time and distance cannot be easily separated because locomotion patterns seem to be very stereotyped. Even though the authors go to some lengths to rule out that locomotion is driving their findings, it is not convincing. For example, PyrUp/Down neurons may simply be correlated/anti-correlated with locomotion, respectively. Since locomotion is stereotypical peaking at a more-or-less fixed timepoint after run-onset, they all peak or dip at that timepoint after run-onset. This (anti)correlation is better in “good” trials because a good locomotion pattern is required there (it is a criterion for selecting “good” trials). In this case, neurons that are either correlated or anticorrelated with locomotion would not constitute “two previously unknown functional subpopulations of CA1 pyramidal neurons”.

Related to this, running speed is only plotted against distance in Fig 1 and only against distance. It would be essential, to show locomotion over time, to match with the firing rate plots. It is possible that locomotion and rates strongly correlate.

- Given these facts, claims like “IGS that encodes the moment-to-moment running distance” seem unconvincing, since throughout the manuscript it never becomes clear whether distance or time is being encoded. If anything, sequences seem clearer when plotted over time rather than distance, but this is not adequately explored.
- Total lack of any fields in the passive task is interesting and quite surprising and needs further examination. This result may be driven by 2 issues:
 - 1) It may be biased by a specific threshold the authors used to determine when a cell has a significant field. Looking into the Methods, they use a long list of criteria for field detection “based on visual inspection”, resulting in only 4% cells having IGFs! One of them is: “minimum mean trial-by-trial Spearman correlation of the firing rate profile > 0.15”. Looking into Fig 2H, it seems that even IGF cells have an average correlation of just under 0.15 when aligning to the cue. Unless I am reading this wrong, it could be that the 0.15 criterion is just right to exclude all cells in the passive task. These cells may still yield more noisy fields than in the IP task, but a different threshold would probably reveal many of them still having a field.
 - 2) Another issue is how firing rates are aligned across trials. Based on Fig 1D, in the passive task mice run with more homogeneous speeds, whereas they accelerate more slowly in the PI task after each reward. This is probably because the authors use a variable ITI defined as follows: “The next trial started at least after 0.5s and until the animal’s speed has exceeded 30 cm/s for more than 0.3s and the time since the last lick has exceeded 0.3s.”. It is unclear why these criteria have to be met to start counting distance travelled by the mouse but in any case, mice are already running when the trial is initiated (meaning distance starts being counted). Without a fixed “landmark” in the passive task (like the cue in the IP task) and with mice already running, using distance as a metric to find IGFs in the passive task is not a fair comparison to the IP task. Distance from what? Later, in Fig 2G, they check for IGFs from run-onset, but it is not clear if that run onset is within the

trial or includes run-onset during the ITI. Run-onset after reward licking is probably a better way to search for fields in the passive task.

- Authors claim that spiking sequences in “bad” trials are more noisy. This could simply be due to undersampling (only 12% of trials are “bad”). IGFs are typically noisy, requiring averaging across many trials to see the field, so using only a small sample of trials could lead to blurrier fields. The authors should repeat their comparisons after subsampling the “good” trials (with multiple iterations) to match the “bad” trials.

- The authors show that they can decode time based on firing rate decay rates of PyrUp/Down. But any time series with fixed temporal profile can be used for time decoding. It does not convincingly show that these cells encode time, rather than just encode locomotion (see comment further up). The authors claim that spontaneous runs don't yield similar responses, but that is probably because spontaneous runs do not share the stereotyped run motifs of regular runs. They maybe short, jerky or slow crawls (the locomotion traces are not shown in Fig S3A-B after all). The locomotion-dependence is supported by the analysis in Fig S3C: when comparing only spontaneous runs with matched-speed to regular runs, the firing rates become much closer, further supporting that locomotion may be underlying these responses.

In fact, it is not clear how matched-speed is defined. In the Methods it is stated: “We found recordings with matching running speed between the passive and PI tasks, and identified 4 animals (10 recordings in total) where the mean speed did not differ from the passive task recordings”. What is the statistical criterion behind “did not differ”? Please clarify.

- The immobile task is not very clear either. Are mice supposed to not lick during the delay or can they be constantly licking throughout the 4 sec? Lick rates are not shown. If it is the latter case, how do the authors know they time integrate? It is also unclear why the authors align firing rates by the last lick of previous trials, rather than cue onset or offset, which is a specific temporal anchor in a trial.

In the immobile task, the increase/decrease of firing in PyrUp/Down seems to happen during the delay period and may be related to timing or simply triggered by the cue, or even correlated with licking. In either case, it still doesn't convincingly exclude that locomotion triggers UP/DOWN responses in the normal task.

- The authors use a method for identifying non-tagged putative SSTs and PVs. Even though their metrics look similar to those from corresponding tagged cells, it is unclear what they look like for other cell types (e.g. pyramidal cells). Why not just analyze tagged cells? Is it a sampling issue?

- Silencing SSTs reduces PyrUp firing. What does it do to locomotion patterns? The authors should include a locomotion comparison (not just average speed across the entire trial).

- I found no details in Methods on how CA1 cells were selected. How much was each probe lowered into the brain? How did the authors determine which channels were situated in CA1?

Minor points:

- The Results are separated into sections whose message bleeds into each other, making it hard to follow the manuscript structure. For example, the first section ends with conclusion: “Thus, IGSs preferentially occur during the task requiring distance integration”. Half way through the 2nd section, results imply that “engaging in distance integration is necessary for generating IGFs”. These analyses could form a single section. The same holds for the Figures and supplemental figures. Often, a supplemental is related to multiple main figures and vice versa. This makes following the analyses difficult.

- Repetition: “In the PI task, the run onset potentially marked the starting point of integration in the PI task”

- Figures not appearing in order in the text. E.g. Fig 4D referenced earlier than 4A-C and similarly elsewhere.

- “In addition to IGSs, CA1 pyramidal neurons generally exhibited improved trial-by-trial activity correlation when aligned to run onset (Figures S2H).” Fig S2H is probably a typo here as it doesn't show that.

(Remarks on code availability)

Reviewer #3

(Remarks to the Author)

In the current manuscript Heldman and colleagues train mice to perform a cued path integration task while recording sequential activity patterns in the hippocampal formation. I like the task. They find that the hippocampus generates fields anchored to particular distances/times from run onsets and that these internally generated fields (‘IGFs’) are not present when the animal runs down the same virtual environment with no visual cues (a passive version). They also observe two sub-populations of cells that either peak (PyrUp) or trough (PyrDown) in their mean firing activity following the start of the cue initiated run. In a series of optogenetic manipulations, they show that disruption of SST interneurons modulates the integrity of PyrUp ‘coding,’ while PV interneurons regulate the integrity of PyrDown responsive cells. This is an interesting story and many of the experimental results are quite compelling. That said, I have several concerns that should be addressed in a revised manuscript.

The tie between PyrUp and PyrDown cells and path integration computations was tenuous to me and I don't feel like the connection between path integration, the cell response, and the behavior was ever super well explained. The authors

should attempt to beef up this portion of the manuscript and potentially include a theoretical/computational model that would explain why this form of tuning would be useful for this computation.

It is not entirely clear why these cells respond this way or how they differentiate from other ramping cell responses reported in the past. Since they are so strongly anchored to the run onset, a better analysis of the relationship between movement behaviors (e.g. acceleration) and the responsivity of these neurons is warranted. This includes a substantially more refined analysis of 'good' and 'bad' trials which I found to be a bit circular and lacking (there are significantly more criteria that one could use to assess task performance than the three main metrics included in this analysis).

The difference between the IGF response on passive versus active versions of the task is interesting, but many prior papers have shown robust place fields in animals navigating in darkness that are not actively required to perform path integration (i.e. are just rewarded on either side of the track or are free foraging in an open environment). The different here is perplexing and makes one wonder whether the primary results would hold in non-VR tasks. While I do not think it is appropriate to ask the authors to show that their results would hold in freely moving animals, I have to admit I find myself wondering why they used a task with head-fixation. At the very least, the authors should discuss the possibility of impoverished vestibular information on their main results.

Relatedly I would like to see more classic metrics of place field integrity explored here. For example, is spatial information disrupted in the passive task or just trial to trial reliability?

An explanation of the statistical test for classifying PyrUp and PyrDown cells should be included in the main manuscript since these classifications are so critical to the story.

(Remarks on code availability)

Version 1:

Reviewer comments:

Reviewer #1

(Remarks to the Author)

The authors have addressed all of my and the other reviewer's concerns. I have no other concerns and recommend publication.

(Remarks on code availability)

Reviewer #2

(Remarks to the Author)

The authors went to great lengths to carefully address all points I raised, as best as possible and in great detail, within the confounds of their existing experiments and datasets. I appreciate all the additional extensive analyses and clarifications. I feel like the claims have been strengthened and the manuscript structure is much improved. I recommend the manuscript for publication.

(Remarks on code availability)

Reviewer #3

(Remarks to the Author)

The authors have mainly addressed my concerns but some of the newly included analyses have raised additional issues/comments outlined below.

I agree with many of the concerns raised by reviewer 2, in particular the concerns about novelty. Ramping responses have been reported in CA1 (Ning, Bladon, and Hasselmo). I do not feel particularly convinced by the inclusion of a theoretical two-phase model regarding the role of UP and DOWN neurons in path integration nor their classification as distinct classes of neurons. There is no manipulation to test their theory nor a computational model showing that it is useful for self-localization via path integration.

I do not believe that the new GLM truly addresses concerns related to the separation of time, distance, and movement coding. These variables all tightly co-vary in the task setup and accordingly it is unclear if the GLM is appropriate from an analytical perspective. I feel only a manipulation of the relationship between reward delivery and distance traveled would truly disentangle things.

Relatedly, what are the GLM results on the passive task?

The authors suggest that there is a 'temporal dissociation' between peak firing and speed that indicates a decoupling of the two. I am not convinced by this argument - several studies have shown neurons can have temporally offset relationships to behavioral/sensory variables. At the very least this should be addressed in the discussion.

I still feel that the good vs bad trial analysis is circular (though I appreciate that the authors have addressed whether the sampling could explain differences).

"Despite clear differences in running speed, the firing rates of both PyrUp neurons and PyrDown neurons 5 remained largely stable, with PyrUp activity stability evident within the first three seconds (Figures 4C-4F)."

Speed-modulated cells in the extended hippocampal formation often saturate at higher speeds (Gois et al. 2018; Hinman et al., 2016) with greatest speed modulation happening at lower speeds. The results here are consistent with this and I don't find this to be an especially compelling argument against pyramidal up or down responses being correlated/anticorrelated responses to the initial acceleration on the track.

I don't understand the logic of the SRO analysis. Why doesn't the animal need to path integrate from the random stopping point to perform the task?

I remain confused by the immobile task results and I do not have a grasp on why IGFs would emerge in the immobile task but not in the passive task. This is especially confusing if IGFs in the immobile task were better aligned to the last lick of the previous trial which should be the case for the passive condition. What is the authors explanation?

"These changes in the decay/rise rate indicated that the PyrUp/PyrDown dynamics adjust to the animal's internal estimate of when sufficient time or distance has elapsed."

Again, I don't understand why there wouldn't be anticipatory responses similar to Up/Down in the passive task - the animal can still track time to reward after the gray screen appears right?

Minor comments.

"Good trials met the following criteria: animals came to a complete stop before trial run onset, allowing this onset to mark the start of integration; animals maintained largely uninterrupted running; and successfully obtained the reward (77.90±2.88% of all trials, see Methods). In contrast, bad trials failed to meet at least one criterion. Despite maintaining a mean running speed similar to good trials (Figure 3A, good trials: 48.12±0.84cm/s, bad trials: 45.20±0.84cm/s), bad trials resulted in a comparatively lower reward rate of 86.34 ± 4.36%" - The percentage reward rate seem off given the context of this sentence.

What is the relationship between late/early lick trials and high/low speed? My guess is that early licks systematically occur with deceleration - both metrics of the animals confidence?

Please include the figure legends with the figures in future revisions. Line numbers would also be useful.

(Remarks on code availability)

Version 2:

Reviewer comments:

Reviewer #3

(Remarks to the Author)

The author's have addressed my concerns. I appreciate their hard work, both in the original body of work and in their efforts to address my final comments. I commend them on an excellent addition to the field.

(Remarks on code availability)

We sincerely thank the editor and reviewers for their time and expertise. We appreciate the detailed and constructive comments and the opportunity to significantly enhance our manuscript. In response, we have undertaken substantial revisions to improve clarity, strengthen our conclusions, and incorporate extensive new analyses. We have now expanded the number of main figures from 6 to 7, and the number of supplementary figures from 5 to 15. Key improvements include:

- **Provided converging evidence for PyrUp/PyrDown responses reflecting internal time or distance integration processes, rather than simply locomotion:**
 - PyrUp/PyrDown neuronal activity is modulated by time/distance rather than simply by locomotion revealed by Generalized Linear Model (GLM) analyses.
 - PyrUp/PyrDown dynamics differ from speed profiles, and their responses peak earlier than running speed.
 - PyrUp/PyrDown dynamics remain similar between trials with high versus low running speed.
 - PyrUp/PyrDown neurons exhibit context-dependent responses, responding significantly stronger at the onset of trial-start runs than spontaneous runs occurring during a trial (new analyses included).
 - PyrUp/PyrDown activity patterns persist in an immobile timing task, which requires temporal integration without locomotion (new analyses included).
- **Provided more evidence that PyrUp/PyrDown responses correlated with behavioral performance:**
 - PyrUp neurons decay more slowly in trials with later first anticipatory licks, a behavioral proxy for the animal's internal estimate of elapsed time or distance.
- **Clarified PyrUp/PyrDown two-phase coding model:** We propose a two-phase coding scheme. This model describes a synchronized initiation phase, providing a reference point for integration, followed by a distance/time encoding phase characterized by heterogeneous decay/ramping rates.
- **Strengthened analysis of internally generated fields (IGFs):**
 - We clarified the criteria for IGF detection and provided further analysis comparing IGF presence and trial-by-trial activity correlation in path integration versus passive tasks.
 - We expanded the discussion on the absence of IGFs in passive tasks, emphasizing the combined impact of reduced sensory (including vestibular) input in head-fixed virtual reality settings and task demands.
- **Improved manuscript cohesion and readability:** We restructured the manuscript to improve the narrative flow and simplified both the main and supplementary figures to enhance readability.

- **Improved contextualization and novelty:** We expanded discussions and comparisons with prior studies to articulate the novelty of our findings, particularly regarding the presence and absence of the IGSs in our behavioral tasks and the functional relevance of two-phase dynamics of PyrUp/PyrDown populations.
- **Refined methodological details:** We provided additional information on depth estimation for neuron localization, cell-type classification, and methods for various analyses.

Once again, we thank the reviewers for their thoughtful comments, which have prompted numerous new analyses that have led to a significantly improved manuscript. These revisions reinforce our conclusion that PyrUp and PyrDown neurons represent an internally driven coding scheme supporting distance or time integration.

We hope this revised and improved manuscript will now be suitable for publication in Nature Communications.

Point-by-point responses to reviewers are included below, with reviewer comments retained in black and our responses added in blue. A revised manuscript with key revisions highlighted in red is included with the resubmission.

Reviewer #1 (Remarks to the Author):

In this manuscript, Heldman and colleagues present a compelling study that uses dense electrophysiological recordings in mice to investigate CA1 neural activity in a novel path-integration task adapted for a head-fixed preparation. In this task, mice are trained to run on a treadmill surrounded by screens, waiting for a cue to appear before running approximately 180 cm and licking to receive an otherwise hidden reward. The authors uncover a previously undescribed activity pattern in CA1 pyramidal neurons, where one subset increases firing following cue presentation at the onset of the run toward the goal (PyrUp), while another decreases (PyrDown). Importantly, they demonstrate that this activity pattern is not merely a reflection of task structure but instead correlates with cognitive demand, as evidenced by differences in firing rates between good and bad trials. They propose a mechanistic model in which PyrUp neurons initiate an integrative process (tracking distance or time), setting the stage for PyrDown neurons to guide accurate estimation of the goal location. Through elegant optogenetic manipulations targeting distinct interneuron subtypes (PV and SST), the authors establish causal relationships between these activity patterns and behavioral performance. Overall, this study is a tour de force. It integrates large-scale neural recordings, a cognitive task, and specific causal manipulations to reveal a previously uncharacterized circuit mechanism underlying distance estimation in path integration. The dataset is extensive and is combined with rigorous analyses and well-controlled experiments. Every potential logical gap is thoughtfully addressed, either experimentally or analytically.

It is evident that this manuscript has undergone substantial revisions and incorporates an impressive amount of additional data since its initial conception. I have no further data requests and only a minor analytical suggestion that could help clarify a small aspect of one result. The depth of the study (its strength) also means that the manuscript is dense, and I have a few minor suggestions to enhance clarity and readability. But these are primarily stylistic and should not impact the core findings. I have no major concerns regarding how data was collected processed and analyzed.

Comments:

1. The authors have examined the anatomical distribution of PyrUp and PyrDown neurons (deep vs. superficial) as reported in Figure S2F, but I could not find a methods description of how depth was estimated. Was this based on CSD, SWR power, or another approach? A “blunt” depth estimation using these metrics may not always yield reliable results and I’d like to suggest using a relative anatomical estimation among neurons recorded from the same shanks, relying on a spike amplitude metric (we describe this approach in Geiller et al., 2017, Nature Communications). Maybe the authors will find it useful. This pairwise comparison method could reveal more subtle differences, given the small size of the CA1 cell layer in mice (~50 μm).

Reply to Reviewer:

We thank the reviewer for the suggestion regarding the estimation of anatomical depth for PyrUp and PyrDown neurons. Our original approach relied on SWR power for depth estimation, following the methodology described in Mizuseki et al. (Nature Neuroscience, 2011). We acknowledge the potential limitations in precision with this method.

In response to the reviewer's suggestion, we have now implemented the relative anatomical depth estimation method proposed by Geiller et al. (Nature Communications, 2017). This approach utilizes pairwise comparisons of spike amplitudes from neurons recorded on the same shank, allowing for a more sensitive assessment of relative depth. Using this method, we observed a small difference in the estimated depth between PyrUp and PyrDown neurons (Figure S6F, mean difference: 1.56 μm ; median difference = 0 μm ; $n = 5150$ pairs; Wilcoxon signed-rank test, $p = 0.03$).

We have incorporated this new analysis in the section “**Subpopulations of pyramidal neurons display distinct responses around run onset**”:

“PyrUp and PyrDown neurons largely overlap in their laminar location within the CA1 pyramidal layer^{26,27} (Figures S6E and S6F, see Methods), implying that each subpopulation likely includes both genetically distinct deep and superficial CA1 neurons^{26,28–31}. ”

This is along with a description of the updated methodology, in the revised Methods section under “**PyrUp/PyrDown neuron laminar location within CA1 pyramidal layer**”.

Figure S6

2. This is subjective, but I suggest reducing the amount of information in the main figures, as the key message of each figure feels somewhat diluted. I found it difficult to parse the primary takeaways from individual panels. For example, the behavioral impact of interneuron manipulations is a critical finding but feels somewhat “buried” within the figures. Of course, the authors should feel free to disregard this suggestion if they strongly prefer their current structure.

Reply to Reviewer:

We appreciate the reviewer's constructive feedback on the figure presentation. To ensure the clarity of the primary findings, such as the behavioral impacts of interneuron manipulations, we have revised our main figures 5 to 7. Specifically, we have simplified these figures by relocating sequence-related analyses to supplementary figures 13, 14, and 16.

3. This is more a point of curiosity than a comment: in Figure 5M, the authors introduce cues on the track to shift the task from path integration to a goal-oriented behavior, yet they still observe PyrUp/PyrDown responses. Could the authors provide some insight into why this occurs? One possibility is that distance integration remains a default strategy because that is what the animals were initially trained to do?

Reply to Reviewer:

The reviewer raises an interesting point regarding the persistence of PyrUp/PyrDown responses even when salient external cues are provided. We concur with the reviewer’s intuition: a possible explanation is that after extensive training on the path integration (PI) task, which necessitates internal distance or time integration, animals may continue to employ this as a default strategy, with external cues as additional guidance.

This interpretation is further supported by the observation that behavioral parameters, such as running speed profiles and licking patterns, remained similar between the original PI task and the cue-rich condition (Figures S13G and S13H).

We have incorporated this discussion in the section **“Inactivating SST interneurons close to reward does not affect PyrUp response or task performance”**

“In this cue-rich context, animals maintained similar running speed and licking profiles (Figures S13G and S13H), and PyrUp/PyrDown responses persisted (Figures 6L and 6M), implying that time or distance integration remains a default strategy, supplemented by guidance from external cues.”

Figure S13

4. Regarding Figure 5N-S, have the authors checked whether the place cells detected in this condition are also IGS cells when cues are removed? I am not sure someone has ever looked at that?

Reply to Reviewer:

We thank the reviewer for this insightful question. Unfortunately, the experimental design of the recording sessions presented in Figures 5N-S (now Figures 6L-6O and S13D-S13F), which focused specifically on interneuron inactivation during the cue-rich task, did not include trials where cues were subsequently removed. As a result, we were unable to directly assess whether

the place cells active in the cue-rich condition also function as neurons with internally generated field (IGF) in the absence of cues within these particular experiments.

However, as noted in our response to point 3, the animals exhibited highly similar behavior—including running speed and licking patterns—across both the path integration (PI) and cue-rich tasks. This behavioral consistency suggests that animals may continue to rely on a default internal integration strategy even when external cues are available. Based on this observation, we hypothesize that there is likely some overlap between the place cells recruited in the cue-rich condition and the neurons with IGFs in the PI task.

This idea is supported by recent work from Qian et al. (Nature Neuroscience, 2025) and Sosa et al. (Nature Neuroscience, 2025), which demonstrated that during a goal-directed navigation task, a subset of hippocampal place cells maintains their position tuning irrespective of changes in the goal location.

Reviewer #2 (Remarks to the Author):

The authors describe time-cell-like spiking sequences in the mouse CA1 as well as two populations of pyramidal neurons - one that is excited and one that is inhibited - during a cued path integration task in mice. Given that these groups exhibit their peak excitation/inhibition at the same time (of peak speed?) and that they have variable decay/rebound rates, time can be decoded during path integration. By optogenetically inactivating SST or PV interneurons in transgenic mice during this behaviour, they then find that SST-silencing suppressed the first subpopulation and PV-silencing suppressed the second one.

The manuscript contains an impressive amount of work (electrophysiology, behavioural tests, analysis, closed-loop optogenetics) with extended analyses and clear, polished figures. Unfortunately, the results and conclusions are not particularly novel or convincing. The main issue stems from the task design where locomotion motifs make it very difficult to dissociate time from distance or exclude the strong possibility that neuronal activation is simply triggered by locomotion and does not necessarily time- or distance-integrate, blurring most conclusions.

The manuscript also suffers from lack of cohesion, making it feel like 2-3 small papers combined in one. E.g. Fig 1 is about distance dependence leading to a major conclusion that firing fields encode distance. Yet practically the rest of the paper is about time encoding and interneuronal circuits. Moreover, the analysis on firing fields and spiking sequences is mostly abandoned from Fig 3 onwards, and it is not even mentioned in the abstract.

Reply to Reviewer (Summary Response):

We thank the reviewer for their thorough assessment and critical feedback. We take seriously the concerns raised regarding the novelty of our findings, the potential confounding influence of locomotion, and the overall cohesion of the manuscript. We have undertaken substantial revisions and included significant new analyses to address these points directly.

Regarding the dissociation of time, distance, and locomotion encoding:

We acknowledge that the stereotyped locomotion in our path integration (PI) task presents a challenge in separating time and distance encoding in neurons with firing fields. We now explicitly address this challenge using Generalized Linear Model (GLM) analyses.

To more rigorously address whether PyrUp/PyrDown neuronal activity primarily reflects locomotion, we have incorporated several new analyses:

1. **Generalized linear models (GLMs):** We performed GLM analyses to quantitatively evaluate the relative contributions of time, distance, running speed, and licking to the activity of PyrUp/PyrDown neurons. These models indicated that while locomotion speed can modulate neuronal activity, the majority of neurons are significantly influenced by time or distance since run onset.

2. **Behavioral covariate analyses:** We presented multiple lines of evidence demonstrating that PyrUp/PyrDown activity does not simply reflect locomotion. This includes: A) Direct comparisons of firing rate profiles with speed profiles, revealing distinct peak timings. B) Comparisons of neuronal activity between trials with high versus low average running speeds, showing stable PyrUp/PyrDown dynamics despite speed differences. C) Analyses of trials segregated by early versus late anticipatory lick initiation, showing stable PyrUp/PyrDown responses in the first 2 seconds after run onset, despite differing speed profiles.

3. Immobile timing task: We provided new analyses from the immobile timing task, where animals performed temporal integration without locomotion. We demonstrated that animals performed integration by exhibiting anticipatory licking. We also provided justification for aligning neuronal activity to the last lick rather than the start cue. The presence of analogous PyrUp/PyrDown neuronal populations in this task supports the idea that these dynamic patterns are not solely dependent on movement.

Together with other evidence we provided supporting that PyrUp/PyrDown neurons encode time or distance, our data suggest that the PyrUp/PyrDown dynamics represent more than locomotion correlates—they encode internal processes relevant to temporal or spatial integration.

Regarding manuscript cohesion:

We appreciate the feedback on the manuscript structure and have worked to improve its narrative flow. We now clarify that the goal of Figure 1 was not to conclude that CA1 neurons encode distance, but to evaluate whether the PI task can serve as a suitable paradigm for studying internally generated hippocampal dynamics under conditions that require distance or time integration. Figure 2 further supported this point after aligning neuronal activity to self-initiated run onset as the potential start of distance or time integration.

The relatively low proportion of neurons involved in IGSs then motivated the broader population-level investigation and led to the identification of PyrUp/PyrDown populations in Figure 3 and beyond.

By restructuring the manuscript's framing and clarifying the progression of results, we aim to present a more cohesive and logically connected narrative that builds toward our central conclusion: PyrUp and PyrDown neurons represent a novel population-level coding mechanism for internal integration processes, shaped by but not simply reflecting motor output.

We address the reviewer's specific comments in detail below.

Major points:

- Internally generated spiking sequences, triggered by run onset and integrating distance is not a novel observation. They have been described before by Villette, Malvache et al Neuron 2015. The authors should discuss their Fig1-2 analyses in relation to those findings.

Reply to Reviewer:

We thank the reviewer for highlighting the work by Villette et al. (Neuron, 2015) on internally generated sequences (IGSs). The IGSs observed in our current study differ in several aspects: task design, behavioral demands, and the characteristics of the observed sequences. We now elaborate on this in the revised manuscript to clarify the differences.

1. Task demands and behavioral relevance:

Our PI task explicitly required animals to integrate distance (or elapsed time, given the stereotyped running) to receive a reward. This created a demand for integration. In contrast, the

task used by Villette et al. did not impose explicit integration requirements, and no reward was involved in the behavior task. These differences imply distinct motivational states and task demands.

2. Relationship between sequences and behavior:

In our study, the observed IGSs were consistently aligned with behaviorally meaningful events — the self-initiated running toward the reward zone within individual trials. While Villette et al. also reported sequences often beginning at run onset, these sequences tended to recur multiple times within a single running bout, and they were not locked to a structured trial-by-trial behavioral epoch. The authors characterized their sequences as "internally **recurring** hippocampal sequences" that encode traveled distance but without a consistent relationship to ongoing behavior.

3. Sequence length:

Villette et al. reported a broad distribution of sequence lengths, from very short (<1 cm) to over 100 cm, further underscoring their potential dissociation from discrete, structured behavioral epochs. In our task, the sequences unfolded with an alignment to the structured locomotion as animals traversed the ~180 cm track segment.

This is now included in Discussion under the section "**Behaviorally aligned IGSs from run onset**".

The primary aim of Figures 1 and 2 is to validate our PI task as a suitable paradigm for studying internally generated hippocampal dynamics under conditions that require distance or time integration. Specifically, Figure 1 demonstrates that our PI task elicits a significantly greater proportion of neurons with internally generated firing fields (IGFs) compared to a passive control condition lacking such integration requirements. Figure 2 further establishes the self-initiated run onset as the putative start of integration: aligning neuronal activity to run onset both confirms the presence of IGF-expressing neurons in the PI task and highlights their near absence in the passive task. Additional analyses in Figures 3C and S5C confirm that these sequences exist in both the time and distance domains.

These findings are consistent with prior work suggesting that IGSs emerge more prominently in contexts relevant to memory or temporal integration (e.g., Pastalkova et al. (Science, 2008); Kraus et al. (Neuron, 2013)). Furthermore, our observation that only a relatively small fraction of CA1 pyramidal neurons (~10% when aligned to run onset) exhibited IGFs motivates the broader population-level analyses and leads to the identification of PyrUp and PyrDown dynamics presented from Figure 3 onwards.

We have now incorporated this discussion in the section "**A behavioral task that requires distance integration generates IGSs**",

"To investigate neuronal dynamics during path integration, we developed a virtual reality (VR)-based behavioral task that required head-fixed mice to accumulate self-motion to estimate distance — referred to as a path integration (PI) task (Figure 1A)."

and

"These results also validated the PI task as a suitable paradigm for investigating the internally generated hippocampal dynamics under conditions that require distance or time integration."

And the section **“The initiation of an IGS aligns with self-initiated onset of running”** (the second paragraph).

- The biggest issue with the manuscript is that time and distance cannot be easily separated because locomotion patterns seem to be very stereotyped. Even though the authors go to some lengths to rule out that locomotion is driving their findings, it is not convincing. For example, PyrUp/PyrDown neurons may simply be correlated/anti-correlated with locomotion, respectively. Since locomotion is stereotypical peaking at a more-or-less fixed timepoint after run-onset, they all peak or dip at that timepoint after run-onset. This (anti)correlation is better in “good” trials because a good locomotion pattern is required there (it is a criterion for selecting “good” trials). In this case, neurons that are either correlated or anticorrelated with locomotion would not constitute “two previously unknown functional subpopulations of CA1 pyramidal neurons”.

Related to this, running speed is only plotted against distance in Fig 1 and only against distance. It would be essential, to show locomotion over time, to match with the firing rate plots. It is possible that locomotion and rates strongly correlate.

Reply to Reviewer:

We thank the reviewer for raising this critical point, as rigorously dissociating neural activity related to internal computations from activity merely reflecting locomotion is central to our study's claims. We acknowledge the challenge posed by stereotyped locomotion in our task. To address this comprehensively, we have performed multiple new and expanded analyses, which we believe provide evidence that PyrUp/PyrDown dynamics are not simply epiphenomena of locomotion.

1. Comparison of PyrUp/PyrDown firing and running speed profiles:

We now explicitly plot running speed over time alongside PyrUp/PyrDown firing profiles. Visualizing running speed with raster plots of example PyrUp neurons from the same trials revealed that PyrUp neurons consistently peak within ~1 second of run onset, whereas running speed typically peaks later, around ~2 seconds (Figures S8C and S8D).

Across all recordings, the average peak timing of PyrUp neurons (Median \pm SEM: 1.24 ± 0.02 s) and PyrDown neurons (1.21 ± 0.03 s) occurred significantly earlier than the peak of running speed (1.69 ± 0.08 s; $p = 1.1e-7$ for PyrUp vs. Speed; $p = 2.9e-8$ for PyrDown vs. Speed, Wilcoxon rank-sum test, Figures S8E-S8H, left column). This temporal dissociation suggests that PyrUp/PyrDown activity is not simply a direct readout of the speed profile.

In “bad” trials (where animals fail to integrate accurately), the PyrUp peak time was no longer distinguishable from the speed peak time (PyrUp: 1.36 ± 0.06 s; Speed: 1.32 ± 0.28 s; $p = 0.49$), while PyrDown neurons still peaked significantly earlier (0.93 ± 0.14 s; $p = 0.01$, Figures S8E-S8H, right column). This differential relationship with behavioral performance further argues against a simple locomotion correlate.

Figure S8

2. Impact of running speed variation (high vs. low speed trials):

We categorized trials based on the mean running speed during the initial 4 seconds post-run onset into "high speed" (>45 cm/s) and "low speed" (35–45 cm/s) groups (2199 PyrUp neurons, 1246 PyrDown neurons, 84 recordings, 41 animals; recordings required at least 15 trials in each category, only considering good trials; Figures 4C-4F).

Despite significant differences in speed profiles across these categories (particularly evident when binned over 1-second intervals from 0 to 4 seconds), the activity of PyrUp neurons did not significantly differ between high and low speed trials within the first 3 seconds post-run onset. Similarly, PyrDown neuron activity remained stable throughout this period across speed categories. These results indicate that the characteristic firing rate profiles of PyrUp and PyrDown neurons are largely independent of variations in running speed.

Figure 4

3. Connection to licking behavior (early vs. late lick trials):

To test whether PyrUp/PyrDown activity reflects the animal's internal estimate of elapsed time/distance, we examined its relationship to the timing of the animal's first anticipatory lick. In the PI task, anticipatory licking serves as a behavioral proxy for animals' expectation of reward, which is contingent on time or distance integration. We divided trials into "early lick" (first lick < 2.5 s) and "late lick" (first lick > 3.1 s) categories (492 PyrUp neurons, 327 PyrDown neurons, 18 recordings, 15 animals; recordings required at least 15 trials per category, only considering good trials; Figures 4G-4J).

PyrUp neuron activity remained similar across these groups during the first 2 seconds. In addition, we observed a significant increase in the decay time constant for PyrUp neurons in late lick trials (early lick: Mean \pm SEM: 1.89 ± 0.05 s; late lick: 2.00 ± 0.05 s; $p = 0.036$, Wilcoxon rank-sum test). This result implies a link between the PyrUp decay and lick timing. In comparison, PyrDown neurons showed no significant changes in either their firing rate profiles or their rise time constants (early lick: 2.34 ± 0.22 s; late lick: 3.02 ± 0.26 s; $p = 0.27$, Wilcoxon rank-sum test).

In contrast, running speed profiles differed significantly between early and late lick trial categories. The stability of PyrUp/PyrDown responses in the first 2 seconds in the face of these speed differences further suggests they are not mere reflections of locomotion.

Figure 4

4. CA1 SST interneuron inactivation:

Bilateral CA1 SST interneuron inactivation did not affect the mean running speed. When plotting the running speed over time, we observed a brief reduction in running speed, localized to the 2–3 second window post-run onset. However, unilateral SST inactivation (a weaker perturbation used during electrophysiological recordings) led to significant reductions in PyrUp neuron activity throughout the 0–3 second window (Figure S12J-S12L). If the reduction in PyrUp firing merely reflected altered locomotion, we would expect their temporal profiles of change to align more closely. This divergence between the minor effect on speed and the more pronounced neuronal effect further supports that PyrUp dynamics may not be solely reflective of locomotion patterns.

Figure S12

5. Generalized linear model (GLM) analyses:

To quantitatively disentangle the contributions of various factors, we employed a GLM approach (based on Kraus et al. (Neuron, 2013)). For each PyrUp and PyrDown neuron, we modeled its firing activity using predictors including time since run onset, distance traveled since run onset, running speed, licking behavior, and the neuron's recent spiking history.

By comparing the performance of the full model to reduced models (each omitting one category of predictors), we estimated the unique contribution of each predictor. The results showed that a majority of PyrUp neurons (63.9%) and PyrDown neurons (64.6%) were not significantly modulated by running speed. Similarly, most were not significantly modulated by licking (PyrUp: 69.2%; PyrDown: 71.2%) (Figure S7A and S7B).

In contrast, nearly half of PyrUp (49.2%) and PyrDown (49.7%) neurons were significantly modulated by time, and a similar proportion by distance (PyrUp: 50.5%; PyrDown: 49.5%). However, there was no clear population-level preference between time and distance (PyrUp: $p = 0.10$; PyrDown: $p = 0.53$, Wilcoxon rank-sum test), reflecting their high correlation in the PI task (Figure 4A and 4B). These GLM findings provide quantitative support that PyrUp and PyrDown activity is preferentially driven by internal task variables (time or distance) rather than simply reflecting external motor outputs.

Figure S7

Figure 4

A T(12.5%) D(13.8%) TD(36.7%) p(0.10)

B T(13.3%) D(13.1%) TD(36.4%) p(0.53)

6. Immobile integration task (detailed in a subsequent response):

To provide more direct evidence against the necessity of locomotion, we presented more evidence to demonstrate PyrUp/PyrDown neuron activity in an immobile task requiring temporal integration (Figure S10). The presence of analogous PyrUp/PyrDown neuronal populations in this non-locomotor context demonstrates that these firing patterns are not dependent on locomotion.

Summary:

While hippocampal dynamics are influenced by locomotion, our extensive set of analyses—spanning temporal dissociations, behavioral covariate modulations, optogenetic manipulation, quantitative GLM modeling, and evidence from an immobile task—collectively demonstrates that the PyrUp and PyrDown subpopulations exhibit structured activity patterns that are not merely correlated or anti-correlated with stereotyped running.

Instead, multiple lines of evidence support that **PyrUp/PyrDown responses reflect time or distance integration**: (1) the GLM analyses demonstrated significant contributions of time and distance to neuronal activity; (2) PyrUp/PyrDown neurons were present in the immobile timing

task; (3) PyrUp/PyrDown responses were more specifically tuned at the trial-start run onset instead of spontaneous run onset (more evidence provided in a subsequent response); (4) Bayesian decoding confirmed that PyrUp population activity effectively encodes time passage (in the original manuscript).

We further provided evidence that **PyrUp/PyrDown responses correlated with behavioral performance**: (1) The strength of responses around run onset correlated with behavioral accuracy, being more pronounced in trials where animals correctly estimated time or distance than in inaccurate trials (in the original manuscript). (2) PyrUp neurons decayed more slowly in trials with a later first anticipatory lick. In the PI task, anticipatory licking serves as a behavioral proxy for animals' expectation of reward, which is contingent on time or distance integration.

We have substantially revised the manuscript to incorporate these new analyses.

Please find these changes in the sections "**PyrUp/PyrDown responses reflect internal integration, not solely locomotion**", "**PyrUp/PyrDown responses likely predict reward-anticipatory behavior**", and "**Inactivating SST interneurons after run onset impairs PyrUp response, IGSs, and task performance**".

We have also included in Discussion "**The relationship between PyrUp/PyrDown responses, distance/time integration and behavioral performance**".

Method descriptions can be found in Methods under "**Comparing PyrUp/PyrDown responses across high and low speed trials**", "**Comparing PyrUp/PyrDown responses across early and late first lick trials**", and "**Generalized Linear Model (GLM)**".

- Given these facts, claims like "IGS that encodes the moment-to-moment running distance" seem unconvincing, since throughout the manuscript it never becomes clear whether distance or time is being encoded. If anything, sequences seem clearer when plotted over time rather than distance, but this is not adequately explored.

Reply to Reviewer:

We agree with the reviewer that the stereotyped running speed in our path integration (PI) task makes it inherently difficult to definitively distinguish whether individual neurons or sequences are encoding distance versus time. Our previous phrasing may have overemphasized distance encoding.

To address this directly, we performed GLM analyses for neurons exhibiting firing fields (IGF neurons). These models aimed to quantify the influence of both time and distance traveled since run onset on neuronal firing rates (Figure S3A).

Our GLM results for IGF neurons indicate that a substantial proportion is significantly modulated by time (53.8%) and a similar proportion by distance (56.4%). Many individual neurons showed significant contributions from both time and distance (42.1% conjunctively tuned), while smaller subsets were predominantly tuned to time (11.7%) or distance (14.3%) exclusively. At the population level of IGF neurons, there was no statistically significant difference in the overall contribution of time compared to distance ($p = 0.13$, Wilcoxon rank-sum test). This result supports

the notion that, due to the task structure, time and distance encoding are highly correlated and difficult to disentangle.

We have now included these results in the section “**The initiation of an IGS aligns with self-initiated onset of running**” (the fourth paragraph). Method description is included in Methods under “**Generalized Linear Model (GLM)**”

To avoid overstating the specificity of encoding, we have revised the sentence highlighted by the reviewer to: “*These neurons collectively formed an IGS that unfolded as the animal progressed through the cue-constant segment (Figure 1G).*” (in the section “**A behavioral task that requires distance integration generates IGSs**”)

Figure S3

A T(11.7%) D(14.3%) TD(42.1%) p(0.13)

• Total lack of any fields in the passive task is interesting and quite surprising and needs further examination. This result may be driven by 2 issues:

- 1) It may be biased by a specific threshold the authors used to determine when a cell has a significant field. Looking into the Methods, they use a long list of criteria for field detection “based on visual inspection”, resulting in only 4% cells having IGFs! One of them is: “minimum mean trial-by-trial Spearman correlation of the firing rate profile > 0.15”. Looking into Fig 2H, it seems that even IGF cells have an average correlation of just under 0.15 when aligning to the cue. Unless I am reading this wrong, it could be that the 0.15 criterion is just right to exclude all cells in the passive task. These cells may still yield more noisy fields than in the IP task, but a different threshold would probably reveal many of them still having a field.

Reply to Reviewer:

We appreciate the reviewer’s careful examination of our field detection methodology. We address the specific concerns as follows:

1. Clarification of field detection thresholds:

We have revised the Methods section to provide greater clarity on the application of our field detection criteria. The threshold of "minimum mean trial-by-trial Spearman correlation > 0.15" was specifically applied when identifying firing fields with activity plotted against distance in Figure 1.

For identifying firing fields with activity plotted against time (aligned to run onset, as in Figure 2), we employed a different set of Spearman correlation thresholds: 0.12 or 0.09—depending on the accompanying threshold for temporal information.

This is now explicitly stated in Methods under ***"Firing field identification"***.

2. Clarification of Figure 2H:

We have clarified that the correlations shown in Figure 2H were not used as selection criteria. Instead, IGFs were first identified based on their activity profiles when aligned to run onset. Figure 2H then illustrated a comparison: for this selected population of IGF neurons, we showed how their trial-by-trial correlation changes if their activity was instead aligned to cue onset. The observation that correlations were lower when aligned to the cue (often below 0.15) supports our argument that run onset is a more behaviorally relevant starting point for these sequences.

This clarification is now included in Methods under ***"Firing field identification"***.

3. Activity correlations in the passive task:

To directly address whether a different threshold might reveal fields in the passive task, we systematically examined trial-by-trial correlations.

When neuronal activity was plotted against distance (as in Figure 1), neurons in the passive task consistently showed significantly lower mean activity correlation values compared to the PI task, with the vast majority falling well below 0.1 (as shown in Figure S1F).

Similarly, when activity was plotted against time (aligned to run onset, as in Figure 2), most neurons in the passive task still exhibited correlation values below 0.1 (Figure S2E).

The few neurons in the passive task that did show higher correlations (outliers) were visually inspected, and they did not exhibit structured, unimodal firing fields characteristic of IGFs. This result indicates that high correlation alone does not reliably predict field presence (Figure S2F).

This is now included in the section ***"The initiation of an IGS aligns with self-initiated onset of running"***: *"However, even after run-onset alignment, IGFs remained nearly absent (Passive: 0.19 ± 0.19% of neurons; Figure 2G), with only one IGF detected across all passive recordings (Figure S2G; see Methods). Pyramidal neurons in the passive task also showed low trial-by-trial activity correlations (Figures S2E and S2F)."*

Figure S2

4. Clarification on parameter tuning:

We clarified that we did not selectively tune parameters to exclude firing fields in the passive task. We first selected parameters to ensure accurate field detection in the PI task. Applying the parameters used for field detection in the PI task after run onset alignment, we detected one neuron with a firing field in the passive task when activity was aligned to run onset.

This is illustrated in Figure S2G above and further detailed in the section **“The initiation of an IGS aligns with self-initiated onset of running”**: *“However, even after run-onset alignment, IGFs remained nearly absent (Passive: $0.19 \pm 0.19\%$ of neurons; Figure 2G), with only one IGF detected across all passive recordings (Figure S2G; see Methods).*

5. Low percentage of neurons with IGFs in the PI task:

The observation that only ~4% of CA1 pyramidal neurons exhibit IGFs when plotted by distance (and ~10% when aligned to run onset and plotted by time) is indeed a relatively low proportion. However, this is consistent with several lines of evidence from previous literature: A) Reduced spatial selectivity and fewer place cells are commonly reported in virtual reality (VR) and head-fixed preparations compared to freely moving animals (e.g., Aghajani et al. (Nature Neuroscience, 2015); Chen et al. (eLife, 2018)). B) Spatial tuning is further attenuated in cue-poor environments (e.g., Bourboulou et al. (eLife, 2019); Sharif et al. (Neuron, 2021)). Our tasks, with minimal visual cues and stereotyped treadmill running, represent such a cue-poor environment. C) The proportion of neurons participating in IGSs can be low, particularly under low-demand tasks. For instance, Villette et al. (2015) noted that “in several instances (37 out of 65 imaging sessions from five mice, see Figure S1G), no recurring pattern was detected”.

The discussion on factors that can influence the generation of IGS is now included in Discussion under **“Factors that influence the expression of IGS”**.

6. Lack of neurons with IGFs in the passive task:

Furthermore, the lack of fields in the passive task aligns with the work by Pastalkova et al. (Science, 2008) and Hirase et al. (European J. Neuroscience, 1999), who also reported an absence of clear, sequentially activated firing fields when freely moving rats were running on a wheel without specific integration demands (see their Fig. 3A), a condition analogous to our passive task.

The discussion about the absence of IGFs in the passive task is now included in Discussion under **“Factors that influence the expression of IGS”**.

Together, these points suggest that the lack of IGFs in our passive condition likely reflects the combined effects of the head-fixed VR environment, the sensory sparseness of the passive task, and the lack of a behavioral requirement for distance/time integration.

- 2) Another issue is how firing rates are aligned across trials. Based on Fig 1D, in the passive task mice run with more homogeneous speeds, whereas they accelerate more slowly in the PI task after each reward. This is probably because the authors use a variable ITI defined as follows: “The next trial started at least after 0.5s and until the animal’s speed has exceeded 30 cm/s for more than 0.3s and the time since the last lick has exceeded 0.3s.”. It is unclear why these criteria have to be met to start counting distance travelled by the mouse but in any case, mice are already running when the trial is initiated (meaning distance starts being counted). Without a fixed “landmark” in the passive task (like the cue in the IP task) and with mice already running, using distance as a metric to find IGFs in the passive task is not a fair comparison to the IP task. Distance from what? Later, in Fig

2G, they check for IGFs from run-onset, but it is not clear if that run onset is within the trial or includes run-onset during the ITI. Run-onset after reward licking is probably a better way to search for fields in the passive task.

Reply to Reviewer:

We thank the reviewer for raising this point regarding trial alignment and the definition of "run onset."

First, we have clarified the rationale behind our use of a variable inter-trial interval (ITI). The ITI criteria ("the next trial started at least 0.5 seconds after the previous trial and only when the animal's speed exceeded 30 cm/s for more than 0.3 seconds and the time since the last lick exceeded 0.3 seconds") were designed to introduce variability in trial initiation. This approach aimed to discourage animals from relying on a fixed distance or temporal structure to anticipate reward delivery.

This rationale is now added in the section "**A behavioral task that requires distance integration generates IGSs**": "*To further discourage distance or time integration, we introduced randomness in the inter-trial interval (see Methods).*"

We also clarified it in Methods under "**Behavioral training**".

Second, we agree that the lack of a defined landmark in the passive task complicates the identification of a reference point for measuring distance. Similarly, in the PI task, although a visual cue marked the trial onset, our analyses suggest that animals began integrating distance or time from the onset of self-initiated locomotion rather than from the cue (Figure 2C,D). That is, using the task-defined trial onset as a reference (Figure 1) would be inconsistent with animals' behavior in both tasks.

However, in Figure 2G, we realigned neuronal activity to run onset to better reflect the behavioral onset of distance/time integration. As shown later in the response to the speed-matching comment, after this alignment, IGFs remained largely absent in the passive task (Figure 2G). To rule out running speed as a factor in IGF generation, we analyzed PI task recordings with running speed matched to the passive task. Even when running speed profiles were comparable, the percentage of neurons with IGF remained significantly higher in the PI task (Figure S2H-S2J).

These results are now included in the section "**The initiation of an IGS aligns with self-initiated onset of running**" (the third paragraph).

Third, we now clarify that in both the PI and passive tasks, "run onset" was defined as the time when animals initiated sustained locomotion following the completion of the previous trial's traversal (typically at 180 cm). This approach ensured a consistent behavioral reference point across tasks and avoided reliance on reward delivery, which was sometimes absent in the PI task due to missed rewards.

This is now included in Methods under "**Alignment with sensory or motor cues**".

- Authors claim that spiking sequences in "bad" trials are more noisy. This could simply be due to undersampling (only 12% of trials are "bad"). IGFs are typically noisy, requiring averaging across many trials to see the field, so using only a small sample of trials could lead to blurrier fields. The authors should repeat their comparisons after subsampling the "good" trials (with multiple iterations) to match the "bad" trials.

Reply to Reviewer:

The reviewer raises a valid concern regarding the potential impact of differing trial numbers on the perceived noisiness of IGFs in "bad" versus "good" trials. We agree that undersampling can contribute to apparently blurrier fields. To address this, we have performed the suggested subsampling analysis.

First, we have now explicitly stated in the Methods section that all analyses comparing "good" and "bad" trials were restricted to recording sessions that contained a minimum of 15 "bad" trials. This ensures a baseline level of sampling for the "bad" trial category.

Second, to directly control for the effect of trial count, we performed a random subsampling procedure. For each recording session included in this analysis, we randomly selected a subset of "good" trials equal in number to the "bad" trials from that same session.

Using these trial-count-matched subsets, we recalculated the trial-by-trial activity correlations (Spearman correlation of firing rate profiles across trials) for IGF neurons, which served as a measure of field reliability.

This entire subsampling and correlation recalculation process was repeated five times to ensure the robustness of the findings and account for variability in the random sampling.

Due to space constraints, representative results from three iterations of the subsampling were shown (Figure S5B). The results of this subsampling analysis demonstrate that even when the number of "good" trials was matched to the number of "bad" trials, the trial-by-trial correlations for IGFs remained significantly higher in the "good" trials compared to the "bad" trials. This indicates that while trial count can influence the apparent clarity of averaged fields, the reduced reliability of IGFs during "bad" trials persisted beyond simple undersampling effects.

These results are now included in the section "**Accurate integration is correlated with the expression of IGSs**": *“Correspondingly, in bad trials, neurons with IGFs showed reduced trial-by-trial activity correlation (Figure 3D), and the IGSs were impaired (Figures 3C, S5A). This impairment persisted after random subsampling good trials to match the number of bad trials (Figure S5B, see Methods)”*.

The method description is included in Methods under "**Comparing firing fields between good and bad trials**".

Figure S5

• The authors show that they can decode time based on firing rate decay rates of PyrUp/PyrDown. But any time series with fixed temporal profile can be used for time decoding. It does not convincingly show that these cells encode time, rather than just encode locomotion (see comment further up). The authors claim that spontaneous runs don't yield similar responses, but that is probably because spontaneous runs do not share the stereotyped run motifs of regular runs. They maybe short, jerky or slow crawls (the locomotion traces are not shown in Fig S3A-B after all). The locomotion-dependence is supported by the analysis in Fig S3C: when comparing only spontaneous runs with matched-speed to regular runs, the firing rates become much closer, further supporting that locomotion may be underlying these responses.

Reply to Reviewer:

We agree with the reviewer that Bayesian decoding of time from PyrUp/PyrDown population activity, while demonstrating that temporal information is present in these patterns, does not establish that these neurons encode time independently of locomotion or other covariates.

However, our analyses presented in the earlier reply support that PyrUp/PyrDown activity is not merely reflecting locomotion (as detailed in our response to the reviewer's second major point). These analyses include:

- The earlier peak of PyrUp/PyrDown activity compared to peak speed.
- The stability of PyrUp/PyrDown firing rate profiles across high vs. low speed trials.
- The stability of PyrUp/PyrDown firing rate profiles in the first 2 seconds across early vs. late lick trials, where the speed profiles differed significantly.

- D) The GLM results demonstrating the contributions from time/distance in addition to speed/acceleration.
- E) The presence of PyrUp/PyrDown dynamics in the immobile timing task.

To address the concerns with spontaneous runs:

1. Locomotion traces for spontaneous runs:

We have now included example locomotion traces (speed over time) from three representative recordings (three animals) in the revised supplementary figure (Figure S9C). Each example showed three trial-start runs and one speed-matched spontaneous run occurring in the middle of the second trial.

These traces were overlaid with the population-averaged PyrUp neuron activity from the same trials. Despite comparable speed profiles in terms of both magnitude and duration, robust PyrUp responses were observed during trial-start runs and were reduced or absent during spontaneous runs.

2. Definition of "matched-speed" spontaneous runs (clarification for Fig S3C):

We have revised the Methods to provide a precise definition of how "matched speed" was determined for the analysis in Figure S3C (now Figure S9E). Specifically, for each recording, we calculated the mean speed and standard deviation of trial start runs during -1.5 to 0 s and 0 to 1.5 s relative to the run onset. Spontaneous run bouts in the same session were considered speed-matched if they fell within one standard deviation of the mean speed for both time windows. Recordings were included if there were >15 of both types of qualifying runs.

This is now included in Methods under "***PyrUp and PyrDown neurons with respect to running bouts in the PI task***".

3. Direct neuron-level comparison:

To evaluate whether speed-matched spontaneous runs elicited comparable PyrUp/PyrDown responses, we further compared the responses of individual PyrUp/PyrDown neurons aligned to trial-onset versus spontaneous run-onset (8 recordings, 6 animals, 775 neurons, 479 run bouts). Results suggest a reduction in run-onset response during the spontaneous runs, even after subsampling the trial start runs to match the number of spontaneous runs per recording (Figure S9D).

4. Duration of spontaneous runs:

Our analysis shows that the average duration of speed-matched spontaneous run bouts is 2.81 ± 0.05 s (mean \pm SEM), while the trial-start run length is 3.83 ± 0.04 s. Collectively, our results suggest that many spontaneous runs were not simply very short and jerky movements, yet still failed to elicit the full PyrUp/PyrDown response.

Summary

While locomotion can modulate hippocampal activity, the significant attenuation of PyrUp/PyrDown responses during speed-matched spontaneous runs—coupled with the other evidence presented earlier—argues that these neuronal signatures are not solely driven by locomotion kinematics. Instead, they appear to reflect internally generated dynamics that are associated with the distance/time integration relevant to the PI task.

We have included these results in the section “**PyrUp/PyrDown responses reflect internal integration, not solely locomotion**” (the fifth paragraph).

Figure S9

In fact, it is not clear how matched-speed is defined. In the Methods it is stated: “We found recordings with matching running speed between the passive and PI tasks, and identified 4 animals (10 recordings in total) where the mean speed did not differ from the passive task recordings”. What is the statistical criterion behind “did not differ”? Please clarify.

Reply to Reviewer:

We have now revised the Methods section to explicitly state the criteria used for speed matching in the spontaneous run analysis.

1. Speed-matching for spontaneous run analysis (related to the previous point and Figure S3C (Now Figure S9E)):

Please find the detailed criteria in our immediately preceding response.

2. Speed-matching between passive task recordings and PI task recordings (related to Figures 1D, 1E, and 1H analysis):

The quote the reviewer refers to (“We found recordings with matching running speed between the passive and PI tasks...”) is related to a different comparison: selecting a subset of recording sessions from the PI task whose overall average running speed was comparable to the overall average running speed observed in passive task recording sessions. This was done to address a potential confound in the analysis of firing field prevalence (Figure 1H). Our rationale was: Our data suggest that animals tended to run, on average, faster in the PI task compared to the passive task. To ensure that the higher prevalence of IGFs in the PI task (Figure

1H) was not simply due to animals running faster, we sought to compare PI and passive sessions where this speed difference was minimized.

To do so, we selected PI task recording sessions where the mean running speed did not significantly differ from the passive task (4 animals, 10 recordings). Figures 1D and 1E in the original manuscript showed the speed profiles from these recordings. Using this subset of speed-matched PI and passive sessions, we re-analyzed the percentage of neurons exhibiting firing fields (as shown in Figure 1H). The percentage of IGFs remained significantly higher in the PI task compared to the passive task (Figure S1E).

Furthermore, when trials from these speed-matched sessions were aligned to run onset, the speed profiles matched between the tasks, while anticipatory licking remained absent in the passive task (Figures S2H-S2I). And the percentage of neurons with IGF remained significantly higher in the PI task (Figure S2J).

These results are now included in the sections “**A behavioral task that requires distance integration generates IGSs**” and “**The initiation of an IGS aligns with self-initiated onset of running**” (the third paragraph).

The method description is included in Methods under “**Behavior analysis**”.

Figure S2

- The immobile task is not very clear either. Are mice supposed to not lick during the delay or can they be constantly licking throughout the 4 sec? Lick rates are not shown. If it is the latter case, how do the authors know they time integrate? It is also unclear why the authors align firing rates by the last lick of previous trials, rather than cue onset or offset, which is a specific temporal anchor in a trial.

In the immobile task, the increase/decrease of firing in PyrUp/PyrDown seems to happen during the delay period and may be related to timing or simply triggered by the cue, or even correlated

with licking. In either case, it still doesn't convincingly exclude that locomotion triggers UP/DOWN responses in the normal task.

Reply to Reviewer:

We appreciate the reviewer's request for more clarity on the immobile task. We have substantially revised the description and added new data to address these points.

Licking behavior during the delay period:

We have now included lick rate plots for the immobile task in a new supplementary figure (Figure S10B). These plots demonstrate that once well-trained, mice learned to suppress licking for the majority of the delay period and then initiated a bout of licking primarily around the time the reward was expected. This anticipatory licking pattern supports that the animals were integrating time to guide their reward-seeking behavior.

Rationale for aligning neuronal activity to the last lick of the previous trial:

We chose to align neuronal activity to the last lick of the previous trial rather than cue onset or offset for two reasons:

1. Temporal dynamics of PyrUp/PyrDown neurons:

We analyzed the firing rate profiles of neurons identified as PyrUp or PyrDown when aligned to either the previous last lick or cue onset (the definition of PyrUp/PyrDown followed that in the PI task). We found that aligning to the last lick of the previous trial revealed clear and rapid increases (PyrUp) or decreases (PyrDown) in firing shortly after this event (Figure S10D-S10F). In contrast, when aligned to cue onset, neurons often appeared to be already partway through their activity ramp (Figures S10G-S10J), indicating that the cue itself was not the primary trigger for the initiation of the ramping activity.

2. Empirical evidence for timing onset:

In an experiment not included in the manuscript, we occasionally omitted the visual start cue. We found that well-trained animals could still perform the timing task accurately, exhibiting anticipatory licking appropriately, even without the explicit cue. This result suggests that the cue is not necessary for initiating time counting.

PyrUp/PyrDown firing pattern similarity with the PI task:

PyrUp and PyrDown neurons identified in this immobile task exhibited firing patterns similar to those seen in the PI task: an initial rapid increase/decrease in firing starting shortly after the "timing onset" (aligned to previous trial's last lick), with the averaged activity reaching peak/bottom at 1.68s (PyrUp) and 1.50s (PyrDown) from the previous trial's last lick. Then these neurons showed a slow decay/increase that spanned the delay period. This similarity in the form of the neuronal response across a locomotion task and a non-locomotion task supports the idea that these subpopulations are recruited in contexts requiring distance or time integration, independent of locomotion.

We have added these clarifications in the section "**PyrUp/PyrDown responses reflect internal integration, not solely locomotion**" (the sixth paragraph).

Figure S10

- The authors use a method for identifying non-tagged putative SSTs and PVs. Even though their metrics look similar to those from corresponding tagged cells, it is unclear what they look like for other cell types (e.g. pyramidal cells). Why not just analyze tagged cells? Is it a sampling issue?

Reply to Reviewer:

The reviewer is correct; the number of optogenetically tagged SST and PV interneurons was limited, which poses a sampling challenge for robustly characterizing their population activity or their influence on pyramidal cells (as shown in Figure S14E). This limited sample size was the primary motivation for developing a method to identify additional putative SST and PV interneurons from the broader population of untagged cells.

We have clarified this procedure more explicitly in the revised Methods section. In brief, all recorded neurons were first broadly classified as either putative pyramidal cells or putative interneurons based on established electrophysiological criteria, including average firing rates, spike waveform, and auto-correlogram characteristics (e.g. Csicsvari et al. (Neuron, 1998)).

Next, the small set of opto-tagged SST and PV interneurons, identified by their direct, short-latency light-evoked responses, served as a "ground truth" reference. We then extracted electrophysiological features, such as auto-correlogram and theta phase histogram, from both tagged and untagged putative interneurons.

Using these electrophysiological features, we applied the K-means clustering algorithm to the entire population of putative interneurons. The cluster exhibiting the highest degree of overlap with the tagged SST or PV interneurons and similar firing phenotypes established for the SST or PV cell group was classified as putative SST or PV interneurons. The similarity of their electrophysiological features to those of the tagged cells (SST: Figures S12D and S12E; PV: Figures S14C and S14D) provided confidence in this classification.

The initial broad classification step separates pyramidal cells from interneurons based on distinct differences in spike waveform and firing rates. The subsequent clustering for SST/PV identification was performed only within the putative interneuron pool. Therefore, the features distinguishing SST and PV cells from each other are different from those distinguishing them from pyramidal cells. We did not attempt to use this interneuron-specific clustering approach to subclassify pyramidal neurons, as we did not perform opto-tagging of specific pyramidal cell types. There is also generally less electrophysiological diversity among CA1 pyramidal cells compared to the wide array of interneuron subtypes.

The method description is included in Methods under "*Interneuron clustering*".

- Silencing SSTs reduces PyrUp firing. What does it do to locomotion patterns? The authors should include a locomotion comparison (not just average speed across the entire trial).

Reply to Reviewer:

To directly assess the impact of SST interneuron inactivation on locomotion, we have now included running speed profiles plotted over time for both control and SST inactivation trials.

As detailed in our earlier response concerning locomotion confounds (Major Point 2, sub-point 4), our analysis revealed the following:

Bilateral CA1 SST inactivation resulted in only a brief and minor, though statistically significant, change in running speed. This change was localized to the period between 2–3 seconds post-run onset, where speed was slightly reduced during inactivation trials (Figure S12K).

In contrast, the unilateral SST inactivation led to significant and more sustained reductions in PyrUp neuron firing rates, from 0 to 3 seconds post-run onset (Figure S12L). If the reduction in PyrUp firing merely reflected altered locomotion, we would expect their temporal profiles of change to align more closely.

These results are included in the section “**Inactivating SST interneurons after run onset impairs PyrUp response, IGSs, and task performance**”: “*However, mean running speed did not significantly change (Figure 5O), and the speed profile exhibited only slight and brief reduction between 2-3 seconds (Figures S12J-S12L), indicating largely normal locomotion.*”

• I found no details in Methods on how CA1 cells were selected. How much was each probe lowered into the brain? How did the authors determine which channels were situated in CA1?

Reply to Reviewer:

We have now included the procedure for targeting and identifying the CA1 region in the revised Methods section. The CA1 pyramidal layer was identified using a combination of stereotaxic depth and electrophysiological signatures:

Craniotomy was centered around the dorsal CA1 region of the hippocampus using stereotaxic coordinates (AP: -2.1 mm from Bregma; ML: \pm 1.7 mm from midline).

Probes were lowered very slowly into the brain under electrophysiological monitoring. The typical target depth from the brain surface to reach the CA1 pyramidal layer is approximately 0.95 to 1.40 mm, though this can vary slightly between animals.

During probe advancement, we continuously monitored online electrophysiological signals, including local field potentials (LFPs) and multi-unit activity (MUA), across the different recording channels of the probe.

The CA1 pyramidal layer (Stratum Pyramidale, SP) was identified based on a convergence of established electrophysiological landmarks, consistent with criteria described in previous literature (e.g., Mizuseki et al. (Nature Neuroscience, 2009)):

* **Prominent theta oscillations:** Strong and coherent theta-band activity (typically 6-10 Hz) in the LFP, particularly evident during locomotion.

* **Sharp-wave ripples (SWRs):** The presence of high-frequency ripple oscillations (100-250 Hz) superimposed on larger sharp waves in the LFP, which are most prominent in CA1 SP and are a hallmark of this layer, especially during quiet wakefulness or immobility.

* **Increased MUA and spike bursts:** A noticeable increase in the density of MUA and the occurrence of complex spike bursts, characteristic of pyramidal neuron firing, as the probe contacts traversed SP.

* **Polarity inversion of sharp-waves/ripples:** A key indicator is the phase or polarity inversion of sharp-waves (and sometimes theta waves) across the radial axis of CA1. We looked for this inversion pattern across the shanks/sites of our high-density probes.

These features were used collectively to determine that the probe was positioned in the CA1 pyramidal cell layer.

This description is now included in Methods under “**Acute extracellular electrophysiology**”.

Minor points:

- The Results are separated into sections whose message bleeds into each other, making it hard to follow the manuscript structure. For example, the first section ends with conclusion: “Thus, IGSs preferentially occur during the task requiring distance integration”. Half way through the 2nd section, results imply that “engaging in distance integration is necessary for generating IGFs”. These analyses could form a single section. The same holds for the Figures and supplemental figures. Often, a supplemental is related to multiple main figures and vice versa. This makes following the analyses difficult.

Reply to Reviewer:

In our revision, we have paid close attention to improving the cohesion between sections. While the fundamental structure of presenting the IGS findings first (Figures 1-2) and then moving to the PyrUp/PyrDown population dynamics (Figure 3 onwards) is maintained, we have worked to:

1. Clarify section goals and avoid overstatement:

In the initial section (Figure 1), we now explicitly state that its primary goal is to “*validate the PI task as a suitable paradigm for investigating the internally generated hippocampal dynamics under conditions that require distance or time integration.*”

We also carefully phrased our conclusions to avoid overemphasizing distance integration alone, by including: “*However, given the consistent running speed profiles across trials, it remained unclear whether animals were integrating over distance or time.*”

2. Strengthen the progression:

Building on the initial finding that “*IGSs preferentially occur during the task requiring distance or time integration*” (from the first section), the subsequent section (corresponding to Figure 2) now provides further evidence to strengthen this hypothesis. Specifically, it demonstrates that the absence of IGFs in the passive task is not merely due to misalignment with run onset, thereby supporting the necessity of distance or time integration for IGS generation.

Following this, in the section “**Subpopulations of pyramidal neurons display distinct responses around run onset**” (corresponding to Figure 3), we clarified the transition from focusing on IGSs to characterizing the broader pyramidal population dynamics. This is reflected in the revised introductory sentence for that section: “*Given that only a small subset of CA1*

pyramidal neurons exhibited IGF in the PI task, we aimed to investigate the dynamics of all pyramidal neurons over time or distance from run onset. ”

3. Enhance figure organization:

We have thoroughly reorganized both the main and supplementary figures to improve readability and ensure that each figure is logically associated with its most relevant section of the text and main figure.

- Repetition: “In the PI task, the run onset potentially marked the starting point of integration in the PI task”

It is now changed to “In the PI task, the run onset potentially marked the starting point of integration.”

- Figures not appearing in order in the text. E.g. Fig 4D referenced earlier than 4A-C and similarly elsewhere.

The figure orders have been readjusted (now Figures 5A-5E).

- “In addition to IGSs, CA1 pyramidal neurons generally exhibited improved trial-by-trial activity correlation when aligned to run onset (Figures S2H).” Fig S2H is probably a typo here as it doesn’t show that.

We have now corrected the typo.

Reviewer #3 (Remarks to the Author):

In the current manuscript Heldman and colleagues train mice to perform a cued path integration task while recording sequential activity patterns in the hippocampal formation. I like the task. They find that the hippocampus generates fields anchored to particular distances/times from run onsets and that these internally generated fields ('IGFs') are not present when the animal runs down the same virtual environment with no visual cues (a passive version). They also observe two sub-populations of cells that either peak (PyrUp) or trough (PyrDown) in their mean firing activity following the start of the cue initiated run. In a series of optogenetic manipulations, they show that disruption of SST interneurons modulates the integrity of PyrUp 'coding,' while PV interneurons regulate the integrity of PyrDown responsive cells. This is an interesting story and many of the experimental results are quite compelling. That said, I have several concerns that should be addressed in a revised manuscript.

The tie between PyrUp and PyrDown cells and path integration computations was tenuous to me and I don't feel like the connection between path integration, the cell response, and the behavior was ever super well explained. The authors should attempt to beef up this portion of the manuscript and potentially include a theoretical/computational model that would explain why this form of tuning would be useful for this computation.

Reply to Reviewer:

We thank the reviewer for this insightful feedback and agree that strengthening the conceptual link between PyrUp/PyrDown cell dynamics, their role in path integration, and behavior is important. In response, we have performed several new analyses and revised the manuscript to better articulate this connection and propose a more explicit functional role for PyrUp/PyrDown populations.

1. Link between PyrUp/PyrDown activity and distance/time integration:

Generalized linear model (GLM) analyses:

We used GLMs based on Kraus et al. (Neuron, 2013) to quantify the contributions of time since run onset, distance traveled, running speed, and licking behavior to PyrUp/PyrDown neuron activity (Figures 4A, 4B, S7A, and S7B). These models revealed that a majority of PyrUp (63.9%) and PyrDown (64.6%) neurons were not significantly modulated by running speed. Similarly, most PyrUp (69.2%) and PyrDown (71.2%) were not significantly influenced by licking. In contrast, a substantial proportion was significantly modulated by either time (PyrUp: 49.2%; PyrDown: 49.7%) or distance (PyrUp: 50.5%; PyrDown: 49.5%), indicating a link to internal representations of elapsed time or distance rather than simply to motor outputs.

Figure 4

A T(12.5%) D(13.8%) TD(36.7%) p(0.10)

B T(13.3%) D(13.1%) TD(36.4%) p(0.53)

Figure S7

A **PyrUp** L(17.5%) S(22.8%) LS(13.3%) p(8.5e-5)

B **PyrDown** L(16.5%) S(23.1%) LS(12.3%) p(3.1e-5)

Spontaneous run bouts:

To evaluate whether PyrUp and PyrDown responses are specifically associated with the onset of time or distance integration (trial-start run onset (TRO)), we presented more data to demonstrate their responses around spontaneous run onset (SRO), occasionally occurring in the middle of a trial (19 animals, 32 sessions, 2902 neurons). Both PyrUp and PyrDown responses were significantly attenuated at SRO, despite similar mean firing rates (Figures S9A and S9B). This attenuation persisted when matching running speed (Figures S9C (example traces), S9D, and S9E, 13 animals, 21 sessions, 1731 neurons) and subsampling the TRO runs to match the number of SRO (Figures S9D). Thus, PyrUp and PyrDown responses are context-dependent and specifically tied to the run onset when the time or distance integration starts.

Figure S9

Immobile timing task:

To provide direct evidence that PyrUp and PyrDown responses are present in different tasks requiring integration rather than obligatorily depend on locomotion, we presented more data in an immobile task where the animal remained stationary and was required to estimate a 4-second time period while viewing grey screens. Even in this immobile state, similar proportions of PyrUp and PyrDown neurons emerged as compared to the PI task (Figures S10D-S10F, and S10K, 5 animals, 15 sessions, 748 neurons, PyrUp-Imm: $39.32 \pm 4.25\%$, PyrDown-Imm: $17.73 \pm 2.79\%$). Thus, PyrUp/PyrDown responses occur during tasks requiring time or distance integration across distinct behavioral states, including locomotion and immobility.

Figure S10

Bayesian decoding:

We applied Bayesian decoding to the single-trial population activity of PyrUp neurons without IGFs and confirmed that this subpopulation effectively encodes time passage (Figures S11A-S11D, in the original manuscript).

2. Connection to behavior:

Good vs. bad trials:

Further supporting a link to accurate integration, PyrUp and PyrDown responses around run onset were more pronounced in good trials compared to bad trials where the animal inaccurately

estimated distance or time (Figures 3G-3J). This implies that response strength correlates with the accuracy of time or distance estimation (in the original manuscript).

Connection to licking behavior:

We probed whether PyrUp/PyrDown dynamics were correlated with the animal's internal estimate of reward timing. We grouped trials based on the timing of the animal's first anticipatory lick (early lick: <2.5 s vs. late lick: >3.1 s, 492 PyrUp and 327 PyrDown neurons, 18 recordings, 15 animals; recordings required at least 15 trials in each category, only considering good trials). In the PI task, anticipatory licking serves as a behavioral proxy for animals' expectation of reward, which is contingent on time or distance integration. Despite divergent speed profiles between these groups, PyrUp activity was similar in the first 2 seconds, and PyrDown neurons showed no significant changes (Figures 4G-4J). However, PyrUp decay time constant increased significantly in late first-lick trials (1.89 ± 0.05 vs. 2.00 ± 0.05 s, $p = 0.036$, Figure 4I). This change in the decay rate indicates that the neural dynamics adjust to the animal's internal estimate of when sufficient time or distance has elapsed, thereby potentially predicting reward-anticipatory behavior.

Figure 4

In addition, the link to behavior is also supported by the behavioral deficits observed upon optogenetic manipulation of SST and PV interneurons, which also impacted PyrUp and PyrDown neurons, respectively (Figures 5 and 7).

3. Interpretation and proposed model:

Based on these findings, we propose that PyrUp and PyrDown neurons implement a two-phase coding scheme relevant to distance or time integration during path integration:

- **Phase I (Integration initiation):** PyrUp and PyrDown neurons exhibit synchronized firing increase or decrease shortly after run onset (or timing initiation in the immobile task). This may act as a neural “start signal” for internal timing or distance integration.
- **Phase II (Time/distance encoding):** Following this initiation phase, PyrUp and PyrDown neurons exhibited heterogeneous decay and ramping trajectories, respectively. We hypothesize that the resulting divergence in their population firing rate collectively encodes the passage of time or the accumulation of distance from the integration initiation. The specific decay or rise rate for individual neurons may represent different “scales” of the integrated variable.

Supporting this model, the decay rate of PyrUp neurons significantly slowed in trials with late anticipatory licks, indicating that the neural dynamics adjust to the animal's internal estimate of when sufficient time or distance has elapsed. This consistency between neural dynamics and behavior suggests that a slower decay may correspond to a slower perceived accumulation rate or a longer integration period.

While a full, detailed computational circuit model is beyond the scope of the current manuscript, we acknowledge the reviewer's suggestion. We are actively developing such models to explore how these two-phase dynamics can be mechanistically implemented at the circuit level (e.g., through specific interneuron interactions and recurrent pyramidal cell connectivity) and how they can be read out for behavior.

We have substantially revised the manuscript to incorporate these new analyses and the model.

Please find these changes in the sections “**PyrUp/PyrDown responses reflect internal integration, not solely locomotion**”, “**PyrUp/PyrDown responses likely predict reward-anticipatory behavior**”, and “**PyrUp and PyrDown neurons employ a two-phase coding mechanism**”.

We have also included in Discussion “**The relationship between PyrUp/PyrDown responses, distance/time integration and behavioral performance**”, and “**PyrUp/PyrDown responses provide a novel coding mechanism for time or distance**”.

Method descriptions can be found in Methods under “**Comparing PyrUp/PyrDown responses across high and low speed trials**”, “**Comparing PyrUp/PyrDown responses across early and late first lick trials**”, and “**Generalized Linear Model (GLM)**”.

It is not entirely clear why these cells respond this way or how they differentiate from other ramping cell responses reported in the past. Since they are so strongly anchored to the run onset, a better analysis of the relationship between movement behaviors (e.g. acceleration) and the responsivity

of these neurons is warranted. This includes a substantially more refined analysis of ‘good’ and ‘bad’ trials which I found to be a bit circular and lacking (there are significantly more criteria that one could use to assess task performance than the three main metrics included in this analysis).

Reply to Reviewer:

We thank the reviewer for raising this important point.

1. Differentiation from other ramping cells:

Ramping activity, characterized by a gradual increase or decrease in firing rate over seconds to minutes, has been reported in various brain regions, including lateral entorhinal cortex (Tsao et al. (Nature, 2018)), human entorhinal cortex (Umbach et al. (PNAS, 2020)), retrosplenial cortex (Alexander & Nitz, (Nature Neuroscience, 2015); Tennant et al. (Current Biology, 2022)), prefrontal cortex (Funahashi et al. (J Neurophysiology, 1989); Cao et al. (PNAS, 2024)), and striatum (Emmons et al. (J. Neuroscience, 2017)). Within the hippocampus itself, long timescale ramping over minutes has been observed (Shikano et al. (Current Biology, 2021)).

In this study, we proposed that PyrUp and PyrDown neurons provide a **two-phase mechanism** for encoding time or distance over seconds, including **Phase I (Integration initiation) and Phase 2 (Time/distance encoding)**.

We hypothesize that this two-phase code first generates a reference signal through the synchronized rise to peak, which then serves as the starting point for subsequent encoding of time or distance.

We sought to understand the mechanisms underlying PyrUp and PyrDown responses (or why these cells respond this way as the reviewer asked) through targeted interneuron manipulation. Our SST inactivation experiments suggest that PyrUp activity is likely driven by excitatory input from upstream regions such as CA3. In contrast, the PV inactivation results indicate that PV interneurons contribute to the suppression of PyrDown neurons around run onset, potentially regulating Phase I of their activity. While these findings provide initial insights, a full understanding of how PyrUp/PyrDown dynamics are generated will require further investigation at both the circuit and cellular levels.

We have included this discussion in the section “**PyrUp and PyrDown neurons employ a two-phase coding mechanism**”, and in Discussion under “**PyrUp/PyrDown responses provide a novel coding mechanism for time or distance**”.

2. Relationship with movement behaviors:

1) Speed analysis:

Comparison of PyrUp/PyrDown firing and running speed profiles:

We plotted running speed over time alongside PyrUp/PyrDown firing profiles. Visualizing running speed with raster plots of example PyrUp neurons from the same trials revealed that PyrUp neurons consistently peak within ~1 second of run onset, whereas running speed typically peaks later, around ~2 seconds (Figures S8C and S8D).

Across all recordings, the average peak timing of PyrUp neurons (Median \pm SEM: 1.24 ± 0.02 s) and PyrDown neurons (1.21 ± 0.03 s) occurred significantly earlier than the peak of running speed (1.69 ± 0.08 s; $p = 1.1e-7$ for PyrUp vs. Speed; $p = 2.9e-8$ for PyrDown vs. Speed, Wilcoxon rank-

sum test, Figures S8E-S8H, left column). This temporal dissociation suggests that PyrUp/PyrDown activity is not simply a direct readout of the speed profile.

Figure S8

Impact of running speed variation (high vs. low speed trials):

We compared neuronal activity across trials with high (>45 cm/s) versus low (35–45 cm/s) average running speeds (2199 PyrUp neurons, 1246 PyrDown neurons, 84 recordings, 41 animals; recordings required at least 15 trials in each category, only considering good trials). Despite significant differences in speed profiles, the characteristic activity of PyrUp neurons (within the first 3 seconds) and PyrDown neurons remained largely stable, arguing against their activity simply reflecting speed (Figures 4C-4F).

Figure 4

2) Acceleration analysis:

As suggested, we analyzed the relationship between acceleration profiles and PyrUp/PyrDown activity. We calculated instantaneous acceleration throughout the run period. Visual comparison of average acceleration traces with average PyrUp/PyrDown firing rates did not reveal a clear correlation across the entire response epoch (Figure S8B). While there is an initial acceleration phase during run onset, the peak of PyrUp/PyrDown activity and their subsequent ramping/decay phases do not appear to be simple reflections of the acceleration/deceleration profile of the animal.

Figure S8

GLM including acceleration:

To quantify the contribution of acceleration on PyrUp/PyrDown neuron activity, we extended our Generalized Linear Model (GLM) analysis described earlier to include acceleration as an additional predictor, together with time, distance, speed, and licking. The results showed that only a very small fraction of PyrUp neurons (4.2%) and PyrDown neurons (3.8%) exhibited firing rates that were significantly modulated by acceleration. This contribution is substantially smaller than that of time or distance, further supporting the idea that acceleration is not a primary driver of these firing dynamics.

These results are now included in the section “**PyrUp/PyrDown responses reflect internal integration, not solely locomotion**” (the second paragraph).

3. Refined analysis of ‘good’ and ‘bad’ Trials:

We acknowledge the reviewer's concern that our initial "good" vs. "bad" trial classification (based on run duration, stop location, and reward outcome) might be somewhat circular or should be expanded.

To address this, we have first refined the description of our 'good' and 'bad' trial analysis within the manuscript to enhance clarity and logic.

Furthermore, we have also incorporated additional behavioral comparisons that provide an alternative view of how neuronal activity relates to behavioral parameters, including high vs. low running speed trials, and early vs. late first-lick trials.

These additional analyses move beyond a simple "good/bad" dichotomy and explore how PyrUp/PyrDown activity covaries with specific aspects of behavior that are relevant to task performance and distance/time integration.

Collectively, these findings further support the idea that PyrUp and PyrDown dynamics are not simply reflections of locomotion, but represent structured neuronal dynamics relevant to time or distance integration.

The refined description can be found in the section “**Accurate integration is correlated with the expression of IGSSs**” (paragraphs 1 and 2).

The difference between the IGF response on passive versus active versions of the task is interesting, but many prior papers have shown robust place fields in animals navigating in darkness that are not actively required to perform path integration (i.e. are just rewarded on either side of the track or are free foraging in an open environment). The difference here is perplexing and makes one wonder whether the primary results would hold in non-VR tasks. While I do not think it is appropriate to ask the authors to show that their results would hold in freely moving animals, I have to admit I find myself wondering why they used a task with head-fixation. At the very least, the authors should discuss the possibility of impoverished vestibular information on their main results.

Reply to Reviewer:

The reviewer raises an important point regarding the discrepancy between our findings in the passive condition and prior reports of robust place fields in animals navigating without explicit path integration demands, including in complete darkness.

To address this point, we summarize **factors that can influence the expression of firing fields**.

1. Impact of sensory and motor cues:

As animals navigated environments with minimal sensory cues, hippocampal neurons often showed reduced spatial tuning, and a relatively small proportion of neurons exhibited firing fields (Bourboulou et al. (eLife, 2019); Sharif et al. (Neuron, 2021); Aghajan et al. (Nature Neuroscience, 2015); Chen et al. (eLife, 2018)). However, robust place fields can still emerge in freely moving animals navigating in complete darkness without explicit path integration demands, such as during random foraging or running on linear tracks with rewards at ends. In these scenarios, animals may rely on non-visual cues, including vestibular and proprioceptive inputs, olfactory traces, and tactile information from the environment. These multimodal cues likely supported stable spatial representations, even in darkness.

2. Impact of head-fixation:

As pointed out by the reviewer, head-fixation presents a distinct challenge by impoverishing vestibular input, which is known to be critical for spatial tuning of place cells and grid cells (e.g., Stackman et al., 2002). Multiple studies have attributed degraded spatial selectivity in head-fixed virtual reality (VR) environments to the lack of vestibular input (Aghajan et al. (Nature Neuroscience, 2015); Chen et al. (eLife, 2018); Ravassard et al. (Nature Neuroscience, 2013)). Nevertheless, a high percentage of neurons exhibited robust firing fields under head-fixed settings when sensory cues were provided (Bourboulou et al. (eLife, 2019); Sharif et al. (Neuron, 2021)). Thus, impoverished vestibular input alone might not fully explain the low prevalence of firing fields reported in some head-fixed behavioral tasks.

3. Impact of task demands:

Beyond sensory constraints, cognitive and memory demands significantly influence IGS generation. Even in cue-poor or cue-absent environments, the prevalence of IGSs typically increases under conditions requiring higher demands, such as in contexts relevant to temporal integration or working memory (e.g. Pastalkova et al. (Science, 2008); Kraus et al. (Neuron, 2013)).

Summary

Consistent with these prior findings, our PI and immobile tasks represent tasks with minimal sensory cues and impoverished vestibular input. These tasks, while less demanding than hippocampus-dependent working memory tasks, still require animals to track time or distance.

Correspondingly, in these tasks, only a small percentage of neurons exhibited IGFs (Figures 2G and S10C). In contrast, in the passive task with minimal sensory cues, impoverished vestibular inputs, and no requirement for integration, IGFs are largely absent (Figures 1H and 2G).

This discussion is now included in Discussion under “**Factors that influence the expression of IGS**”.

Lack of neurons with IGF in freely moving animals:

The absence of IGFs in our passive task aligns with previous findings. Pastalkova et al. (Science, 2008) and Hirase et al. (European J. Neuroscience, 1999) reported no clear firing fields during spontaneous wheel running in freely moving rats when no task demands were present. In summary, studies in freely moving animals also suggest that behavioral engagement and task demand play a role in recruiting internally generated sequences (e.g. Pastalkova et al. (Science, 2008); Kraus et al. (Neuron, 2013)), supporting our observations when comparing the head-fixed PI and passive tasks.

This discussion is now included in Discussion under “**Factors that influence the expression of IGS**” (the last paragraph).

Potential advantages of VR setting:

While we acknowledge that head-fixation imposes constraints on the vestibular inputs, it can offer some advantages for experimental control.

The VR setup allowed us to minimize confounding sensory cues and maintain consistent visual input across trials. The small treadmill size also limited tactile and olfactory cues. These design features enabled us to better isolate time and distance integration under well-controlled sensory and behavioral conditions. Disentangling these variables in freely moving animals—where rich and dynamic multimodal cues are present—remains an open challenge.

This discussion is now included in Discussion under “**Potential advantages of head-fixed behavioral settings**”.

Relatedly I would like to see more classic metrics of place field integrity explored here. For example, is spatial information disrupted in the passive task or just trial to trial reliability?

Reply to Reviewer:

To address this point, we calculated the temporal information for all CA1 pyramidal neurons in both the passive and path integration (PI) tasks after run onset alignment.

The analysis revealed that, at the population level, the average temporal information per spike was only slightly, though significantly, lower in the passive task compared to the PI task (e.g., PI task: 0.158 ± 0.003 bits/spike; Passive task: 0.153 ± 0.007 bits/spike; $p = 0.01$, two-sample Kolmogorov-Smirnov test). The statistical significance is potentially related to the large number of neurons in the PI task.

This relatively small difference in overall temporal information contrasts with the much more dramatic reduction in the percentage of neurons exhibiting firing fields (IGFs) and the significantly lower trial-to-trial activity correlation in the passive task (as shown in Figures 2G and S2E).

These results support that the primary disruption in the passive task lies in the consistency of firing across trials.

An explanation of the statistical test for classifying PyrUp and PyrDown cells should be included in the main manuscript since these classifications are so critical to the story.

Reply to Reviewer:

We have included the description in the section “**Subpopulations of pyramidal neurons display distinct responses around run onset**” (paragraphs 1 to 3).

The method description is updated in Methods under “***Identification of PyrUp and PyrDown neurons***”.

We sincerely thank the editor and reviewers for their time and expertise. We appreciate the detailed and constructive comments and the opportunity to significantly enhance our manuscript. In response, we have undertaken substantial revisions to improve clarity, strengthen our conclusions, and incorporate new analyses. We have now expanded the number of supplementary figures from 15 to 18.

Our main revisions address the third reviewer's central concerns as follows:

1. On Novelty and the Classification of PyrUp/PyrDown Neurons: To address the concern about novelty, we have clarified our contributions beyond prior work. Furthermore, we now provide multiple lines of evidence that PyrUp and PyrDown neurons are functionally distinct classes, including their differential modulation by specific interneuron subtypes and the distinct changes of these two populations in the passive task.

2. On Causal Evidence for the Two-Phase Model: We acknowledge the reviewer's primary concern regarding the lack of direct manipulation to test our model. To address this, we now summarize key results from two manipulation experiments: **phase-specific optogenetic silencing** and a task in which we **varied the required travel distance**. These results provide strong evidence for the model and help disentangle the encoding of an internal, time/distance integration from simple locomotor response.

3. On Dissociation from Motor Variables: We have conducted a new analysis to test the speed modulation of PyrUp/PyrDown neurons. A time-resolved correlation analysis clarifies that the PyrUp/PyrDown neurons were only weakly speed-modulated, with a significant increase in modulation during the first second after run onset. This result further supports that PyrUp/PyrDown dynamics do not merely reflect locomotion.

4. On Clarifying Control Analyses and Task Logic: We have clarified the logic behind our control conditions. We now explicitly detail why the task design of the **passive task** intentionally eliminated a reliable "start" signal. We also performed a new analysis with relaxed criteria for our **"good vs. bad trial" comparison**, demonstrating that our findings are not an artifact of circular definitions.

In summary, the revised manuscript is substantially improved thanks to the reviewer's feedback. By incorporating new analyses, we now provide a stronger foundation for the novelty of the opposing ramping populations, their functional distinction, and the role of the two-phase dynamics in time/distance integration. We have detailed these changes in the point-by-point responses that follow.

We hope this revised and improved manuscript will now be suitable for publication in Nature Communications.

REVIEWER COMMENTS

Reviewer #1 (Remarks to the Author):

The authors have addressed all of my and the other reviewer's concerns. I have no other concerns and recommend publication.

Reviewer #2 (Remarks to the Author):

The authors went to great lengths to carefully address all points I raised, as best as possible and in great detail, within the confines of their existing experiments and datasets. I appreciate all the additional extensive analyses and clarifications. I feel like the claims have been strengthened and the manuscript structure is much improved. I recommend the manuscript for publication.

Reviewer #3 (Remarks to the Author):

The authors have mainly addressed my concerns but some of the newly included analyses have raised additional issues/comments outlined below.

I agree with many of the concerns raised by reviewer 2, in particular the concerns about novelty. Ramping responses have been reported in CA1 (Ning, Bladon, and Hasselmo). I do not feel particularly convinced by the inclusion of a theoretical two-phase model regarding the role of UP and DOWN neurons in path integration nor their classification as distinct classes of neurons. There is no manipulation to test their theory nor a computational model showing that it is useful for self-localization via path integration.

Reply to Reviewer:

We thank the reviewer for the thoughtful feedback and for raising important points regarding the novelty of our findings and the functional relevance of our proposed two-phase model. In response, we have revised the manuscript to better contextualize our work, clarify our analyses, and strengthen our conclusions.

1. Novelty of the Finding Beyond Previously Reported Ramping Activity

We appreciate the reviewer citing the work of Ning et al. (2022), which we now discuss more extensively. Building upon their key initial observations of ramping activity, our work delivers several significant and novel advances:

- a) **Discovery of an Opposing Neuronal Population (PyrDown):** Ning et al. provided examples exhibiting a similar firing pattern to the PyrUp neurons; however, the opposing population — the PyrDown neurons was not reported. These neurons exhibit a rapid decrease in activity at the onset of path integration, followed by a slow ramp-up toward reward.
- b) **From Descriptive Observation to a Quantitative Framework:** While Ning et al. provided descriptive examples of PyrUp-like ramping activity, in this study, we identified PyrUp/PyrDown populations as major functional subpopulations of CA1 pyramidal neurons during a path integration task. Our study offers a quantitative characterization of both populations. We defined their dynamic properties (e.g., peak/trough time, time constants), proposed their relevance to distance/time integration, and demonstrated their specific roles in integration through their differential regulation by different interneuron subtypes during optogenetic manipulations. Our investigation provides a framework for further study into their functions and underlying mechanisms.

- c) **Generalizability:** The observation of PyrUp-like activity across species (mice vs. rats) and behavioral paradigms (virtual navigation vs. freely moving) suggests that this may be a general dynamic pattern in CA1.

2. Evidence for PyrUp and PyrDown as Functionally Distinct Populations

We acknowledge the reviewer's skepticism regarding the classification of PyrUp/PyrDown neurons as distinct classes. We have strengthened the manuscript with multiple lines of evidence supporting that they are not only statistically separable but also functionally distinct.

- **Statistically Robust Classification:** We would like to clarify that our classification is based on the distinct neuronal responses immediately following the onset of integration, not on the subsequent slower ramping/decaying activity. This classification is robust, yielding identical groupings from two independent statistical methods (thresholding and a shuffle-based test, Figures 3E-3J and S7B-S7D), confirming the presence of two non-overlapping response types.
- **PyrUp/PyrDown neurons are preferentially regulated by distinct inhibitory interneurons:** Optogenetic inhibition of SST and PV interneurons preferentially modulated PyrUp and PyrDown neurons, respectively, and resulted in distinct behavioral impairments (Figures 5 and 7). These results suggest that PyrUp/PyrDown neurons not only exhibit opposing dynamic patterns but likely play different roles in distance/time integration.
- **PyrUp/PyrDown neurons exhibit independent changes in the passive task:** We have added new analyses from the passive task where path integration is not required. In this condition, the proportion of PyrDown neurons was significantly reduced compared to the PI task, while the proportion of PyrUp neurons remained largely unchanged (Figures S14C and S14D). This dissociation supports that the two populations can be independently modulated by task demands. Please find the details in the reply to "What are the GLM results on the passive task?"

3. Evidence for the Role of the Two-Phase Dynamics in Path Integration

The reviewer's primary concern was the lack of experimental support for our theoretical two-phase model. We have now strengthened the manuscript with the following evidence that supports our model's predictions.

- **Neural Dynamics Correlate with the Animal's Internal Estimate of Distance or Time:** To test whether PyrUp/PyrDown responses relate to the animal's internal estimate of elapsed distance/time, we examined their relationship to the timing of the animal's first anticipatory lick, which is contingent on time or distance integration. For that, we sorted trials into "early lick" and "late lick" categories (Figures 4G-4J). Our results showed that while the initial response of PyrUp/PyrDown neuron activity (corresponding to our proposed Phase I in the model) remained similar across these categories, the decay time constant of PyrUp neurons in Phase II was significantly longer in late lick trials ($p = 0.036$, Wilcoxon rank-sum test, Figure 4I).

These neural dynamics were stable in the first 2 seconds when running speed varied across categories (Figures 4G-4J), suggesting they do not merely reflect locomotion.

- **Altered PyrUp Dynamics in the Passive Task:** Our analysis of the passive task revealed that while the proportion of PyrUp neurons remained similar to the PI task, these neurons exhibited significantly longer time constants, and the correlation between these neurons' decay time constants and firing rate changes at run onset was lost (Figures S14E-S14G). These changes in PyrUp dynamics in a task that does not require integration are consistent with our two-phase dynamics hypothesis, in which the heterogeneous decay rates of PyrUp neurons in Phase II are relevant to distance or time integration. Please find the details in the reply to “What are the GLM results on the passive task?”

4. Evidence from Manipulation Experiments for the Role of the Two-Phase Dynamics in Path Integration

We agree with the reviewer that manipulations are necessary for testing the functional roles of our proposed two-phase model. While we opt to present these complex experiments in a follow-up manuscript to maintain a focused narrative, we agree it is important to summarize the key results here to address the reviewer's concerns.

- **Phase-Specific Silencing:** We conducted optogenetic experiments where we silenced CA1 pyramidal cell activity by activating PV interneurons during either the first or the second phase of PyrUp/PyrDown responses. Suppressing CA1 activity during Phase II led to significant behavioral impairments in the task, including licking pattern and reward rate. In contrast, shutting down activity in Phase I had no clear impact on task performance (Heldman et.al., bioRxiv, 2025).
- **Varying Travel Distance:** In a separate experiment, we varied the required travel distance in the PI task across blocks of trials by randomly selecting from 100, 180, and 260 cm. We found that the PyrUp neurons' decay time constants in Phase II scaled with the required distance, with activity decaying more slowly for longer-distance trials (Mean±SEM: 100 cm: 1.62±0.05s, 180 cm: 2.29±0.05s, 260 cm: 3.22±0.08s, $p < 0.001$ for all pairs, Kruskal-Wallis test with post-hoc multiple comparison tests). In contrast, the initial response in Phase I, particularly the first second after run onset, remained unchanged (mean FR 0 to 1 s: 100 cm: 2.97±0.19Hz, 180 cm: 3.03±0.14Hz, 260 cm: 3.36±0.16s, $p = 0.47$, Kruskal-Wallis test).

The rise time constant of PyrDown neurons in Phase II also showed significant modulation with travel distance, particularly between 100 cm and other distances (100 cm: 2.17±0.20s, 180 cm: 3.36±0.25s, 260 cm: 3.45±0.26s, 100 vs 180 cm: $p < 0.001$, 100 vs 260 cm: $p < 0.001$, 180 vs 260 cm: $p = 1.0$. Kruskal-Wallis test with post-hoc multiple comparison tests), while the initial response in Phase I remained similar (mean FR 0 to 1 s: 100 cm: 2.03±0.14Hz, 180 cm: 1.93±0.14Hz, 260 cm: 1.89±0.14Hz, $p = 0.49$, Kruskal-Wallis test).

In contrast, the locomotion speed remained similar across different travel distances in the first 2 seconds (0-1s: $p = 0.90$, 1-2s: $p = 0.60$, Kruskal-Wallis test), and exhibited prolonged elevated speed in the blocks with longer travel distances. This result supports that PyrUp/PyrDown responses do not merely reflect locomotion.

Together, these results, which dissociate the Phase II dynamics from the initial response, further support a role for PyrUp/PyrDown neurons' Phase II dynamics in encoding an internal estimate of distance/time.

In conclusion, we have strengthened the manuscript to better highlight our novel contributions. Furthermore, the results from the manipulation experiments summarized above provide further validation for our proposed two-phase model. We have summarized this reply in the **Discussion** section under “**PyrUp/PyrDown responses provide a novel coding mechanism for time or distance**”.

I do not believe that the new GLM truly addresses concerns related to the separation of time, distance, and movement coding. These variables all tightly co-vary in the task setup and accordingly it is unclear if the GLM is appropriate from an analytical perspective. I feel only a manipulation of the relationship between reward delivery and distance traveled would truly disentangle things.

Reply to Reviewer:

We thank the reviewer for this critical point. We agree that in the PI task, time, distance, and movement variables are highly correlated, limiting the conclusions that can be drawn from a GLM alone. A better way to disentangle these factors is through a direct manipulation.

As detailed in our response to the reviewer’s previous point, we performed the suggested experiment: We varied the required travel distance for reward in the PI task across blocks of trials. This experiment allowed us to test how PyrUp/PyrDown dynamics relate to integrated distance/time, as opposed to general motor variables.

Our results show that the decay time constants of PyrUp neurons during Phase II scaled with the required travel distance, with activity decaying more slowly in longer-distance trials. The rise time constant of PyrDown neurons also showed significant modulation.

In contrast, the initial response in Phase I was largely unchanged (within the first second) across the travel-distance conditions. In addition, the animals’ running speed during the initial phase of the run (within the first 2 seconds) was consistent across all conditions. Speed was maintained for a longer duration in longer-distance trials.

This dissociation—where the later-phase neural dynamics scale with the total travel distance while the initial neural response does not—provides evidence suggesting that the Phase II ramping activity is more closely related to an internal estimation of integrated distance/time to the goal.

Relatedly, what are the GLM results on the passive task?

Reply to Reviewer:

We thank the reviewer for this question. We have performed the requested GLM analysis and a series of related analyses on the passive task data. These new results, now included in the manuscript, provide a critical comparison to the PI task and lend support to the two-phase model.

Our findings revealed several key differences between the passive and PI tasks:

1. A Specific Reduction in the Proportion of PyrDown Neurons

The proportion of PyrUp and PyrDown neurons showed a differential change. While the percentage of PyrUp neurons was similar between tasks (160/423 neurons, 37.8% in passive vs. 37.2% in PI), the percentage of PyrDown neurons was significantly reduced in the passive task (47/423, 11.1% in passive vs. 19.4% in PI, Figures S14A-S14D). This change aligns with the behavioral difference that animals showed significantly less predictive licking in the passive task (Figure 1E). The reduced prevalence of the PyrDown population, which ramps up toward the reward location, is consistent with a potential role for these neurons in reward prediction. This result also supports that PyrUp and PyrDown neurons might form different functional classes.

2. Altered Temporal Dynamics in PyrUp Population

Although the proportion of PyrUp neurons was similar, their temporal dynamics were altered.

- a) **Loss of Heterogeneous Coding:** In the PI task, the decay rates of PyrUp neurons were heterogeneous and correlated with their firing response changes around run onset (the after-to-before run-onset firing rate ratio (R)). This relationship became insignificant in the passive task (Figures S14E and S14F).
- b) **Slower Ramping:** Correspondingly, the PyrUp neurons in the passive task exhibited a significantly longer decay time constant on average and a delayed peak time compared to the PI task (Figures S14G and S14H).

The difference in decay time constant remained similar when we analyzed the PI recordings with a matching running speed profile to the passive task (Figures S2H and S2I, for speed and licking profiles after speed matching, R vs. decay time constant: PI: correlation $r = 0.45$, $p = 3.4e-24$ (t-statistics); Tau: PI: $2.94 \pm 0.06s$, Passive: $3.32 \pm 0.11s$, $p = 1.9e-03$; Peak time: PI: $1.49 \pm 0.05s$, Passive: $1.67 \pm 0.10s$, $p = 0.14$. Wilcoxon rank-sum test).

In contrast, the rise time constants of the small number of PyrDown neurons remained largely intact and were significantly correlated with their firing response changes around run onset, as in the PI task (R vs. decay time constant: correlation $r = -0.62$, $p = 3.0e-06$; Tau: PI: $2.30 \pm 0.07s$, Passive: $3.40 \pm 0.59s$, $p = 0.26$).

The loss of heterogeneous ramping and the delay in peak time in the absence of an integration demand provide evidence that PyrUp dynamics are a feature of the distance/time integration process. The altered dynamics specific to PyrUp neurons further support that PyrUp and PyrDown neurons might form different functional classes.

3. GLM Results

Finally, we performed the GLM analysis as suggested. We observed similar results on the time, distance, and speed modulation of individual PyrUp/PyrDown neurons compared to the PI task. The majority of PyrUp and PyrDown neurons were significantly influenced by time and distance (PyrUp: time: 58.4%, distance: 73.2%, $p = 0.026$. PyrDown: time: 58.5%, distance: 60.9%, $p = 0.45$. p-value is calculated using Wilcoxon rank-sum test between the deviance by time and by distance). In contrast, most neurons were not significantly modulated by speed or licking (Not significantly modulated: speed: PyrUp: 60.6%; PyrDown: 60.0%; licking: PyrUp: 69.4%; PyrDown: 66.7%). This result is as expected, given that on the individual neuron level, PyrUp/PyrDown neurons exhibited a similar shape of dynamics in the passive and PI tasks.

However, on the population level, there were significant changes in the prevalence of PyrDown neurons and PyrUp neuron dynamics. We thus interpret the GLM result as a confirmation that

these neurons remain modulated by basic task variables. Our finding lies in how they are modulated: the heterogeneous decay rates across the PyrUp population and a significantly larger percentage of the PyrDown population are present only during the PI task.

In summary, the passive task serves as a control. The specific reduction of the PyrDown population and the loss of the heterogeneous decay rates in the PyrUp population (even after matching speed) provide evidence that these neuronal dynamics are not mere correlates of locomotion but are features specifically associated with the distance/time integration process.

We have included this reply in the **Results** section “**PyrUp and PyrDown neurons employ a two-phase coding mechanism**”.

Figure S14

The authors suggest that there is a ‘temporal dissociation’ between peak firing and speed that indicates a decoupling of the two. I am not convinced by this argument - several studies have shown neurons can have temporally offset relationships to behavioral/sensory variables. At the very least this should be addressed in the discussion.

Reply to Reviewer:

We thank the reviewer for this thoughtful comment. We agree that a temporal offset is not sufficient evidence to claim a decoupling between neural activity and a behavioral variable.

We would like to clarify that the purpose of the high/low speed analysis is to provide evidence that PyrUp/PyrDown responses are not merely reflecting running speed. Our argument hinges on the direction of this offset. A neuron that encodes speed would be expected to lag behind the behavioral variable due to sensory and processing delays. In contrast, we observe that the peak activity of PyrUp/PyrDown neurons precedes the peak in the animal's speed (Figures S9C-S9H).

This anticipatory firing suggests that PyrUp and PyrDown responses are not merely a direct readout of locomotion patterns.

We rephrased the sentence to clarify the point in the **Results** section “**PyrUp/PyrDown responses reflect internal integration, not solely locomotion**”.

I still feel that the good vs bad trial analysis is circular (though I appreciate that the authors have addressed whether the sampling could explain differences).

Reply to Reviewer:

We thank the reviewer for raising this question. We understand the concern that our trial selection criteria may be circular. To address this, we have both clarified our original analysis and performed a new analysis with relaxed inclusion criteria.

1. Clarification of Alignment in "Bad Trials"

We have clarified in the Methods that even in bad trials where the animal did not come to a complete stop, it typically slowed down significantly. We therefore defined the run onset in these trials as the moment of lowest speed between when the animal reached the reward zone in the previous trial and the start of the current trial. This analysis allowed us to capture a behaviorally relevant "start" point for a potential integration period, making the comparison to "good trials" more direct.

2. A New Analysis with Relaxed Criteria to Test for Circularity

To directly test if the "complete stop" criterion was responsible for our findings, we re-ran the analyses with this criterion removed. In this new analysis, "good trials" were defined only by two criteria: largely uninterrupted running and successful reward collection.

Even with the relaxed criteria, we still observed similar behavioral differences (Figures S6A-S6B). Animals still exhibited more licking in the early part of running in the bad trials, despite maintaining similar mean running speed. Meanwhile, IGS remained impaired in the bad trials (Figure S6C).

This new analysis demonstrates that the impairment of IGSs may not be an artifact of our trial selection criteria.

We have incorporated this discussion in the **Results** section “**Accurate integration is correlated with the expression of IGSs**” and **Methods** under “**Criteria for good and bad trials**”.

Figure S6

“Despite clear differences in running speed, the firing rates of both PyrUp neurons and PyrDown neurons 5 remained largely stable, with PyrUp activity stability evident within the first three seconds (Figures 4C-4F).”

Speed-modulated cells in the extended hippocampal formation often saturate at higher speeds (Gois et al. 2018; Hinman et al., 2016) with greatest speed modulation happening at lower speeds. The results here are consistent with this and I don't find this to be an especially compelling argument against pyramidal up or down responses being correlated/anticorrelated responses to the initial acceleration on the track.

Reply to Reviewer:

We thank the reviewer for this valuable feedback. We agree that the saturation of speed modulation at high speeds cannot rule out that PyrUp/PyrDown responses reflect the initial acceleration on the track.

The following two analyses tested the hypothesis above.

1. Dissociating the PyrUp/PyrDown Responses from the Speed Increase at Run Onset

To test if the response is tied to the speed increase at run onset, we identified instances where the animal initiated a spontaneous run bout in the middle of a trial and matched these spontaneous run bouts for speed with the runs from the start of a trial. This creates a control condition where the motor action (accelerating from stop) is present, but the context of initiating a distance/time integration is absent. Behaviorally, animals did not exhibit a significant increase in reward zone overshooting after a spontaneous run onset (percent of rewarded trials: trials with spontaneous run (SR): $92.98 \pm 1.93\%$, trial without spontaneous run (TR): $96.17 \pm 1.63\%$, $p = 0.07$, Wilcoxon rank-sum test), and the lick patterns in trials with spontaneous run bouts remained unchanged (mean lick rates: 30-100 cm: SR: 0.02 ± 0.005 licks/cm, TR: 0.02 ± 0.005 licks/cm, $p = 0.77$; 100-140 cm: SR: 0.09 ± 0.01 licks/cm, TR: 0.11 ± 0.02 licks/cm, $p = 0.51$; 140-180 cm: SR: 0.25 ± 0.02 licks/cm, TR: 0.28 ± 0.03 licks/cm, $p = 0.55$, Wilcoxon rank-sum test). These results indicate animals were continuing, not restarting, their internal estimate of distance (Figure S10C).

Meanwhile, PyrUp/PyrDown responses were significantly attenuated during the spontaneous, mid-trial run onset compared to trial-start onset (Figure S10). This finding dissociates neuronal responses from merely motor responses and demonstrates that they are specifically linked to the initiation of an integration.

2. Quantifying the Speed Modulation of PyrUp/PyrDown Responses During Phase I

Guided by the literature the reviewer cited (Gois et al., 2018; Hinman et al., 2016), we next quantified the time-resolved relationship between running speed and instantaneous firing rate in one-second bins. This analysis revealed:

- a) **A Transient Increase of Correlation in Phase I:** Running speed and the instantaneous firing rate of PyrUp/PyrDown neurons were only weakly correlated, with a significant increase in correlation during the first second after run onset (Phase I) when the animals were accelerating at lower speeds (Figures S13A-S13D).
- b) **Distinct from Potential Speed Cells:** The initial correlation in the first second after run onset was significantly weaker than that observed in interneurons recorded simultaneously, a population known to exhibit strong speed modulation (Figure S13E).

A Refined Interpretation of Two-Phase Dynamics

Taken together, these new analyses lead to a more refined interpretation that strengthens our two-phase model:

- **Phase I** appears to mark the onset of integration. This signal is time-locked to the trial-start run onset (as shown by the spontaneous run analysis). And it is modulated by the initial speed increase, as revealed by the correlation analysis.
- **Phase II** appears to represent the integration component. During this phase, the PyrUp/PyrDown neural activity is significantly less modulated by speed and instead relates to the integrated distance/time, as supported by its correlation with the animal's first

anticipatory lick timing, its alteration in the passive task, and its scaling with total travel distance.

We have incorporated these new analyses and this revised interpretation into the **Results** section under “**PyrUp/PyrDown responses reflect internal integration, not solely locomotion**” and “**PyrUp and PyrDown neurons employ a two-phase coding mechanism**”.

Figure S13

I don't understand the logic of the SRO analysis. Why doesn't the animal need to path integrate from the random stopping point to perform the task?

Reply to Reviewer:

We appreciate that the reviewer raises this point, which allows us to clarify the logic of this control analysis.

We agree with the reviewer that the animal needs path integration from the random stop. However, behavioral data support that animals likely did not restart path integration after a random stop, as mentioned in the reply to the last question. Our SRO analysis is built upon this exact premise.

The analysis is designed to distinguish between two distinct run-onset events that share a similar speed profile:

1. **Trial-Start Run Onset:** At the beginning of a trial, the animal initializes a new distance/time integration.
2. **Spontaneous Run Onset (SRO):** After stopping mid-trial, the animal resumes its ongoing distance/time integration from its current accumulated value.

Our SRO analysis was designed as a control to test whether the PyrUp/PyrDown responses were specifically linked to the initialization event (1) or were simply a general response from the start of running (which occurs in both 1 and 2).

The fact that these responses were present at the trial start but significantly attenuated at SROs, even after speed matching (Figure S10), suggests that they are not generic responses. Instead, they preferentially mark the initialization of distance/time integration.

We have clarified this point in the **Results** section under “**PyrUp/PyrDown responses reflect internal integration, not solely locomotion**”.

I remain confused by the immobile task results and I do not have a grasp on why IGFs would emerge in the immobile task but not in the passive task. This is especially confusing if IGFs in the immobile task were better aligned to the last lick of the previous trial which should be the case for the passive condition. What is the authors explanation?

Reply to Reviewer:

We thank the reviewer for raising this question, as it highlights a crucial aspect of our experimental design. The presence of IGFs in the immobile task and their absence in the passive task is not a contradiction. Instead, this difference is a result of the task designs and provides evidence for the conditions under which IGFs are generated.

The difference between the two tasks is the reliability of the integration start signals available to the animal.

Immobile Task: A Reliable Start Signal Exists. In this task, the trial structure is predictable. The "last lick" of a trial serves as a sufficiently reliable temporal anchor for the animal to initiate time integration to predict the next reward. The animals' behavior confirms their strategy: they developed robust predictive licking. We found that IGFs were present in this task.

Passive Task: The Reliable Start Signal Does Not Exist. We designed this task specifically to disrupt a robust distance/time integration strategy. First, there was no start cue in the passive task. The screens remained gray throughout each recording session. Second, by requiring the animal to meet a specific speed criterion (>30 cm/s) and enforcing a minimum delay since the last reward before starting the distance counter in a trial, we introduce significant temporal jitter. This makes the "last lick" or "run onset" an unreliable anchor for the animal to initiate distance or time integration. The animal's behavior again confirms the effect of our design: predictive licking was significantly diminished (Figure 1E). Correspondingly, we observed an absence of IGFs in this task (Figures 1F-1H and 2G).

In summary, the presence of IGFs reflects that the animals have a reliable strategy to initiate and integrate distance/time.

We have revised the **Discussion** section “**Factors that influence the expression of IGS**” to make this distinction clearer.

“These changes in the decay/rise rate indicated that the PyrUp/PyrDown dynamics adjust to the animal's internal estimate of when sufficient time or distance has elapsed.” Again, I don't understand why there wouldn't be anticipatory responses similar to Up/Down in the passive task - the animal can still track time to reward after the gray screen appears right?

Reply to Reviewer:

We thank the reviewer for raising this question, and we would like to clarify our experimental logic. This explanation might hinge on that there is no reliable "start" signal for distance/time integration in the passive task. Our task was deliberately designed to eliminate such a start signal.

1. **No Start Signal:** Unlike the PI task, there was no visual cue, and the screens stayed gray throughout the entire recording session in the passive task. The distance count only began once the animal initiated a run and met specific criteria based on running speed and the last lick time. There was no reliable signal available to the animal to start integration from.
2. **Behavior:** Because there was no reliable start signal to initiate integration, the animal may not form a reliable prediction for the reward. Correspondingly, the predictive licking was significantly diminished (Figure 1E).
3. **Neuronal Activity:** The changes in neuronal activity reflected this behavioral change. IGSs were absent (Figures 1F, 1H, and 2G), PyrDown population was diminished, and PyrUp dynamics were altered (Figure S14).

In summary, the animal cannot effectively track distance/time in the passive task, likely because there is no reliable "start" signal.

We have modified the **Discussion** section “**Factors that influence the expression of IGS**” to make this distinction clearer and the **Methods** section “**Behavioral training**” to clarify these points.

Minor comments.

“Good trials met the following criteria: animals came to a complete stop before trial run onset, allowing this onset to mark the start of integration; animals maintained largely uninterrupted running; and successfully obtained the reward (77.90±2.88% of all trials, see Methods). In contrast, bad trials failed to meet at least one criterion. Despite maintaining a mean running speed similar to good trials (Figure 3A, good trials: 48.12±0.84cm/s, bad trials: 45.20±0.84cm/s), bad trials resulted in a comparatively lower reward rate of 86.34 ± 4.36%” - The percentage reward rate seem off given the context of this sentence.

Reply to Reviewer:

Sorry for the confusion. The confusion arises from our definition of a "bad trial." For our analysis, a trial was categorized as "bad" if it failed any of the following criteria: the animal did not stop before the run, running was interrupted, or the reward was missed. This means a trial can be labeled "bad" even if the animal ultimately received a reward.

To clarify the statistics, the reward rate for "good trials" was 100%, as successfully obtaining the reward was one of the criteria. $86.34 \pm 4.36\%$ was the reward rate for "bad trials". Therefore, the statement "a comparatively lower reward rate" refers to the comparison between the 100% success rate on good trials and the 86.34% success rate on bad trials.

In addition, based on the trial selection criteria, $77.90 \pm 2.88\%$ trials were "good trials" in the PI task.

We have revised the sentences in the **Results** section under "**Accurate integration is correlated with the expression of IGSs**".

What is the relationship between late/early lick trials and high/low speed? My guess is that early licks systematically occur with deceleration - both metrics of the animals confidence?

Reply to Reviewer:

We thank the reviewer for raising this interesting question. Based on our data, the relationship between lick and speed can be complex.

An early initiation of licking can arise for different reasons. We listed two of them here: a) the animal ran faster and reached its estimated reward distance earlier, and b) the animal was less accurate in integrating distance/time.

In the early/late lick trial analysis (Figures 4G-4J), on average, the animals exhibited faster speed in the first 2 seconds after run onset in the early licking trials. This analysis might partially align with reason a). Correspondingly, the PyrUp decay time constant scaled with the first lick timing, supporting its role in tracking the animal's internal estimation.

In the high/low speed trial analysis (Figures 4C-4F), trials with low mean speed on average exhibited slightly more early licks between 1-2s. This analysis might partially reflect reason b).

In addition, as mentioned by the reviewer, the initiation of predictive licking was more likely to correlate with the start of deceleration (Figures 4G-4H).

In summary, our data suggest a complex interaction between lick and speed, both of which can reflect the animal's internal estimation of distance/time to some extent. However, we would like to be cautious about drawing strong conclusions.

Please include the figure legends with the figures in future revisions. Line numbers would also be useful.

Reply to Reviewer:

We thank the reviewer for these suggestions. We have now included the figure legends with their corresponding figures and added line numbers throughout the revised manuscript to improve readability.